# Marine aerosol distributions from shipborne observations over the South China Sea: Diurnal variation characteristics and their controlling factors

Zhi Qiao[1,2,3], Shengcheng Cui[1,3,4,*], Huiqiang Xu[1,2,3], Xiaoqing Wu[1,3,4], Xiaodan Liu[1,2,3], Zihan Zhang[1,4], Mengying Zhai[1,2,3], Yue Pan[5], Tao Luo[1,4], Xuebin Li[1,4]

[1]Key Laboratory of Atmospheric Optics, Anhui Institute of Optics and Fine Mechanics, HFIPS, Chinese Academy of Sciences, Hefei 230031, China
[2]Science Island Branch of Graduate School, University of Science and Technology of China, Hefei 230026, China
[3]Nanhu Laser Laboratory, National University of Defense Technology, Changsha 410073, China
[4]Advanced Laser Technology Laboratory of Anhui Province, Hefei 230037, China
[5]School of Electronic Engineering, Chaohu University, Chaohu, 238024, China

*Correspondence to*: Shengcheng Cui (csc@aiofm.ac.cn)

**Abstract.** Marine aerosols critically influence Earth's radiation budget and climate dynamics through their spatial distributions and composition due to their production and transport processes. However, in situ observational datasets remain limited, particularly in the South China Sea (SCS). Based on our comprehensive shipborne measurements, this study presents a quantitative analysis of marine aerosol distributions and compositional variations between the offshore and pelagic regions over the SCS. Our data demonstrate a 120% increase in offshore aerosol number concentrations (NCs, $Dp < 10.37$ µm) relative to pelagic baselines, featuring 120% higher accumulation-mode particles ($Dp \leq 1.981$ µm) and 70% higher coarse-mode particles ($1.981$ µm $< Dp < 10.37$ µm), quantitatively confirming continental transport affects spatial distribution of marine aerosols. In contrast, in the pelagic regions, marine aerosols are virtually unaffected by continental source and distinctly represent characteristics of the local production. Meteorological analyses identified wind speed (WS) and sea surface temperature (SST) as primary regulators of NC. However, observed NC variations at fixed WS and SST values suggest additional controlling factors. We demonstrate that sea-air temperature differentials (SST-$T_{2m}$) exhibit a stronger correlation ($r = -0.82, p < 0.01$) with NC than the other meteorological parameters, where increased SST-$T_{2m}$ led to decreased marine aerosol production. This temperature gradient effect drives pronounced diurnal NC variations, with maximum differences of 35% observed between daytime, nighttime, and transition periods. These findings provide concrete evidence for the spatial and diurnal variability in marine aerosol distributions over the SCS, thereby further improving understanding of marine aerosol transport and production.

## Graphical Abstract

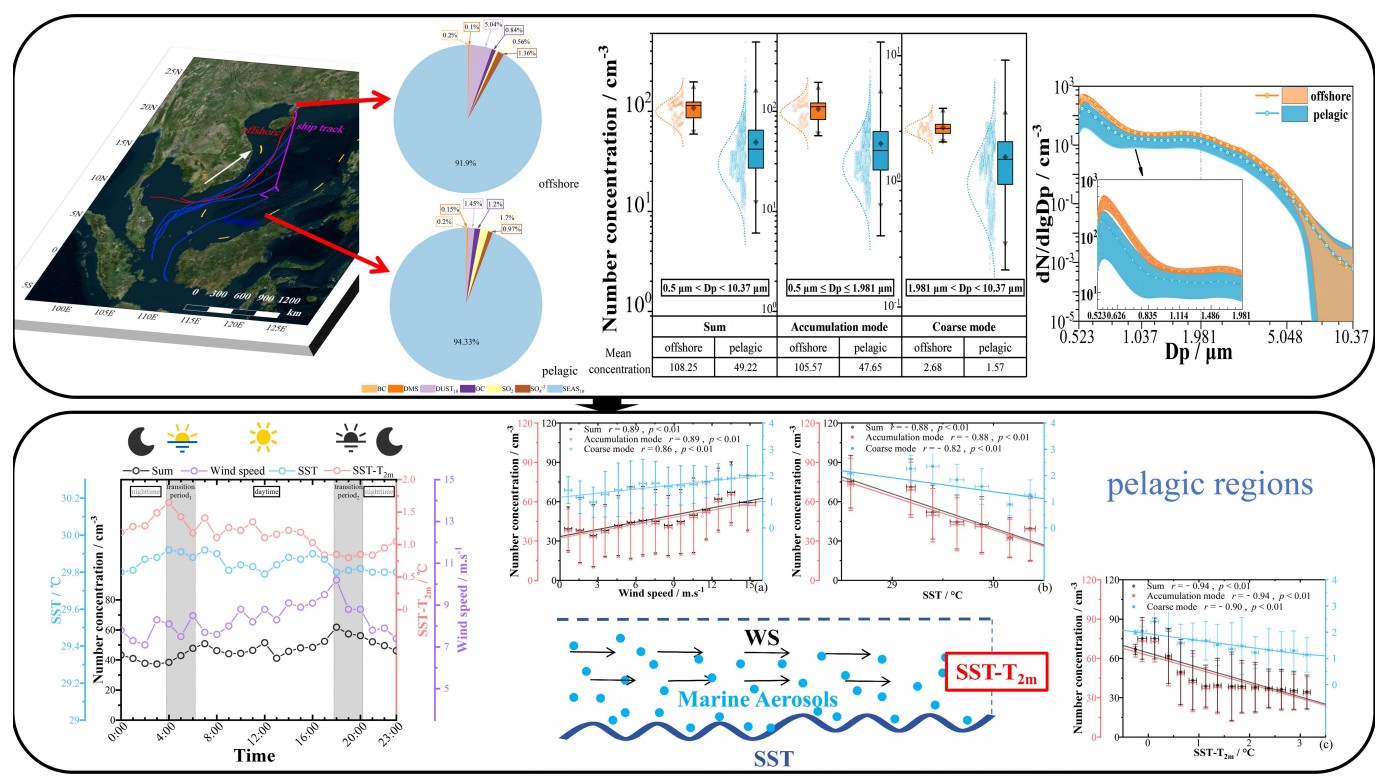

**Keywords: Shipborne observation; marine aerosol distributions; aerosol production and transport; sea-air temperature differences; the South China Sea**

## 1. Introduction

Atmospheric aerosols represent one of the largest uncertainties in the climate system projections for past and future (Andreae & Rosenfeld, 2008; Bauer et al., 2020; Bzdek et al., 2020). The ocean covers more than two-thirds of Earth's surface; marine aerosols are generated from the ocean surface and gas-to-particle conversion in the atmosphere (Korhonen et al., 2008). Globally, they are estimated to account for the largest proportion of natural aerosol emissions (Nascimento et al., 2021; Nguyen et al., 2017; Textor et al., 2006). Therefore, they represent an essential component of atmospheric aerosols. Marine aerosols are divided into two types: anthropogenic aerosols and natural aerosols. The natural category primarily includes sea salt particles and sea spray aerosols (Dedrick et al., 2022; Duce et al., 1965; Sander et al., 2003; Troitskaya et al., 2018). These aerosols modulate the radiative properties by influencing the indirect and direct radiation budget (Decesari et al., 2011; Myhre et al., 2004; Solomon et al., 2011; Woods et al., 2010). Additionally, they affect the nature of the marine cloud microphysics and precipitation patterns (Feingold et al., 1999; Levin et al., 2005; Woodcock, 1952, 1953) and drive

the geochemical cycles at the ocean surface (Alexander et al., 2005; Eriksson, 1960; Lawler et al., 2011; Long et al., 2014). As the essential aerosol type in the atmosphere, marine aerosols play a non-negligible role in the radiation budget. Thus, the role of marine aerosols in the climate system cannot be ignored (Li et al., 2022; Andreae & Crutzen, 1997).

Due to their non-negligible influence on both radiation budget and climate change, there has been an increasing research focus on marine aerosols over the last forty years. Early observations by Prospero (1979) across multiple marine areas showed notable variations in marine aerosol concentrations, ranging from 3.34 to 8.71 µg m$^{-3}$. Subsequent data verify substantial regional marine aerosol concentration differences between different ocean regions. In polar regions, submicrometer aerosol ($Dp \leq 1000$ nm) mass concentrations averaged 0.76 µg m$^{-3}$ in the Arctic (Leck & Persson, 1996) versus 3.15 µg m$^{-3}$ in the Antarctic (Savoie et al., 1993). In the Pacific Ocean, the PM$_{2.5}$ ($Dp \leq 2500$ nm) concentration averaged 12.3 ± 9.1 µg m$^{-3}$ in the Western Pacific (Ma et al., 2022) versus 140 ± 48.1 µg m$^{-3}$ in the Bohai Sea (Han et al., 2019). In the Indian Ocean, Pant et al. (2009) observed that the average accumulation- to coarse-mode aerosols (500 nm ≤ $Dp \leq 10000$ nm) mass concentrations were 8.89 µg m$^{-3}$. In addition to aerosol mass concentrations, researchers have also observed aerosol number concentrations (NCs) differences. For instance, in marine regions off the coast of China, Kim et al. (2009) found that the average submicrometer aerosol particle (10 nm ≤ $Dp \leq 300$ nm) concentrations were 4335 ± 2736 cm$^{-3}$ over the East China Sea and 5972 ± 2736 cm$^{-3}$ over the Yellow Sea. In summary, there are differences in marine aerosol mass concentrations and NCs between the different ocean regions. However, most available marine aerosol measurements for the SCS come from coastal monitoring stations, while shipboard observations remain sparse and outdated (Kong et al., 2016; Su et al., 2022). Given that shipboard measurements can provide better spatial and temporal context for marine aerosol measurements across diverse ocean areas such as the SCS, expanding and updating such shipboard observations has the potential to improve the characterization of marine aerosol in these regions.

Aerosol transport and production can lead to differences in marine aerosol concentration and size distribution. Some studies revealed that marine aerosol composition (e.g. sea salt, dust, sulfate, organic carbon) and particle size distribution are influenced by both mesoscale weather events (e.g. thunderstorm, sea breeze, typhoon) and continental transport (Athanasopoulou et al., 2016; Chen et al., 2018; Croft et al., 2021; O'Dowd & De Leeuw, 2007; Sakerin et al., 2015; Sellegri et al., 2006). The SCS, one of the largest marginal seas, is located on the continental margin and separated from the open ocean by islands or island arcs. It is significantly influenced by continental and anthropogenic aerosols transported through continental air masses. Previous studies reveal that continental and anthropogenic aerosols play an important role in determining aerosol concentration and size distribution (Braun et al., 2020; Wu & Boor, 2021). Liang et al. (2021) observed an increase in submicron aerosol NCs and a different number size distribution shape (20 nm ≤ $Dp \leq 400$ nm) when observational data were influenced by continental transport in the SCS. Atwood et al. (2017) further found that under continental transport, the number size distribution exhibits a unimodal structure (20 nm ≤ $Dp \leq 500$ nm). In contrast, a distinct bimodal size distribution (20 nm ≤ $Dp \leq 500$ nm) emerges without continental transport. Given the limited observational data and previous focus on the submicron size range, it is crucial to conduct studies on the impact of aerosol transport on larger aerosol particles ($Dp \geq 500$ nm) for a more thorough understanding of how transport influences size

distributions.

Furthermore, some key meteorological parameters of the air-sea interface could affect aerosol production and transport, such as wind direction (WD) and speed (WS), relative humidity (RH), and sea surface temperature (SST) (Dasarathy et al., 2023; Carslaw et al., 2010; Hoppel, 1979; Hoppel et al., 1985). Previous studies found that WS is the major driver of production and transport of marine aerosols. Some subsequent studies attempted to link NCs to observed WS (Andreas, 1998, 2010; Gong, 2003; Ovadnevaite et al., 2014; Smith et al., 1993; Yang et al., 2019). These studies derived source functions based on the relationship between aerosol particle size distribution and WS, thereby enabling the simulation of number size distribution and total aerosol NCs. Some studies revealed that rising RH increases particle dry deposition rates (Arimoto & Duce, 1986; Lo et al., 1999), which are important to aerosol transport, as higher dry deposition rates reduce the residence time of aerosols in the atmosphere and shorten their transport distance therein. Ding et al. (2021) found that elevated RH enhances secondary aerosol (e.g. nitrate and sulfate) formation, which directly affects aerosol production. Therefore, RH also affects aerosol transport and production. In addition, SST dramatically influences the production of marine aerosols by affecting bubble bursting time and jet drop production efficiency (Zábori et al., 2012a). Jaeglé et al. (2011) and Mårtensson et al. (2003) further revealed that warmer SST might reduce seawater density and surface tension, ultimately leading to higher marine aerosol production. The reduced surface tension increases wave breaking efficiency, entraining more air into seawater to form bubbles. In addition, the reduced seawater density leads to more bubbles rising back to the sea surface. As these bubbles reach the surface and burst, they subsequently form marine aerosols. However, previous studies indicated that sea-air temperature differentials (SST-$T_{2m}$) influence the air-sea interaction through air-sea heat exchanges and turbulent mixing (O'Neill et al., 2010); meanwhile, it can comprehensively reflect the characteristics of the ocean and atmosphere near the sea surface (Jing et al., 2019; Ma et al., 2016). Hence, SST-$T_{2m}$ might affect marine aerosol production and transport, but the exact effects of SST-$T_{2m}$ on marine aerosols need further investigation. To better quantify and understand the effect of these meteorological parameters on marine aerosols, more thorough information about the variations of marine aerosol and these factors, especially regarding SST-$T_{2m}$, is needed in the SCS. In addition, the diurnal scale of marine aerosol variation can provide valuable information about their production and transport (Flores et al., 2021), and how these processes are influenced by meteorological parameters. Understanding the diurnal variation is also crucial for improving atmospheric models. Studies on the scale of diurnal variation in marine aerosol remain scarce, and there is an urgent need to clarify the specific connection between these diurnal variations and meteorological parameters to better understand aerosol production and transport.

To address these, we acquired and updated observations of marine aerosol and meteorological parameters over the SCS, then quantitatively compared marine aerosol composition and distributions in the offshore and pelagic regions, as well as influence of aerosol transport on marine aerosol. Subsequently, temporal variations of shipborne observational data were investigated in detail; meanwhile, the differences in distribution of marine aerosol in diurnal variations, especially the diurnal transition, were further analyzed. Based on these analyses, specific relationships between different meteorological

parameters and marine aerosols were examined. Finally, overall results of marine aerosol particle size distributions and NCs in the SCS, as well as the possible influence factors were summarized.

## 2. Cruise observation and data analysis

### 2.1. Cruise details

In May and June 2023, a scientific cruise was conducted in the SCS by the South China Sea Institute of Oceanology, Chinese Academy of Sciences, onboard the *Yuezhanyuke No. 6*. This study analyzes the aerosol-meteorology (AM) measurements along the section from the latitude 21°02′ N to 8°5′ N and the longitude 110°33′ E to 115°25′ E. All AM data were collected from 21 May to 15 June 2023.

### 2.2. Instrument setup

**2.2.1. Aerosol sampling instrument**

The NCs of aerosol particles were measured with the Model 3321 Aerodynamic Particle Size (APS) spectrometer (TSI Incorporated, USA), which has 52 size channels in the 0.5 to 20 μm diameter range. This Model 3321 APS spectrometer employs relative light-scattering intensity along with sophisticated time-of-flight techniques; two complementary techniques can measure the information of each aerosol particle to obtain aerosol concentrations and 

distributions. We used the Particle Loss Calculator (PLC) to calculate particle losses for the APS in this cruise (Fig. S1) (Von Der Weiden et al., 2009). Fig. S1 revealed a dramatic increase in aerosol particle loss at particle diameters exceeding 10 μm. Meanwhile, the accuracy of aerosol data for particle diameters between 0.5 μm and 10 μm, as measured by the APS, had been fully validated in previous studies (Pagels et al., 2005; Peters et al., 2003; Peters, 2006). Therefore, aerosol data within size range of 0.5 to 10 μm were selected for analysis in this study.

To further verify the accuracy of APS aerosol measurement results, we conducted a 15-day field inter-comparison experiment, using multiple aerosol instruments to validate the APS. A detailed description of this field inter-comparison experiment was provided in Supplement S1. By comparing aerosol size distributions measured by the three instruments (Fig. S2a), the good consistency confirms the accuracy of the APS in capturing aerosol particle distributions. Since direct channel-to-channel matching was not feasible due to the differing size bins for different aerosol measurement instruments, we

compared summed NCs within overlapping ranges: 0.5–1.981 μm for APS, 0.475–1.99 μm for Portable Optical Particle Spectrometer (POPS) (Handix Scientific, USA), and 0.488–2.14 μm for Model 11-D Portable Aerosol Spectrometer (GRIMM, Germany). All three instruments exhibited consistent diurnal trends (Fig. S2b). Fig. S3 showed high correlations between APS and the other instruments. The consistent trends and strong correlations further validate accuracy of the APS. Furthermore, during instrument channel matching, we observed that the APS lacks the standard 2 μm size bin typically used

to distinguish accumulation mode and coarse mode particles. The closest available diameter in APS channels is 1.981 μm. A

distinct peak consistently appeared at this size in aerosol size distributions. Based on these observations, we established 1.981 μm as the threshold for separating accumulation mode ($0.5\ \mu m \le Dp \le 1.981\ \mu m$) from coarse mode ($1.981\ \mu m < Dp < 10.37\ \mu m$) aerosols.

The APS was loaded in the captain's cabin and coupled with a 10 cm long tube and a 1.8 cm internal diameter, as shown in Fig. 1a. The tube was fixed in an exterior wall of the captain's cabin at 30° to the horizontal and faced the direction of the sea surface. Meanwhile, the tube's inlet was approximately 7 m above the mean water level, and this location was thought to be less affected by human factors and bow splashing. The flow rate was 1.0 liter per minute, and sample length was 15 seconds. The atmospheric aerosol data resolution was set to 5 min in this SCS cruise observation. The detailed definition and calculation formula for aerosol number concentration were shown below.

$$N = \frac{C}{LQ} \times \frac{H}{K},$$

where $N$ is the number concentration per channel, $C$ is the particle counts per channel, $L$ is the total sample time, $Q$ is the sample flow rate, $H$ is the sample dilution factor, and $K$ is the sample efficiency factor per channel.

### 2.2.2. Meteorological instruments

The automatic meteorological observation system (AMOS), including the Vaisala WXT530 weather station, the Campbell CSTA3B sonic anemometer, and the Belfort Model 6400 visibility sensor, was installed on the top deck to continuously collect meteorological observational data. The height of the AMOS above the mean water level was approximately 10 m, as shown in Fig. 1b. The Vaisala WXT530 measured atmospheric parameters such as air temperature ($T_{OBS}$), RH and rainfall intensity with a temporal resolution of 1 s. The two-dimensional wind field (i.e. horizontal components $u_x$ and $u_y$) was measured by the Campbell CSAT3B, with its temporal resolution being 0.05 s. To support ancillary research, the Belfort Model 6400 observed atmospheric visibility (VIS) with a temporal resolution of 1 s. More detailed specifications of the AMOS were provided in Table 1.

**Table 1**

*Configurations and specifications of AMOS*

| WXT530 | | CSTA3B | | Model 6400 | |
|---|---|---|---|---|---|
| Performance index | Description | Performance index | Description | Performance index | Description |
| Observation Range ($T_{OBS}$) | -52–60 °C | Observation Range (WS) | 0–60 m s$^{-1}$ | Observation Range (VIS) | 0–50 km |
| Resolution ($T_{OBS}$) | 0.1 °C | Resolution (WS) | 0.1 m s$^{-1}$ | Resolution (VIS) | 0.1 km |
| Accuracy ($T_{OBS}$) | ±0.3 °C | Accuracy (WS) | ±0.3 m s$^{-1}$ | Accuracy (VIS) | ±1 km |
| Observation Range (RH) | 0–100 % | Observation Range (WD) | 0–360° | | |
| Resolution (RH) | 0.1 % | Resolution (WD) | 1° | | |

| | | | |
|---|---|---|---|
| Accuracy (RH) | ±3 % | Accuracy (WD) | ±3° |
| Observation Range (Rain) | 0–200 mm h$^{-1}$ | | |
| Resolution (Rain) | 0.1 mm h$^{-1}$ | | |
| Accuracy (Rain) | ±0.5 mm h$^{-1}$ | | |

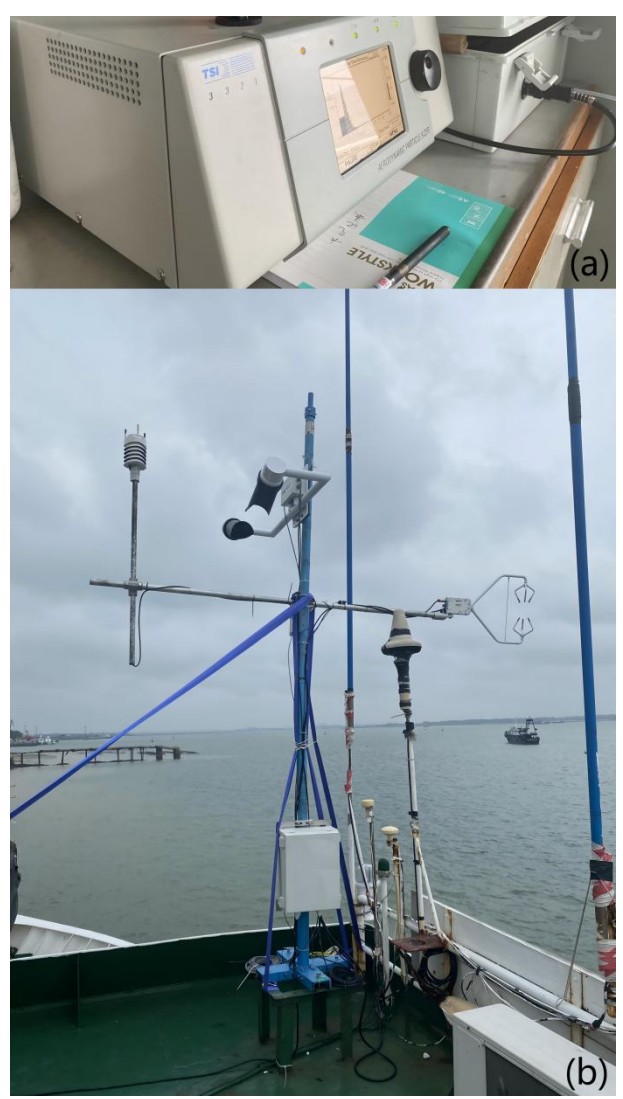

**Fig. 1 The total view of (a) the Model 3321 APS spectrometer and (b) the automatic meteorological observation system.**

### 2.3. Auxiliary data

#### 2.3.1. Reanalysis data

In this study, 10-m wind speed ($WS_{10}$), direction ($WD_{10}$), and friction velocity ($U_{zust}$) were obtained from the ERA5 hourly dataset with a spatial resolution of $0.25° \times 0.25°$. The ERA5 hourly dataset used in this study was provided by the European Centre for Medium-Range Weather Forecasts (ECMWF) (Hersbach et al., 2023); ERA5 was selected for these dynamical properties due to better in situ agreement (Li et al., 2025). To determine values of SST-$T_{2m}$, we needed to know temperature at 2 m ($T_{2m}$) and SST. The Modern-Era Retrospective Analysis for Research and Applications, Version 2 (MERRA-2) provides reanalyzed SST and $T_{2m}$ data that show excellent agreement with observational data in the SCS ($r > 0.9$) (Jiang et al., 2021). We selected SST and $T_{2m}$ data from the MERRA-2 meteorological dataset in this context (Gelaro et al., 2017).

For atmospheric aerosol composition, the NASA Goddard Space Flight Center MERRA-2 aerosol dataset was used in this study due to its good performance over Europe and China (Provençal et al., 2017a, 2017b). The MERRA-2 aerosol dataset includes the assimilated aerosol diagnostics data, such as surface mass concentrations of aerosol composition (e.g. sea salt ($SEAS_{10}$; $Dp \leq 10$ μm), dust ($DUST_{10}$; $Dp \leq 10$ μm), black carbon (BC), sulfate ($SO_4^{2-}$), and organic carbon (OC)) with a spatial resolution of $0.5° \times 0.625°$ and a temporal resolution of 1 hour (Randles et al., 2017). We used above aerosol composition data to discuss differences in aerosol distribution over the SCS.

#### 2.3.2. Back trajectory analysis

The Hybrid Single-Particle Lagrangian Integrated Trajectory (HYSPLIT) transport and dispersion model (http://www.arl.noaa.gov/ready/hysplit4.html), developed by the National Oceanic and Atmospheric Administration Air Resources Laboratory (NOAA ARL), was employed to analyze air mass backward trajectories. The meteorological data for the backward trajectories were obtained from the Global Data Assimilation System (GDAS) archive dataset (http://ready.arl.noaa.gov/gdasl.php). The backward trajectories were calculated for 72 hours. The trajectories were calculated at an altitude of 10 m above ground level to match the instrument sampling height, and the top of the HYSPLIT model was set to 5,000 m to clarify the influence of the source region on the marine aerosols.

#### 2.3.3. Distances from the coast

The ArcGIS path distance method was used to calculate distances from the coast. In equidistant projection, ship positions were used as input data, and coastline position data were used as reference lines for distance analyses. Considering the actual surface distance as well as horizontal and vertical factors, the shortest distance from the ship to the coastline can be calculated.

## 2.4. Contaminated data screening

To observe actual aerosol loadings in the typical marine environment, it is necessary to exclude aerosol observational data contaminated by ship emissions during the cruise (including data collected in offshore and pelagic regions). To achieve this, we performed quality control during aerosol observational data collection with accompanied WS, WD, rainfall intensity, and NC observational data as follows:

i) *Data screening with wind observations*. The north arrow of the Vaisala WXT530 was oriented perpendicular to the ship's longitudinal axis, pointing east relative to the vessel's heading. Observed WD thus represented the relative angle between the wind direction and the ship's course. When the ship was sailing, WD sampled ranged from 225° to 315°. We excluded these data because aerosol observations were directly affected by the ship emissions. As depicted in Fig. 2a, a sharp decrease and subsequent increase in the observed aerosol size distributions can be easily identified on 25 May. The underlying cause of abnormal changes in aerosol size distribution could be the removal of coarse aerosol mode by rainfall, as depicted in Fig. 2b. However, this "jump" was located at 7:00 a.m. (circled in black in Fig. 2a) while the rainfall occurred at 8:00 a.m. and lasted for approximately 6 hours, so it is not the rainfall that influences the data jump in aerosol size distribution. With the help of WD sequence data shown in Fig. 2b (circled in red), we found that the jump in WD curve was also located at 7:20 a.m. and thus identified the accurate cause of aerosol data jump; it was wind direction rather than rainfall that led to the anomaly in aerosol size distributions.

ii) *Further data screening with unreasonable NCs*. Aerosol NCs remain relatively constant under stable meteorological conditions (Hoppel, 1979; Hoppel et al., 1985; Russell et al., 1996). In the presence of continental transport, sustained high NCs would persist for several hours (Saha et al., 2022; Wang et al., 2020). Therefore, to further screen out the possible influence of ship emissions, we excluded data points where NCs exhibited a sharp short-term fluctuation (i.e. one order of magnitude higher or lower than the average NCs at that time) in the absence of changes in meteorological parameters and influences of continental transport.

Applying these criteria, 88 % of the observational data in this cruise were retained for analysis.

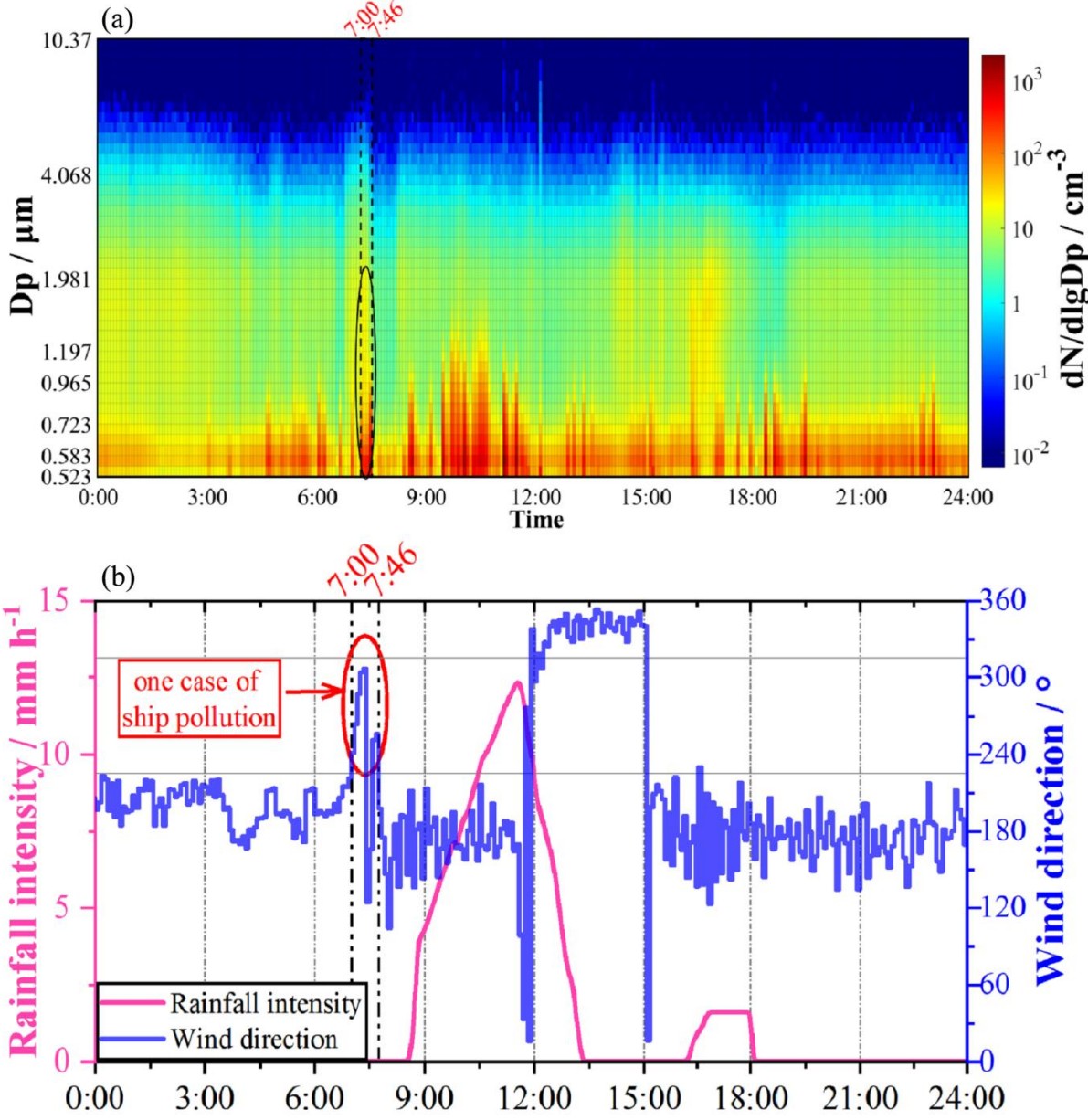

**Fig. 2 The time series of the observations on 25 May 2023. The black circle represented one case of ship pollution. (a) Trends of the aerosol size distributions. (b) Trends of the rainfall intensity and the WD.**

## 3. Results and discussion

### 3.1. Temporal distributions of the observations

Fig. 3 showed the time series of marine aerosol distributions and meteorological parameters during the observation shipboard in the SCS. Due to the pump of the Model 3321 APS spectrometer failing to work from 4 June to 15 June 2023 during the cruise period, flow rate could not reach the minimum standard. Thus, we only analyzed variations of observational data from 21 May to 3 June 2023. Fig. 3 a-b presented the trends of aerosol size distribution and comparison of the accumulation ($0.5\ \mu m \leq Dp \leq 1.981\ \mu m$) and coarse ($1.981\ \mu m < Dp < 10.37\ \mu m$) mode particle NCs. During the shipboard observation period, the summed ($Dp < 10.37\ \mu m$) NC varied from 18.46 to 89.38 cm$^{-3}$, NC of accumulation mode varied from 17.39 to 87.31 cm$^{-3}$, and NC of coarse mode varied from 0.83 to 2.49 cm$^{-3}$. The NCs exhibited substantial temporal fluctuations. The average values for summed NC, accumulation mode NC, and coarse mode NC were 54.01 cm$^{-3}$, 52.35 cm$^{-3}$, and 1.66 cm$^{-3}$, respectively. Published observational data of marine aerosol NC relevant to this study were shown in Table 2. The shipboard observational data showed overall average values and standard deviations of maritime aerosol NCs under different temporal and geographical conditions. We used these data to compare with the marine aerosol NCs during this cruise period.

In this study, observed NC of accumulation mode was lower than the observation in the East China Sea (Lin et al., 2007; Ma et al., 2022). Aerosol emissions from the Yangtze River Delta region are higher than those from the Pearl River Delta region (Li et al., 2017). Due to the influence of aerosol transport, a greater amount of continental and anthropogenic aerosols from the Yangtze River Delta were delivered to the East China Sea compared to the amount transported from the Pearl River Delta to the South China Sea. Hence, the accumulation mode NC was significantly lower in the SCS. The SCS is one of the marginal seas of the Western Pacific. The average summed NC observed in this study (54 cm$^{-3}$, $0.5\ \mu m \leq Dp \leq 1.98\ \mu m$) was slightly lower than the NC reported for the Western Pacific (83 cm$^{-3}$, $0.1\ \mu m \leq Dp \leq 1.98\ \mu m$) by Flores et al. (2020). Notably, this study focused on particles from 0.5 to 1.98 μm, while Flores et al. (2020) included smaller particles ($0.5\ \mu m \geq Dp \geq 0.1\ \mu m$). Differences in the measured particle size ranges are a potential leading cause of the differences in marine aerosol NCs. Regarding marine aerosol NCs in the same ocean regions, the observations of Cai et al. (2020) and Kong et al. (2016) were significantly higher than those in this study. Although the differences in observation seasons and particle size ranges might influence the average NC observations, these differences may still indicate that the marine aerosol was affected by the continental transport and the anthropogenic activity in the offshore regions, based on the latitude and longitude.

Wet deposition through scavenging by rainfall is a critical sink for aerosols (Atlas & Giam, 1998; Radke et al., 1980). However, intense precipitation events might paradoxically elevate aerosol NCs. Some studies indicate that impaction of liquid droplets on porous surfaces (e.g. the ship surfaces) may generate aerosol particles (Bird et al., 2010; Joung & Buie, 2015; Zhou et al., 2020). By accounting for observation environment and rainfall intensity, short-duration heavy rainfall resulted in numerous raindrops impacting the ocean and ship surfaces, generating aerosol particles. Subsequently, the monitoring instrument captured some of these aerosol particles, ultimately contributing to the increased aerosol particle

concentration observed in Fig. 3 (the blue-shaded regions). In addition to the elevated concentrations of marine aerosols resulting from these rain events, the aerosol NC spectrum distributions shown in Fig. 3a demonstrate continuous marine aerosol NC distributions in the size ranges of 523 to 583 nm, 1715 to 1981 nm, and 3786 to 4371 nm during the cruise period. This indicated background characteristics of marine aerosol particle distributions in the marine environment. During the cruise period, comparisons of the time series of NCs between the two aerosol particle modes were made, as shown in Fig. 3b. The temporal trend of the NC of accumulation mode was approximately consistent with the coarse mode. The correlation coefficient between these two modes was approximately $r = 0.71$. However, there were some differences in the NC of the different particle sizes in aerosols caused by the different marine aerosol sources. We also found that the temporal trend of accumulation mode was more variable than that of coarse mode, suggesting the accumulation mode may be more obviously influenced by changes in the marine environment.

For the meteorological parameters, the ship remained in the northeast trade winds during the entire cruise period; therefore, this mainly led to south-westerly and southerly winds, as shown in Fig. 3c. Fig. 3d represented air temperature and water temperature. The observed air temperature was consistent with MERRA-2 reanalyzed air temperature ($r = 0.719$), whereas the average $T_{2m}$ and SST over the whole observation period were closer to 29.0 °C and 29.7 °C, respectively. RH had a negative correlation with the visibility ($r = -0.74$), and the average RH and VIS were equal to 83.0 % and 45.1 km. Since wind is a major driver of marine aerosol production and transport, we attempted to explain marine aerosol NCs of the accumulation mode and the coarse mode as the functions of the WS and WD (Fig. 4a, b). The RH and the rainfall intensity observations were used to aid analysis (Fig. 4c, d). High NCs ($\geq 150$ cm$^{-3}$) were observed almost entirely in which the WD were between NW and N that were caused by the high RH accompanied by the rainfall events, and distributions of NCs were uniform when the wind was blowing in the other directions.

Fig. 5 showed variations of NCs of two aerosol particle modes with WS, and observational data were binned at 3 m s$^{-1}$ WS intervals. The variations in NCs with WS observed in this study were consistent with a previous study (Bruch et al., 2023). For example, marine aerosol NCs generated by the bubble bursting process at low WS showed little variation, and the low WS was insufficient to activate spume droplet production (Pietsch et al., 2018; Russell et al., 2023). Consequently, no significant variation in NCs were observed in the 0–6 m s$^{-1}$ WS range. NCs increased with the increase in WS from 6 m s$^{-1}$ to 15 m s$^{-1}$; however, the increase slowed down when WS exceeded 13 m s$^{-1}$. Previous study proposed that this phenomenon may be linked to the scavenging of marine aerosols through collision by the larger water drops at high WS (Pant et al., 2009).

**Table 2**

*Summary of the available study results on the shipboard observation of marine aerosol NC (cm$^{-3}$)*

| Region | Time | Season | Latitude | Longitude | Parameter | Value | Parameter | Value | Reference |
|---|---|---|---|---|---|---|---|---|---|
| South China Sea | 2023.05–2023.06 | Spring | 21° N–8° N | 115° E–110° E | Accumulation mode ($n_{500-1981}$) | $52.4 \pm 35.0$ | $n_{500-10000}$ | $54.0 \pm 35.3$ | This Study |

| | | | | | Parameter | | | | Reference |
|---|---|---|---|---|---|---|---|---|---|
| South China Sea | 2018.08 | Summer | 23° N– 19° N | 118° E– 108° E | $n_{400-32000}$ | 61 | | | Cai et al., 2020 |
| South China Sea | 2012.09– 2012.10 | Autumn | 21° N– 20° N | 118° E– 113° E | $n_{120-10000}$ | 175 | | | Kong et al., 2016 |
| South China Sea | 2005.05 | Spring | 20° N– 18° N | 118° E– 113° E | Accumulation mode ($n_{50-2000}$) | $50.3 \pm 19.5$ | | | Lin et al., 2007 |
| East China Sea | 2005.05 | Spring | 30° N– 26° N | 122° E– 117° E | Accumulation mode ($n_{50-2000}$) | $109.2 \pm 51.8$ | | | Lin et al., 2007 |
| East China Sea | 2017.04– 2017.05 | Winter | 28° N– 20° N | 130° E– 120° E | $n_{250-2500}$ | $57.4 \pm 40.9$ | $n_{2500-10000}$ | $57.5 \pm 41.3$ | Ma et al., 2022 |
| Western Pacific | 2017.04– 2017.05 | Spring | 20° N– 0° N | 180° E– 130° E | $n_{100-19800}$ | $83 \pm 30$ | | | Flores et al., 2020 |

*Note.* In the column of the "Parameter", "*n*" indicated the NC and the subscripts indicated the particle size (nm); in the column of the "Latitude", "N" represented north latitude. The results of this study and these references were the overall 285 average aerosol NCs.

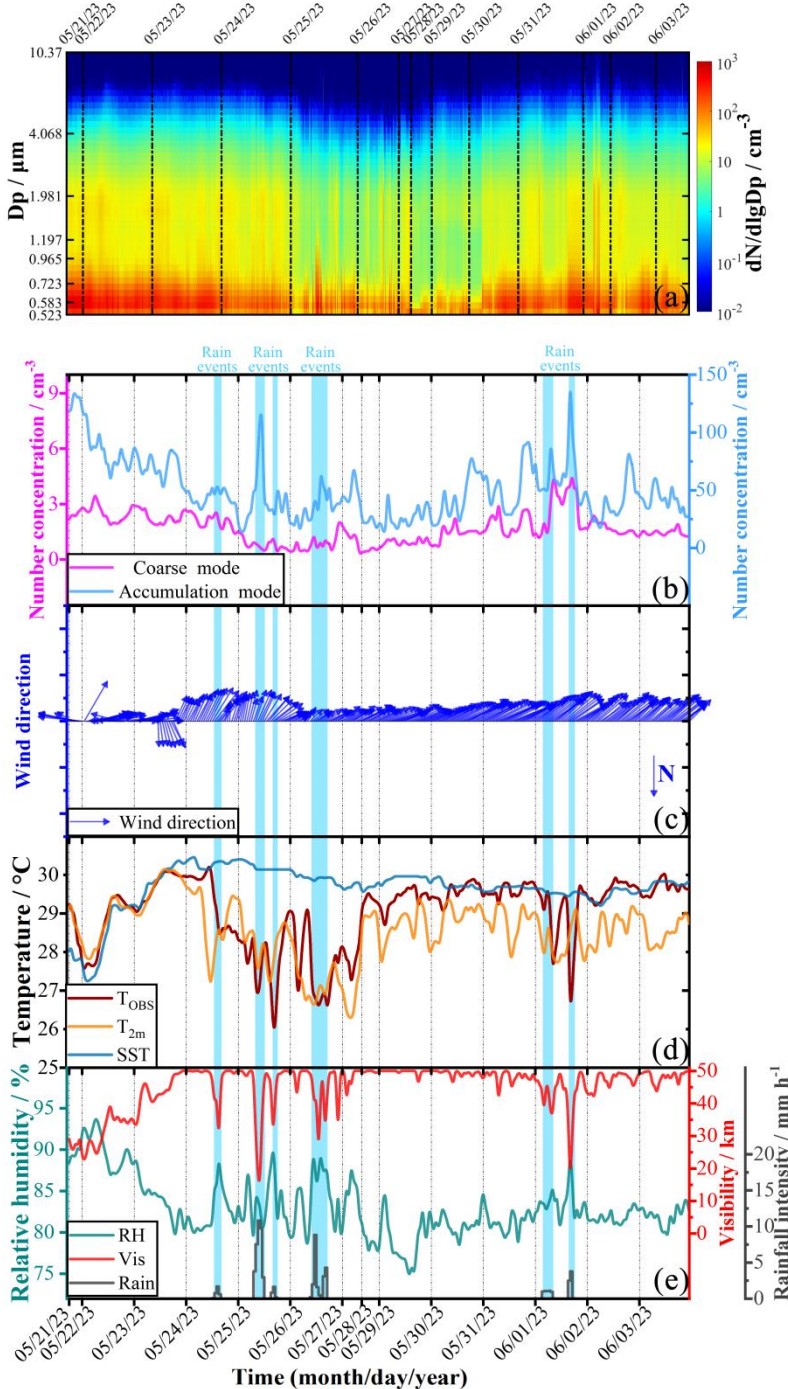

**Fig. 3 The time series of the shipboard observations in the SCS from 21 May to 3 June 2023. The blue-shaded regions represented periods affected by rain events. (a) Trend of the aerosol size distributions. (b) Trends of NCs of the two aerosol particle modes (black solid line represented the NC of the coarse mode, and red solid line represented the NC of the accumulation mode). (c)**

**Trend of the WD. (d) Trends of the $T_{OBS}$ (dark orange solid line), $T_{2m}$ (light orange solid line), and SST (blue solid line). (e) Trends in the RH (gray solid line), the VIS (red solid line), and the rainfall intensity (dark blue solid line).**

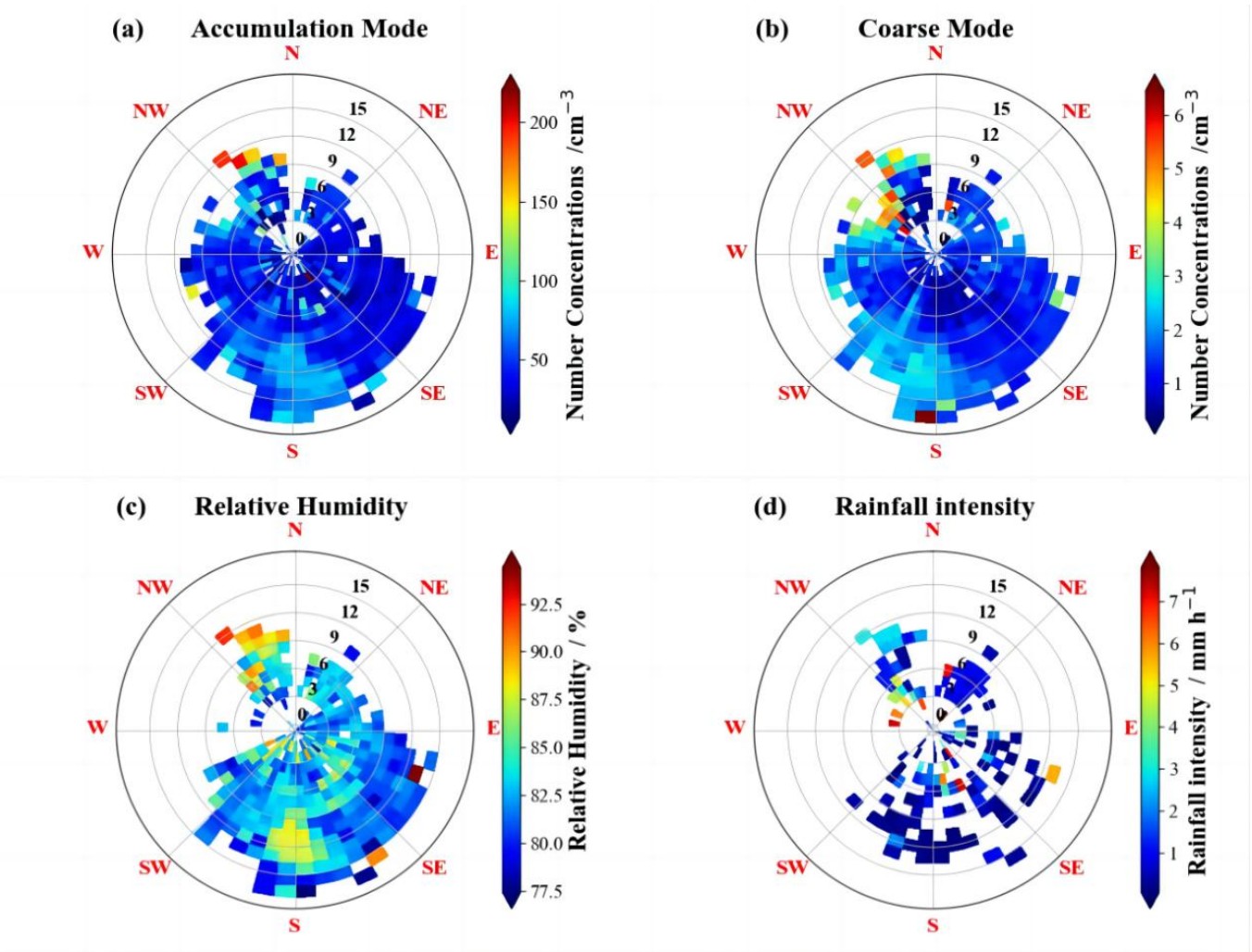

**Fig. 4 (a) NC of the aerosol accumulation mode, (b) NC of the aerosol coarse mode, (c) RH, and (d) rainfall intensity as the functions of the WS and WD for the observations in the SCS.**

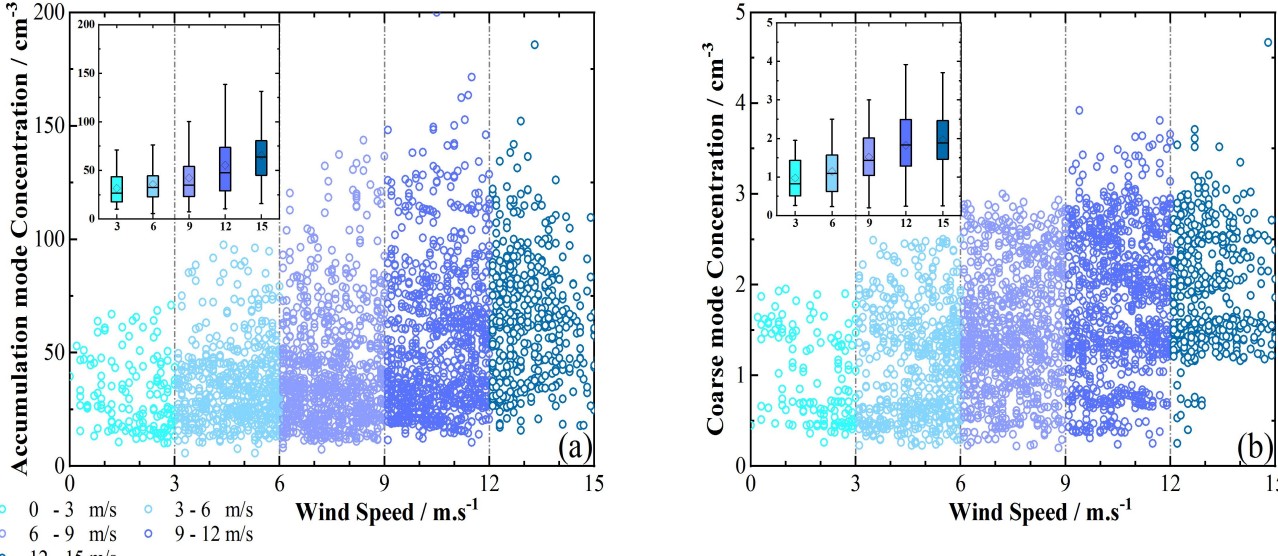

**Fig. 5 The scatter plots of (a) NCs of the aerosol accumulation mode and WS, (b) NCs of the aerosol coarse mode and WS. The observational data were binned to the WS intervals equal to 3 m s$^{-1}$; the boxes represented the 25th to 75th percentile value, the black whisker represented the 1.5 inter-quartile range, the black diamond marker represented the mean value, and the black horizontal line represented the median value in the box plots.**

### 3.2. Marine aerosol distributions in the different distance from the coast

This marine scientific research campaign started southward from the harbor of Zhanjiang (21°16′21.12″ N, 110°23′45.17″ E) on 21 May and reached up to the southernmost (8°5′ N) point of this cruise on 3 June. In different latitudes of the SCS, there were vastly different marine aerosol distribution characteristics, meteorological parameters, and marine aerosol transport sources. Therefore, we assessed the features of marine aerosol distribution at various distances from the coast. We conducted real-time analysis of the 72-hour backward trajectories of air masses at the ship's location (Fig. 6a, b). The backward trajectory analysis indicated that the air masses had last passed over continental areas on 22 May 2023, 11:00 local time (LT), at a point 50 km from the coast (the red solid lines in Fig. 6b). Consequently, for all sampling locations within this 50 km boundary, the air masses had directly passed over mainland areas. This meant they carried continental and anthropogenic aerosols that ultimately influenced the aerosol distributions (Braun et al., 2020; Wu & Boor, 2021). For regions more than 50 km from the coast, the backward trajectory results consistently showed that the air masses did not pass over any mainland areas before reaching the sampling site (the blue solid lines in Fig. 6a). The prevailing wind direction was primarily from the southwest (Fig. 3c) in these regions, so aerosols could not be directly transported from the continent to the ship's location. Additionally, continental and anthropogenic aerosols, which were emitted from islands and countries surrounding the SCS, lost their original characteristics through the long-duration (over 72 hours) transport. These aerosols likely underwent dry deposition, wet deposition, and aging processes associated with long-range transport. Such processes could have led to the removal of continental aerosols or their gradual dilution and mixing with natural aerosols (Hodshire et al., 2019; Ohata et

al., 2016; Xu et al., 2021). Over time, the continental and anthropogenic aerosols may have transformed into or been integrated with the background aerosols. Hence, 50 km from the coast was taken as the boundary distance to distinguish offshore and pelagic regions in this study. We could analyze differences in aerosol transport and production in offshore and pelagic regions. To eliminate the effects of rainfall on aerosol concentrations, we removed aerosol data associated with rainfall intensity greater than 1 mm h$^{-1}$ during the observation period.

Fig. 7 revealed marine aerosol distribution characteristics with different modes in the SCS. From Fig. 7a, we can find that NCs of different aerosol particle modes in offshore regions showed significant differences from those in pelagic regions. The average summed NC in offshore regions during the cruise period was 108.25 cm$^{-3}$, registering a 1.2-fold increase compared to NC in pelagic regions (49.22 cm$^{-3}$). The NC of accumulation mode in offshore regions was 105.57 cm$^{-3}$, and it was 47.65 cm$^{-3}$ in pelagic regions. The average accumulation mode NC in offshore regions was 2.2 times higher than that in pelagic regions, with a statistically difference ($p < 0.001$). Similarly, the comparison of the coarse mode NC revealed the same result as that for the accumulation mode. The coarse mode NC in the offshore regions (2.68 cm$^{-3}$) was also significantly higher than that in pelagic regions (1.57 cm$^{-3}$), with a statistically significant difference ($p < 0.001$). Fig. 7b showed the number size distribution for marine aerosols of 0.5 to 10 μm diameters in the offshore and pelagic regions, where d$N$/dlg$D_p$ represents the number size distribution. The comparisons in Fig. 7b showed that the number size distributions in the offshore and pelagic regions exhibited a bimodal distribution, and the peak values both occurred at 0.542 and 1.981 μm, which were consistent with the previous studies (Andronache, 2003; Braun et al., 2020). The number size distributions exhibited a relatively stable value in the 0.835–1.981 μm particle size range. Due to aerosol transport, continental air masses may have carried continental and anthropogenic aerosols, which could have ultimately affected aerosol distributions in the 0.5–5.0 μm particle size range. The number size distributions in the offshore regions were obviously higher than in the pelagic regions in the 0.5–5.0 μm particle size range. These findings were consistent with previous studies (Braun et al., 2020; Lorenzo et al., 2023). However, in the 5.0–10 μm particle size range, the number size distributions from offshore and pelagic regions were largely consistent, and this consistency remained robust against instrumentation limitations. This comparability persists despite APS measurements in this range having inherent uncertainties reaching up to 130% (Pfeifer et al., 2016), primarily due to inefficient particle detection at concentrations approaching 1 cm$^{-3}$. Throughout the cruise, continuous 5-second sampling yielded 64,180 valid samples, which through statistical averaging reduced measurement uncertainty to 0.5%. At this negligible level, the distribution characteristics and cross-regional correlations are considered reliably preserved.

Fig. 7c revealed a decreasing trend in NCs with increasing distance from the coast, and the correlation coefficients between the daily average NCs of accumulation mode, coarse mode, and the summed NCs and the distance from the coast were calculated as $r$ = -0.87, -0.67, and -0.81, respectively. The correlation analysis, based on hourly average NCs of accumulation and coarse modes versus the distance from the coast, yielded $r$ = -0.59 and -0.50 for offshore regions, and $r$ = -0.28 and -0.33 for pelagic regions. The same was true for the summed NC; the correlation coefficient between the hourly average summed NC and the distance was -0.56 in the offshore regions and -0.29 in the pelagic regions. These correlations indicated that the correlation between distance from the coast and NC was mainly reflected in the offshore regions and had

almost no significant correlation with marine aerosol in the pelagic regions. In offshore regions, aerosol NC was influenced by continental aerosol transport. This influence diminished with increasing distance from the coast. After the ship entered the pelagic regions, the influence of continental air mass transport almost disappeared. Marine aerosol NCs were lower in pelagic regions compared to offshore regions.

Table 3 revealed differences in meteorological parameters between offshore regions and pelagic regions. For example, the average WS in offshore regions was 10.74 m s$^{-1}$, slightly higher than that in pelagic regions (8.64 m s$^{-1}$). Higher WS can enhance marine aerosol production. Therefore, there was a higher production of aerosols in offshore regions. In addition, the offshore regions were relatively close to the coastline (Fig. 6b). Compared to pelagic regions, southwest and west winds in offshore regions could directly transport continental and anthropogenic aerosols to the ship's location from Guangdong and Hainan, China. Therefore, aerosol transport was higher in offshore regions. Fig. 7d indicated a difference in the distribution of marine aerosol composition between offshore and pelagic regions. In the offshore regions, the proportions of dust (DUST$_{10}$; $Dp \leq 10$ µm) and sulfate aerosols ($SO_4^{2-}$) were 5.04 % and 1.36 %, which were higher than those in the pelagic regions (1.45 % and 0.97 %, respectively). The higher concentrations of dust and sulfate aerosols may further indicate that continental aerosols have influenced the aerosol composition in the offshore regions (Geng et al., 2023; VanCuren, 2003). In the pelagic regions, the proportions of dimethyl sulfide (DMS), organic carbon (OC), and sulfur dioxide (SO$_2$) were 0.15 %, 1.2 %, and 1.7 %. These proportions were higher than those in the offshore regions (0.1 %, 0.84 %, and 0.56 %, respectively) due to the more frequent marine biological activities (e.g. phytoplankton metabolism) in the pelagic environments. For instance, phytoplankton releases DMS through cellular metabolism and cellular lysis (Saliba et al., 2020); DMS then undergoes atmospheric oxidation to form SO$_2$ (Kettle & Andreae, 2000). Additionally, phytoplankton produces OC (O'Dowd et al., 2004). These marine biological activities directly contribute to higher proportions of DMS, SO$_2$, and OC in pelagic regions. The proportion of sea salt aerosol (SEAS$_{10}$; $Dp \leq 10$ µm) in the pelagic regions (94.33 %) was higher than that in the offshore regions (91.9 %), indicating a significant contribution of SEAS$_{10}$ to the total marine aerosols in the pelagic environments. To sum up, our results indicated that the distance from the coast had a great influence on marine aerosol production and transport. It ultimately led to obvious differences between size distributions of marine aerosols in the offshore and pelagic regions.

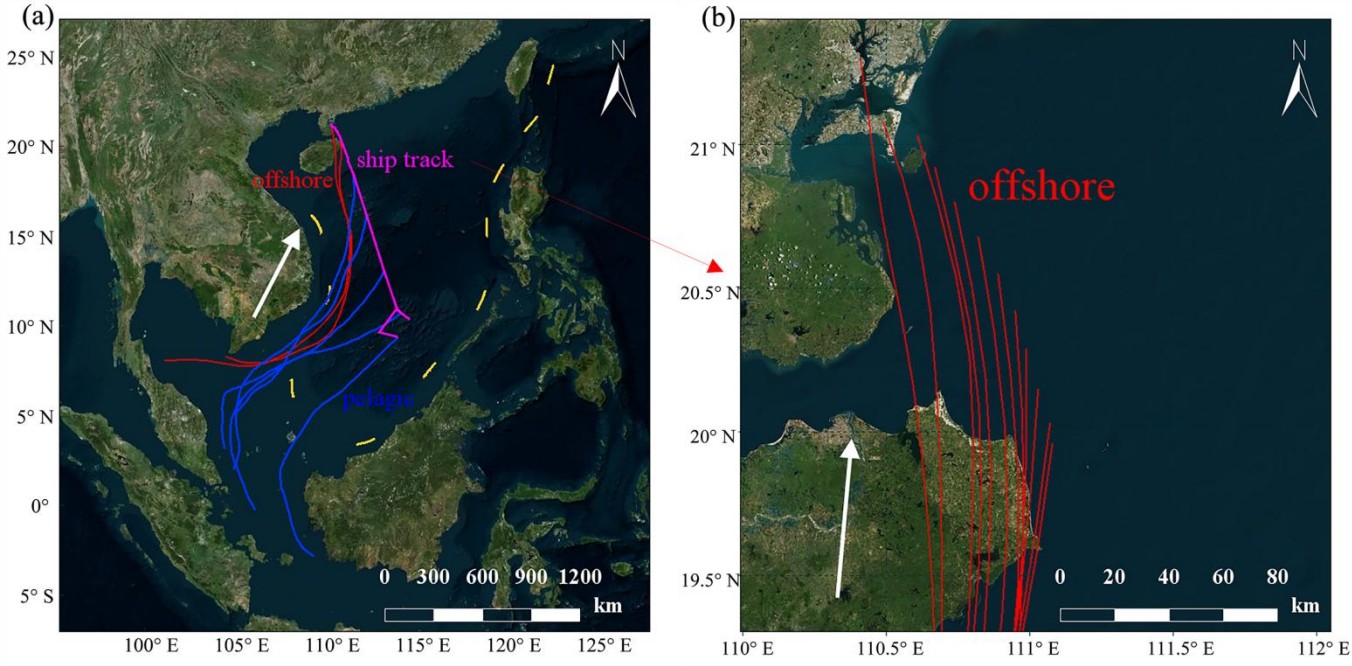

**Fig. 6 (a) The 72-h backward trajectory air mass source traces in the offshore (red solid lines) and pelagic (blue solid lines) regions. The light purple solid lines represented the ship track. (b) Detailed map of the backward trajectory air mass source traces passing through the mainland areas (© Google Earth). The white arrows represented the direction of air mass transport.**

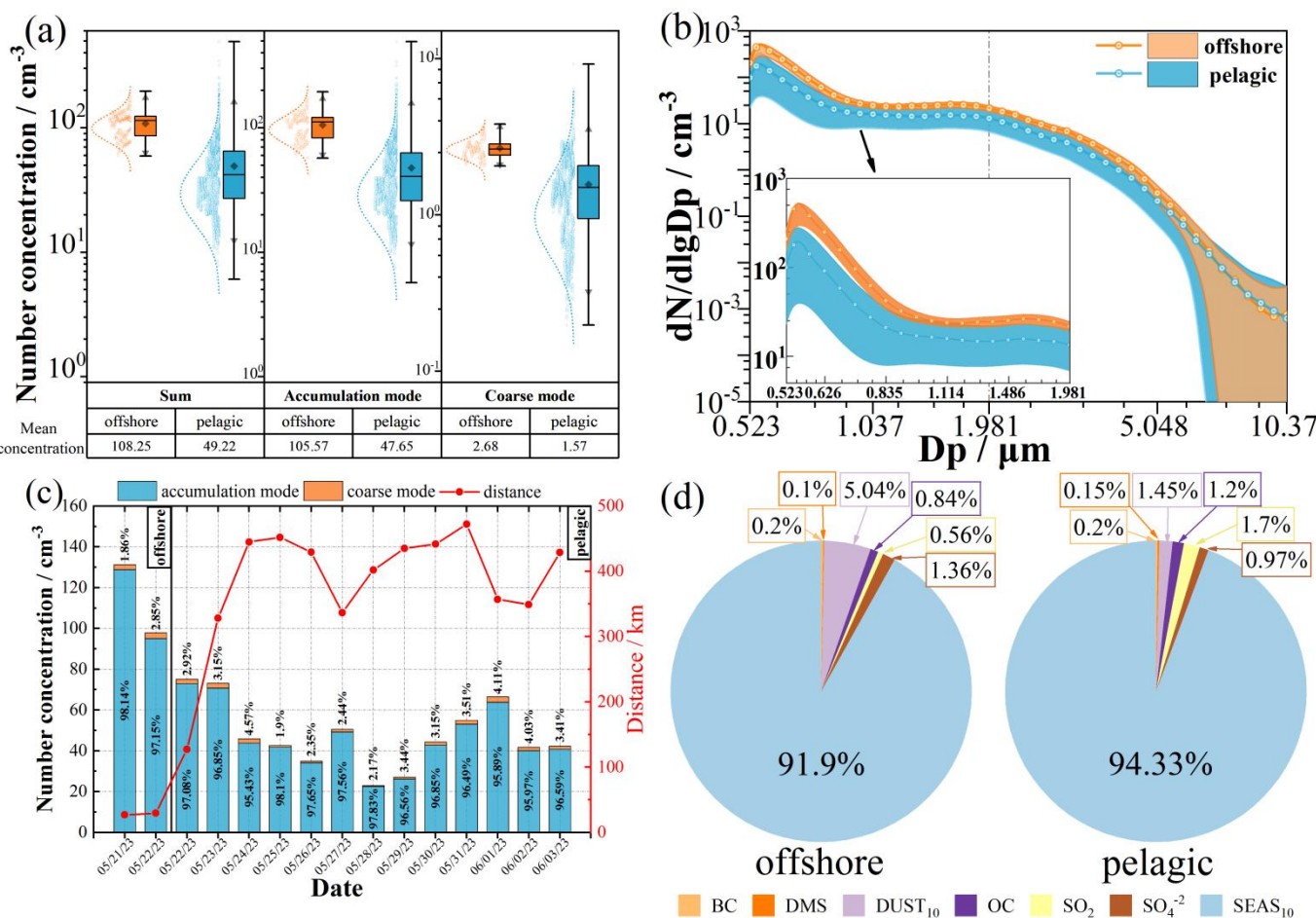

**Fig. 7 Classification of the shipboard observation path in the SCS: (a) Accumulation and coarse mode particle sizes graded NCs in the offshore and pelagic regions. For the box plots, the boxes represented the 25th to 75th percentile value, the black whisker represented the maximum and minimum range, the black triangle represented the 1.5 inter-quartile range, the black diamond marker represented the mean value, and the black horizontal line represented the median value. (b) The NCs of average size distributions (the solid lines and circles) and standard deviations (the shaded areas) for marine aerosols of 0.5 to 10 μm diameters in the offshore and pelagic regions. (c) The daily average variations of the proportions and the NCs of two aerosol particle modes were shown with distances from the coast. (d) The distributions of marine aerosol composition in the offshore and pelagic regions. The pie charts showed the average aerosol composition based on the mass concentrations from the Merra-2 aerosol dataset during the whole cruise period.**

**Table 3**

*Distributions of NCs for different aerosol particle modes in different ocean regions. Mean and SD, respectively, represent the mean values and standard deviations of the related meteorological parameters.*

| Observation Area | South China Sea | | | |
|---|---|---|---|---|
| Route Location | Offshore Region | | Pelagic Region | |
| Marine Aerosol Parameters | Mean | SD | Mean | SD |

| | | | | |
|---|---|---|---|---|
| Accumulation Mode (cm$^{-3}$) | 105.57 | 25.52 | 47.65 | 31.63 |
| Coarse Mode (cm$^{-3}$) | 2.68 | 0.38 | 1.57 | 0.80 |
| Sum (cm$^{-3}$) | 108.25 | 25.43 | 49.22 | 31.97 |
| Accumulation Mode / Sum (%) | 97.52 | - | 96.81 | - |
| Coarse Mode / Sum (%) | 2.48 | - | 3.19 | - |
| Meteorological Parameters | | | | |
| WS (m s$^{-1}$) | 10.74 | 1.95 | 8.64 | 3.70 |
| RH (%) | 91.20 | 1.72 | 82.41 | 3.40 |
| $T_{OBS}$ (°C) | 28.19 | 0.57 | 29.18 | 0.87 |
| SST (°C) | 27.71 | 0.37 | 29.78 | 0.33 |

### 3.3. Diurnal variation of NCs and their affecting factors in pelagic regions

The results of Section 3.2 showed that distributions and composition of marine aerosols in different regions were influenced by marine production and continental transport. Among them, the degree of impact on marine transport and background aerosols caused by continental transport and marine production differs greatly due to differences in the degree of continental transport and marine biological activities at different distances from the coast. Beyond that, many meteorological parameters, such as WS, T2m, SST, and SST-T$_{2m}$, might influence concentration and distribution of marine aerosols. It is expected that

there are diurnal differences. Therefore, we analyzed the diurnal variations of NCs and the correlations between NCs and different meteorological parameters in this section. In pelagic regions, the sources of 72-h backward trajectory air masses were from the ocean, and observational data were processed to exclude the continental influence. These aerosol data conformed to clean marine periods, which were proposed by Saliba et al. (2019) to extract relatively clean marine aerosol data; these NCs thus distinctly represented characteristics of local production of marine aerosol in these regions.

Fig. 8a showed diurnal variations of the mean values of summed NCs in the pelagic region. To better compare the diurnal variation, we divided the data into different periods according to sunrise and sunset times. Consequently, we selected the sunrise and sunset moments, along with the surrounding one-hour interval, as the transition periods to eliminate the effects of day-to-night transitions. Fig. 8a showed a clear diurnal variation trend. For accumulation mode, the NCs remained falling trend in the nighttime (00:00–03:00 and 21:00–23:00 LT), and they began steadily rising in the night-to-day transition (NDT)

period (04:00–06:00 LT). Then they showed a slight upward trend during the daytime (07:00–17:00 LT), but began to fall steadily during the day-to-night transition (DNT) period (18:00–20:00 LT). We also found that NCs exhibited apparent differences between the daytime and nighttime, and tended to increase or decrease significantly during the transition periods. Comparisons of size distributions (Fig. 8b) showed that number size distributions exhibit a relatively stable value in the 0.835–1.981 µm particle size range, and subtle differences emerged in this particle range. Quantitatively, peak diameter

varied slightly across periods: 0.571 µm in nighttime, 0.567 µm in the NDT period, 0.569 µm in daytime, and 0.570 µm in the DNT period. More notably, the peak value was 147.05 cm$^{-3}$ in nighttime, then rose to 155.87 cm$^{-3}$ in the NDT period,

further increased to 165.60 cm$^{-3}$ in daytime, and reached the highest value of 206.79 cm$^{-3}$ in the DNT period, registering a 0.4-fold increase relative to the nighttime baseline. The peak value showed a clear and continuous increasing trend, which may reveal variations in aerosol production. In addition, all size distributions for marine aerosols had the same shape. The consistent shape can be explained by their common marine origin and production mechanisms. The NCs of different aerosol particle modes in different periods were counted in Fig. 8c, and we can find that the average summed NCs were significantly different in different periods. Specifically, the summed NC was 44.03 cm$^{-3}$ in the nighttime, and 47.5 cm$^{-3}$ in the first transition period (i.e. NDT), but it was 49.28 cm$^{-3}$ in the daytime, and reached the highest (59.59 cm$^{-3}$) in the second transition period (i.e. DNT). The differences were all statistically significant ($p < 0.01$). The comparisons of the accumulation modes were consistent with those of the summed NCs. The NC of the accumulation mode was 42.43 cm$^{-3}$ in the nighttime, 45.90 cm$^{-3}$ in the NDT period, 47.62 cm$^{-3}$ in the daytime, and 57.91 cm$^{-3}$ in the DNT period. The differences were also statistically significant ($p < 0.01$). There were pronounced diurnal NC variations, with maximum differences of 35% observed between daytime, nighttime, and transition periods. However, there were no significant differences in the NCs of the coarse modes between different periods, as mentioned above, which were 1.60 cm$^{-3}$, 1.58 cm$^{-3}$, 1.66 cm$^{-3}$, and 1.68 cm$^{-3}$, respectively.

To explore the reasons for these differences, we further analyzed the correlation coefficients between the NCs and the meteorological parameters (Fig. 9a). The correlation coefficients were 0.41 for the coarse mode, 0.57 for the accumulation mode, and 0.58 for summed NCs. This correlation suggested that the NCs of all aerosol particle modes have positive correlations with the WSs. By contrast, negative correlations between SST and NC were found, which were -0.24, -0.45, and -0.47, respectively, and the NCs were low in the time periods with high SST. Compared to the WS and SST, our results showed that SST-T$_{2m}$ has a negative correlation with NCs, with correlation coefficients of -0.30, -0.82, and -0.83. Fig. 9b showed that WS was lower during nighttime, and SST and SST-T$_{2m}$ values were higher, resulting in NCs lower than those in daytime. In the NDT period, WS did not change significantly, but an obvious decrease in the SST and SST-T$_{2m}$ was found, ultimately resulting in a noticeable increase in NCs. In this period, SST and SST-T$_{2m}$ were the dominant factors. However, in the DNT period, NCs were highest due to the lowest values of SST and SST-T$_{2m}$ and the highest WS. In this period, the significant reduction in the WS led to a decrease in NCs. From the above analysis, the meteorological parameters have a joint impact on the production and distribution of marine aerosols.

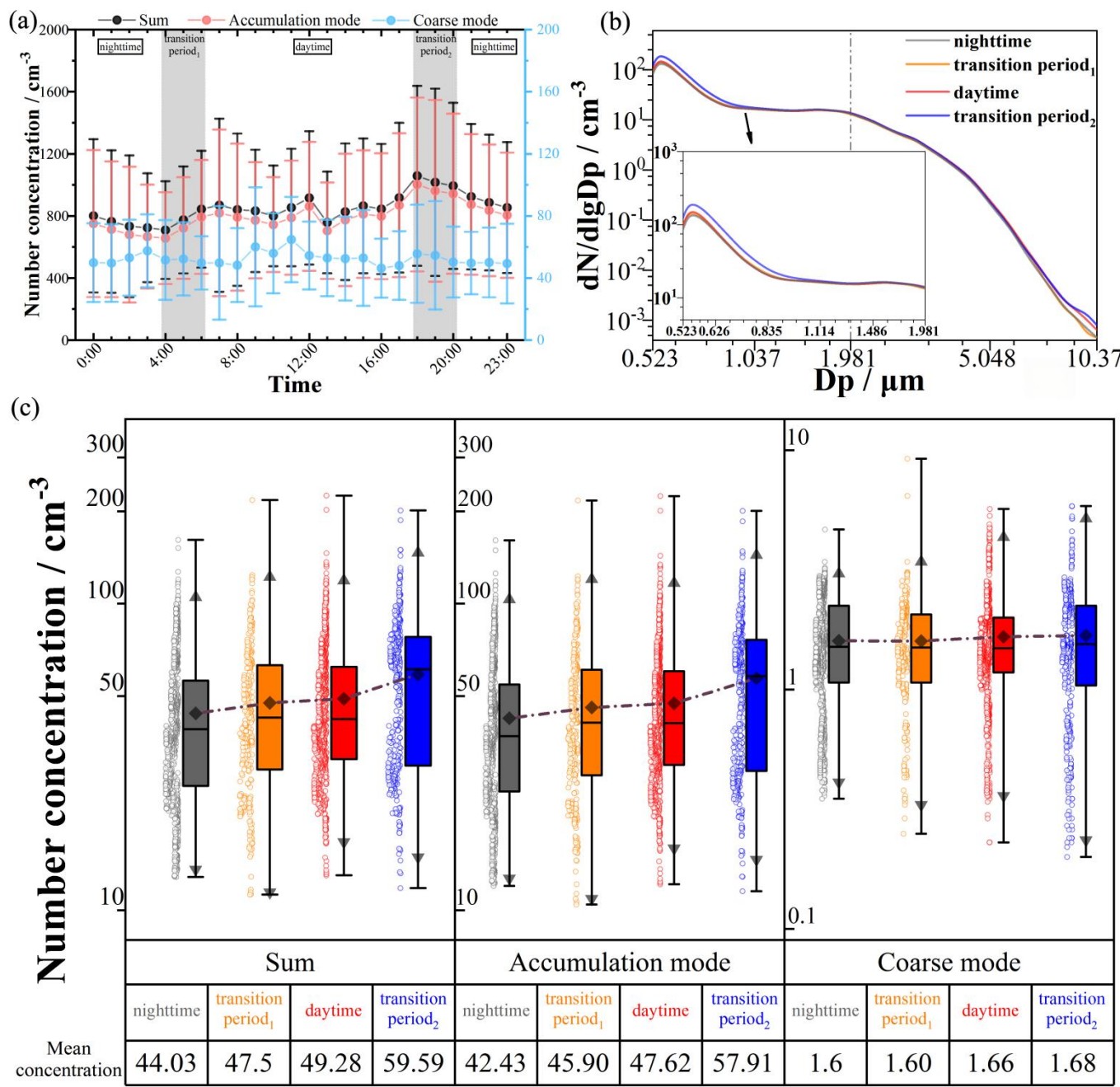

**Fig. 8 (a)** Diurnal variations of the total mean values of the NCs in the different aerosol particle modes. The vertical bars showed the standard errors (the shadow areas represented the transition periods between daytime and nighttime). **(b)** The NCs of average size distributions for marine aerosols of 0.5 to 10 µm diameters in different time periods. **(c)** The NCs of the different aerosol particle modes in different time periods. For the box plots, the boxes represented the 25th to 75th percentile value, the black whisker represented the maximum and minimum range, the black triangle represented the 1.5 inter-quartile range, the black diamond marker represented the mean value, and the black horizontal line represented the median value.

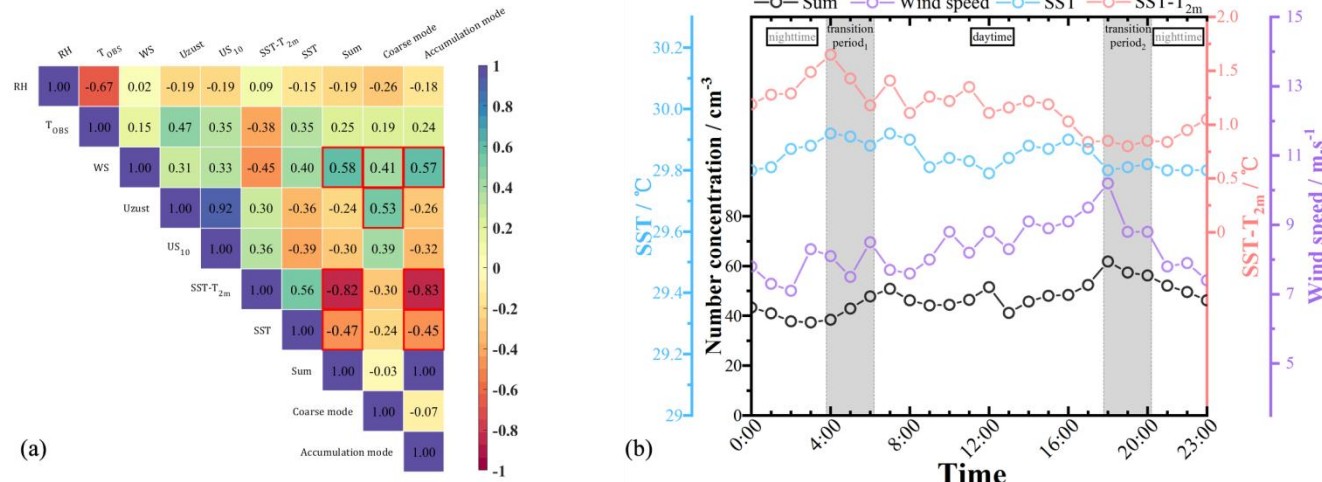

**Fig. 9 (a) The correlation coefficients between the NCs of all aerosol particle modes and different meteorological parameters. Correlation plot showing the Pearson correlation values of all marine aerosol NCs and meteorological parameters measured in pelagic regions. (b) The comparisons between the diurnal variations of summed NCs, SST, SST-T$_{2m}$, and WS.**

### 3.3.1. Influence of the WS on marine aerosol NCs

Our measurement results provided robust evidence for wind-driven marine aerosol production mechanism in the pelagic region. In all SST intervals, we observed a positive correlation between WS and NCs of all aerosol particle modes (Fig. 10), with *r* values greater than 0.8. In the pelagic region, the NCs were strongly influenced by the local production of marine aerosols, which had a relatively short lifetime compared with continental aerosols (Liu et al., 2005; Qureshi et al., 2009). Under the influence of sea surface wind, ocean wave fluctuations and sea surface friction may increase with intensified wind

stresses. Air bubbles generated and present on the sea surface might rupture to form numerous water droplets, which could eventually produce primary marine aerosol after evaporation and crystallization processes (Blanchard et al., 1980; Saliba et al., 2019). Therefore, increased WS may both intensify bubble rupture by enhancing sea surface friction and promote air-sea gas transfer (Jaeglé et al., 2011; Mårtensson et al., 2003). These processes might elevate the production of marine aerosols and natural marine precursors, ultimately raising the NCs in the pelagic region.

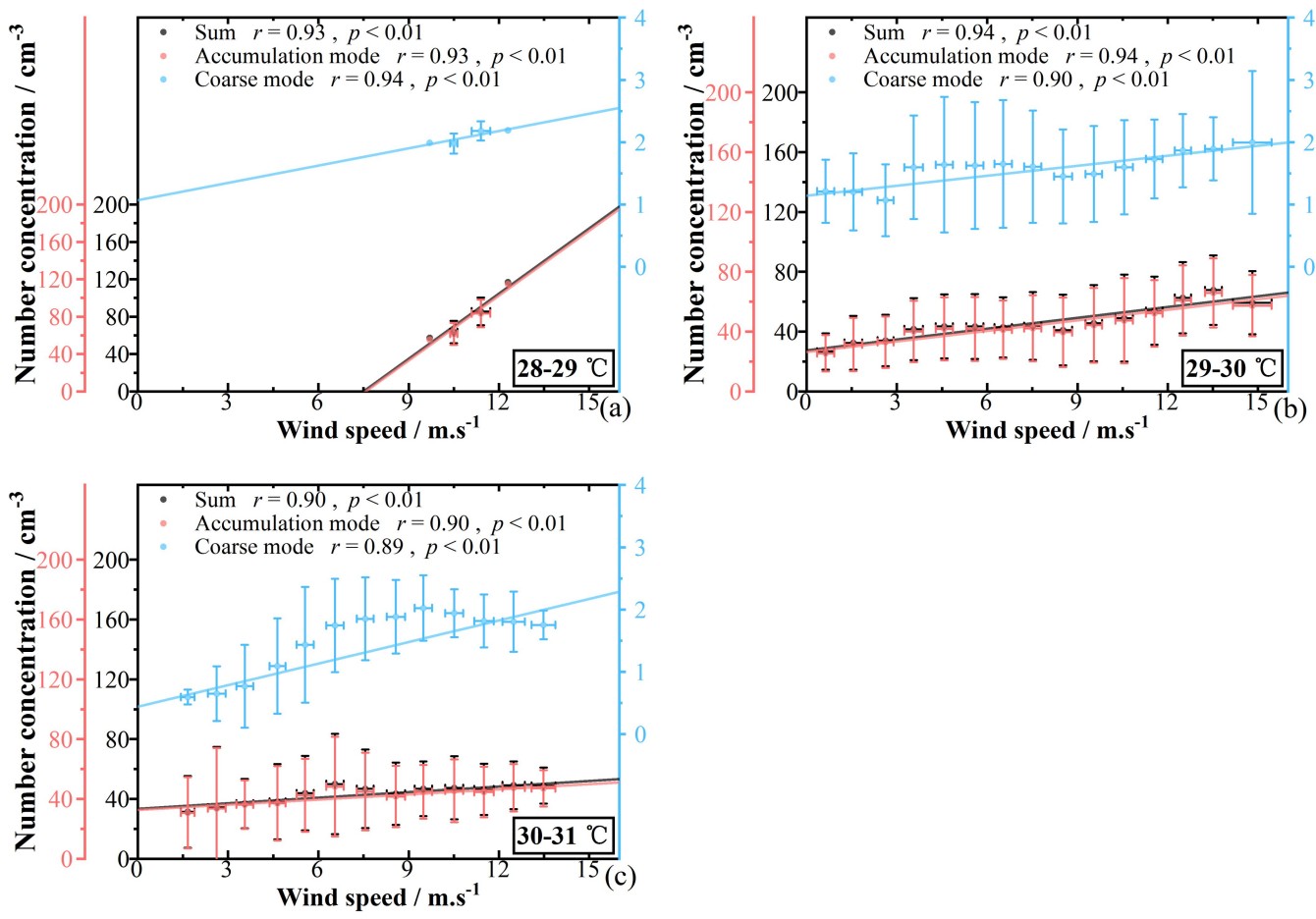

**Fig. 10 The NCs versus WS in the pelagic region. The NC of all aerosol particle modes versus WS for 28–29 °C (a), 29–30 °C (b), and 30–31 °C (c) SST intervals. The error bars represented the standard deviations. The *r* represented the Pearson correlation coefficients, and the *p* values were performed to test whether the correlations were significant.**

**3.3.2. Influence of the SST and SST-T$_{2m}$ on marine aerosol NCs**

Although WS can partly explain the variability of NCs, NCs exhibited a negative dependence on SST in all WS intervals. To analyze how SST influenced NCs in the pelagic region, Fig. S4 showed the NCs of all aerosol particle modes versus SST (28 °C ≤ SST ≤ 31 °C) in the WS intervals, in which WS was approximately constant. The design may largely exclude the influences of WS on NC. A negative correlation existed between SST and NCs for WS of 0–15 m s$^{-1}$ in the SCS, with all

475 their *r* values being more negative than -0.75 (Fig. S4). The steeper negative slope observed for the accumulation mode, compared to the coarse mode, pointed to a greater responsiveness of its NC to changes in SST, suggesting that accumulation mode particles might be more sensitive to these variations. This observed negative correlation between SST and NCs was

inconsistent with some laboratory studies (Keene et al., 2017; Forestieri et al., 2018) but consistent with other laboratory studies (Christiansen et al., 2019; Salter et al., 2014; Zábori et al., 2012b). These laboratory studies have shown disparate

results, which may stem from differences in experimental setups (e.g. plunging jet, water jet, or diffuser systems) or the water types used (e.g. natural, artificial, or synthetic seawater). A recent field study also reported decreasing NCs with rising SST based on shipborne measurements in the North Atlantic (Lehahn et al., 2014). In pelagic region of the SCS, combined evidence from prior studies (Christiansen et al., 2019; Lehahn et al., 2014; Salter et al., 2014; Zábori et al., 2012b) and our observational negative trends suggested that elevated SST may suppress near-surface air entrainment volumes, consequently

decreasing plunging jets. Near-surface air entrainment volumes and plunging jets were changed. Accordingly, the bubble rupture changed. Small daughter bubbles (secondary bubbles with smaller diameters, generated at the edges of central bubbles) were likely produced because larger central bubbles (the primary bubbles rising to the sea surface) ruptured; then, these smaller bubbles could produce submicron aerosol. These daughter bubbles are critical for the formation of submicron marine aerosols (Miguet et al., 2021; Sellegri et al., 2023). The generation of daughter bubbles decreases with an increasing

ratio of seawater density to viscosity and a decreasing ratio of seawater viscosity to surface tension. Therefore, under increasing SST, the ratio of seawater density to viscosity might increase and the ratio of seawater viscosity to surface tension might decrease, and the number of sea surface bubbles might decrease (Miguet et al., 2021; Sellegri et al., 2023). Therefore, these factors might ultimately result in decreased marine aerosol NCs, especially for the accumulation mode, with the increasing SST in the SCS.

Compared to the WS and SST, SST-$T_{2m}$ can better reflect the variations of the NCs ($r > 0.90$, Fig. 11). The correlations can explain that NCs had a significant negative correlation with the SST-$T_{2m}$ (-1 °C $\leq$ SST $\leq$ 4 °C). Figs. S5 and S6 illustrated the NC of all aerosol particle modes versus SST-$T_{2m}$ respectively for WS and SST intervals, and further presented this negative correlation under controlled WS and SST intervals. Prior studies (Lewis & Schwartz, 2004; Yuan et al., 2019) had suggested that SST-$T_{2m}$ may be related to atmospheric stability and play a role in air convection, mechanical mixing over the ocean,

and plume rise processes. Song et al. (2023) had also indicated that SST-$T_{2m}$ influence marine aerosol production by affecting atmospheric stability and thus the interfacial and effective production fluxes of marine aerosols by affecting the sea state, sea wave, and the process of the whitecap formation. Combining these previous inferences with our observational negative correlation between SST-$T_{2m}$ and NCs, it was plausible that SST-$T_{2m}$ could influence marine aerosol transport (e.g. potential upward transport driven by plume rise) and production. For example, increased SST-$T_{2m}$ may intensify plume rise,

leading to reduced NCs near the sea surface. Additionally, increased SST-$T_{2m}$ might indirectly decrease aerosol production by altering atmospheric stability. These phenomena might be an important factor affecting the differences in the marine aerosol NC distributions; they might also be important for the abovementioned different conclusions of the previous studies on the relationship between the SST and NCs. The differences in the SST-$T_{2m}$ might cause the inter-study differences despite the consistent SST during the experiment and should be considered further in subsequent targeted research. In summary, the

SST-$T_{2m}$ might influence the marine aerosol transport and production processes, resulting in differences in NCs. Hence, the

SST-T$_{2m}$ may be a new and significant parameter to better quantify the impact on the marine aerosol transport and production in the SCS.

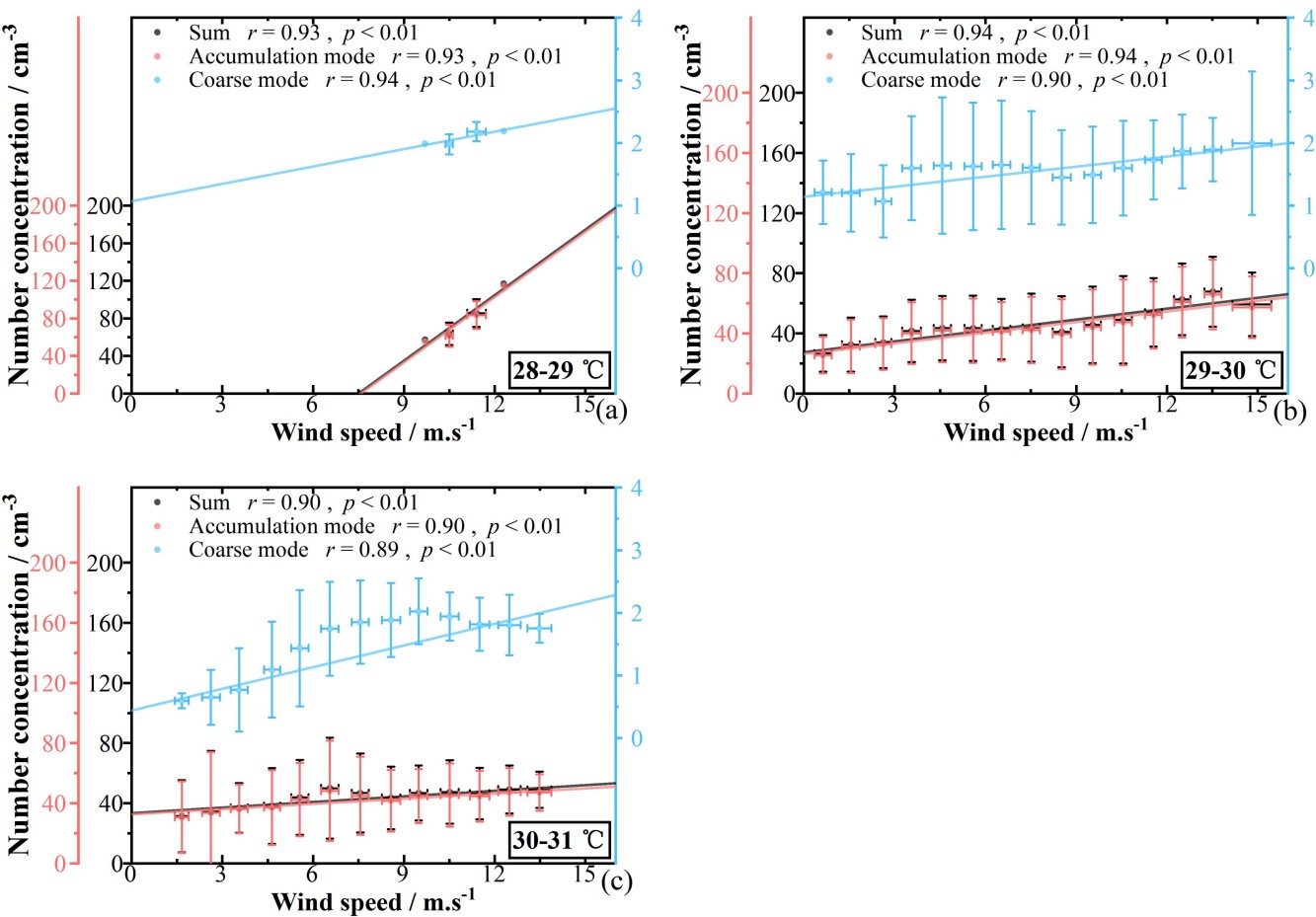

**Fig. 11 The relationship between the NC of all aerosol particle modes and WS (a), SST (b), and SST-T$_{2m}$ (c). The error bars represented the standard deviations. The *r* represented the Pearson correlation coefficients, and the *p* values were performed to test whether the correlations were significant.**

## 4. Conclusions

This study utilized cruise-based observational data collected from 21 May to 15 June 2023 to examine marine aerosol NCs and composition within the data-sparse SCS. Measurements revealed accumulation mode (0.5 μm ≤ $Dp$ ≤ 1.981 μm), coarse mode (1.981 μm < $Dp$ < 10.37 μm), and summed NCs of 52.35 ± 34.96 cm$^{-3}$, 1.66 ± 0.83 cm$^{-3}$, and 54.01 ± 35.37 cm$^{-3}$, respectively. Analysis of marine aerosol size distributions ($Dp$ < 10.37 μm) exhibited a bimodal structure, with modes at 0.542 μm and 1.981 μm. Spatial characterization of NCs and aerosol composition between offshore and pelagic regions revealed distinct differences, demonstrating that distance from the coast significantly influences distributions due to variations in aerosol transport and production. Furthermore, meteorological parameters, particularly SST-T$_{2m}$, were shown to

potentially induce changes in aerosol transport and production, ultimately leading to differences in NCs. Diurnal cycles in meteorological parameters also drove pronounced aerosol NCs variations, especially during daytime-nighttime transitions. Collectively, these findings proved to be key explanations of spatiotemporal marine aerosol variations in the SCS and their potential influencing factors.

The results obtained in the SCS demonstrated that distributions of marine aerosol NCs and composition depended on distance from the coast. In offshore regions, aerosol composition was strongly influenced by anthropogenic activities and continental transport, with both elevated NCs and higher proportions of continental aerosol composition ($DUST_{10}$) compared to pelagic regions. Furthermore, NCs exhibited a negative correlation with distance from the coast, and this trend was consistent with diminishing continental aerosol concentrations. Conversely, marine-derived composition ($SEAS_{10}$ and DMS) dominated in pelagic regions, reflecting intensified marine aerosol production.

The influences of meteorological parameters on marine aerosols differed in pelagic regions. Increasing WS likely drove ocean wave fluctuations and heightened sea surface friction, promoting aerosol particle production. Conversely, rising SST could reduce plunging jet intensity and entrapped air volume, thereby potentially altering bubble rupture processes and decreasing NCs. Notably, $SST-T_{2m}$ exhibited the strongest correlation with NCs. Higher $SST-T_{2m}$ likely reduced interfacial and effective aerosol production fluxes while intensifying vertical transport, collectively lowering NCs. WS, SST, and $SST-T_{2m}$ displayed distinct diurnal cycles, which may drive a distinct diurnal variation in NCs. Compared with the daytime, the combination of lower WS and higher SST and $SST-T_{2m}$ caused lower NCs at nighttime. During sunrise and sunset, rapid variations in meteorological parameters triggered NC fluctuations. In the NDT transition (the transition period$_1$), stable WS left SST and $SST-T_{2m}$ as dominant NC regulators. In the DNT transition (the transition period$_2$), all aforementioned three factors jointly influenced NCs.

Overall, this study filled data gaps and updated observational data for the SCS, while comprehensively analyzing diurnal variations in marine aerosols, impacts of continental transport, and potential influencing factors, especially $SST-T_{2m}$ among meteorological parameters. These findings enable subsequent refinement of traditional marine aerosol source functions, which rely solely on WS and SST. However, short-duration cruise data limited robust theoretical conclusions about SST and $SST-T_{2m}$ effects on aerosols. Additionally, validating sea surface phenomena (e.g. whitecap coverage, an indicator of wave-driven aerosol production) against meteorological parameters remains challenging due to scarce in site observations. Thus, more detailed observations and laboratory experiments will be critical in future studies to validate proposed influences of meteorological parameters and specific mechanisms underlying aerosol production and transport.

**Data Availability**

All data from this research are available from the corresponding author upon reasonable request.

## Author contributions

SC designed this study. ZQ and HX performed the measurements during the cruise. MZ, TL and YP implemented the back trajectory analysis. ZQ, XL, ZZ and SC analyzed the data. ZQ, SC and XW wrote the paper. All co-authors proofread and commented on the paper.

## Competing interests

The authors declare that they have no known competing financial interests or personal relationships that could have appeared to influence the work reported in this paper.

## Disclaimer

Publisher's note: Copernicus Publications remains neutral with regard to jurisdictional claims made in the text, published maps, institutional affiliations, or any other geographical representation in this paper. While Copernicus Publications makes every effort to include appropriate place names, the final responsibility lies with the authors.

## Acknowledgments

The authors acknowledge the South China Sea Institute of Oceanology, the Chinese Academy of Sciences for supporting this research cruise. The authors thank Kun Zhang and Haoda Yang for helping install the instruments. The authors thank the Max-Planck Society for the provision of the PLC model used in this study. The authors thank the NOAA ARL for the provision of the HYSPLIT transport and dispersion model used in this study. The ERA5 hourly dataset used in this study was provided by the ECMWF, and the MERRA-2 dataset was provided by the GMAO at NASA Goddard Space Flight Center.

## Financial support

This study was funded by the National 173 Basic Strengthening Program (E23D0Ha25T2), Key Scientific Research Project of Anhui Education Department (2022AH051712), Dreams Foundation of Jianghuai Advance Technology Center (2023-ZM01K015), and Scientific Research Foundation for the Advanced Talents, Chaohu University (KYQD-202208).

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

**Figure captions**

Fig. 1 The total view of (a) the Model 3321 APS spectrometer and (b) the automatic meteorological observation system.

Fig. 2 The time series of the observations on 25 May 2023. The black circle represented one case of ship pollution. (a) Trends of the aerosol size distributions. (b) Trends of the rainfall intensity and the WD.

Fig. 3 The time series of the shipboard observations in the SCS from 21 May to 3 June 2023. The blue-shaded regions represented periods affected by rain events. (a) Trend of the aerosol size distributions. (b) Trends of NCs of the two aerosol particle modes (black solid line represented the NC of the coarse mode, and red solid line represented the NC of the accumulation mode). (c) Trend of the WD. (d) Trends of the $T_{OBS}$ (dark orange solid line), $T_{2m}$ (light orange solid line), and SST (blue solid line). (e) Trends in the RH (gray solid line), the VIS (red solid line), and the rainfall intensity (dark blue solid line).

Fig. 4 (a) NC of the aerosol accumulation mode, (b) NC of the aerosol coarse mode, (c) RH, and (d) rainfall intensity as the functions of the WS and WD for the observations in the SCS.

Fig. 5 The scatter plots of (a) NCs of the aerosol accumulation mode and WS, (b) NCs of the aerosol coarse mode and WS. The observational data were binned to the WS intervals equal to 3 m s$^{-1}$; the boxes represented the 25th to 75th percentile value, the black whisker represented the 1.5 inter-quartile range, the black diamond marker represented the mean value, and the black horizontal line represented the median value in the box plots.

Fig. 6 (a) The 72-h backward trajectory air mass source traces in the offshore (red solid lines) and pelagic (blue solid lines) regions. The light purple solid lines represented the ship track. (b) Detailed map of the backward trajectory air mass source traces passing through the mainland areas (© Google Earth).

Fig. 7 Classification of the shipboard observation path in the SCS: (a) Accumulation and coarse mode particle sizes graded NCs in the offshore and pelagic regions. For the box plots, the boxes represented the 25th to 75th percentile value, the black whisker represented the maximum and minimum range, the black triangle represented the 1.5 inter-quartile range, the black diamond marker represented the mean value, and the black horizontal line represented the median value. (b) The NCs of average size distributions (the solid lines and circles) and standard deviations (the shaded areas) for marine aerosols of 0.5 to 10 µm diameters in the offshore and pelagic regions. (c) The daily average variations of the proportions and the NCs of two aerosol particle modes were shown with distances from the coast. (d) The distributions of marine aerosol composition in the offshore and pelagic regions. The pie charts showed the average aerosol composition based on the mass concentrations from the Merra-2 aerosol dataset during the whole cruise period.

Fig. 8 (a) Diurnal variations of the total mean values of the NCs in the different aerosol particle modes. The vertical bars showed the standard errors (the shadow areas represented the transition periods between daytime and nighttime). (b) The NCs of average size distributions for marine aerosols of 0.5 to 10 µm diameters in different time periods. (c) The NCs of the different aerosol particle modes in different time periods. For the box plots, the boxes represented the 25th to 75th percentile value, the black whisker represented the maximum and minimum range, the black triangle represented the 1.5 inter-quartile range, the black diamond marker represented the mean value, and the black horizontal line represented the median value.

Fig. 9 (a) The correlation coefficients between the NCs of all aerosol particle modes and different meteorological parameters. Correlation plot showing the Pearson correlation values of all marine aerosol NCs and meteorological parameters measured in pelagic regions. (b) The comparisons between the diurnal variations of summed NCs, SST, SST-$T_{2m}$, and WS.

Fig. 10 The NCs versus WS in the pelagic region. The NC of all aerosol particle modes versus WS for 28–29 °C (a), 29–30 °C (b), and 30–31 °C (c) SST intervals. The error bars represented the standard deviations. The $r$ represented the Pearson correlation coefficients, and the $p$ values were performed to test whether the correlations were significant.

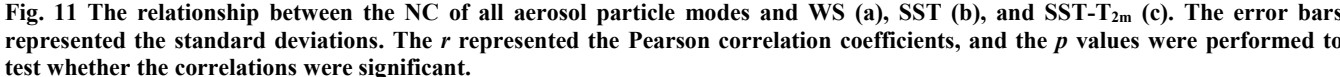

**Fig. 11 The relationship between the NC of all aerosol particle modes and WS (a), SST (b), and SST-T$_{2m}$ (c). The error bars represented the standard deviations. The *r* represented the Pearson correlation coefficients, and the *p* values were performed to test whether the correlations were significant.**