# Peer review of "Marine aerosol distributions from shipborne observations over the South China Sea: Diurnal variation characteristics and their controlling factors"

_EGUsphere, 2025_

## Referee Comment (RC1)

Comments:

1. Graphical abstract: make the y-axis on the right same limits for the size distributions and whats the unit? Is the right y-axis same as left y-axis?
2. The second panel of graphical abstract: are those correlations for pelagic or offchore? Not clear from the figure
3. Introduction: line 45: " Hoppel (1979, 1985) studied the aerosol NC and the particle size distribution on the east coast of the United States, and the significant changes in the particle size distribution can be associated with the changes in meteorological parameters and oceanic air mass." . When you say significant changes in PSD, my question is change from what? To what? Did he compare spatial distribution of aerosols? Or what you mean? Please clarify
4. Line 50: "In the Arctic, Leck (1996) reported that the submicrometer aerosol (Dp ≤ 1000 nm) mass concentrations during the International Arctic Ocean Expedition (IAOE-91) cruise; for instance, the average mass concentration was 0.76 µg m-3 over the ocean.:" rewrite the sentence. "reported that" doesn't make sense if you don't follow it with result.
5. Rewrite this too for clarity: ". In terms of the Antarctic, Savoie (1993) reported the submicrometer aerosol (Dp ≤ 1000 nm) concentrations, and the mean concentrations were 3.15 µg m-3 at Marsh."
6. All your references measure in ug/m3, and Sakerin (2015) 's value is in ng/m3. I suggest use same units so its easy for the reader to compare
7. ". For the China waters, Kim (2009) found that the average submicrometer aerosol particle (10 nm ≤ Dp ≤ 300 nm) concentrations were 4335 ± 2736 cm-3 over the East China Sea and 5972 ± 2736 cm-3 over the Yellow Sea." This reference feels out of place specially when the previous and following references mention units in ug/m3.
8. "All in all, there were some discrepancies in the marine aerosol concentrations and size distributions between the different ocean areas;" what discrepancy? That they aren't able to come to a common consensus for mass concentration? Or what you mean here?
9. "the major measurement data are relatively outdated and need to be updated." What you mean by outdated? Like previous measured values have changed or science behind them changed? What exactly and why outdated?
10. Line 77: are generation functions and source functions same? In that case use source function. May be cite Gong et al for source function involving wind speed?
11. Is SCS analogous to any other place on earth in terms of meteorology? I ask this because studying marine aerosols at a single location which isn't representative of similar areas would make little less sense but if it were to explain (or represent) marine aerosols for larger areas, it would make more sense. What I mean to ask is why is this location important to study?
12. Line 94. It is mentioned that diurnal variations are lacking. My question is why is it important? Add a hypothesis or reasoning why we want to learn about diurnal variations of marine aerosols and what impacts do diurnal variations have?
13. Line 119: ". Fig. 1 showed that the particle losses were small in the size range from 0.5 µm and 10 µm. Thereby, aerosol data within the size range of 0.5 to 10 µm were selected for future analysis in this study." You only measured this size range and calculated loss for this range, which is understandable. But saying that this range was selected for future analysis due to a small loss raises a question of whether you know the losses of other size ranges?
14. I don't think equation 1 is readable. Its all symbols and I think format needs to be changed
15. Line 204; avoid using words like drastically. If you use, give a quantitative measure of it'

16. Line 203: During the shipboard observation period, the average total marine aerosol NC was 54.01 ± 35.37 cm-3, the NC of aerosol accumulation mode was 52.35 ± 34.96 cm-3, and the NC of aerosol coarse mode was 1.66 ± 0.83 cm-3.; do the mean values include/exclude measurements during data events?

17. Line 205: unnecessary use of 'meanwhile'

18. Rephrase this sentence for clarity: Line 207 ". The shipboard observation data recorded and showed the overall average values and standard deviations of marine aerosol NCs under different temporal and geographical conditions, which were used to compare with the marine aerosol NCs observed."

19. Line 212: "This suggested that the NC of aerosol accumulation mode in the East China Sea might affected by the higher frequency of the new marine aerosol particle" can you cite some studies? Do newly formed particle have high growth rate that they can impact accumulation mode aerosols so much? Cite some studies which have shown the growth of small aerosols in the mentioned region. And studies which show npf is sparse in SCS.

20. Line 214: please rephrase this for clarity "Meanwhile, the total marine aerosol NC observed in this study contained the aerosol coarse mode (2 μm ≤ Dp ≤ 10 μm) and the part of aerosol accumulation mode (500 nm ≤ Dp ≤ 2000 nm), and the NC was slightly lower than the marine aerosol NC in the Atlantic by Flores et al. (2020)". Why sudden comparison with Atlantic? Are SCS and Atlantic supposed to have similar concentrations? Or why just compare with Atlantic?

21. What seasons did Cai et al. (2020) and Kong et al. (2016) carry their study in? easier to mention, it leaves the reader well informed. Did you compare the rain events between their study and your? is it possible they had less rain which kept the concentrations higher?

22. Line 218: "Although the differences in the observation seasons, the study region, and the particle size might influenced the average NC observations, it can still show that the marine aerosol was significantly affected by the continental transport and the anthropogenic activity in the offshore areas according to the latitude and longitude." Why did you mention differences in study region? The line before this says Cai and Kong did their study in the same region as yours. Next line says differences in study region. Its confusing.

23. Line 222: "However, some studies found that the aerosols might be generated on the porous surface when impinged by liquid droplets" porous surface of what? I know the next lines make it clear but it would be better if you introduce the concept when you first mention it.

24. Line 229: you mention only one size 4068nm. It's hard to say if ambient aerosols are just of one particular size. Please give a range.

25. Line 243: ". High NCs (≥ 150 cm-3) were observed almost entirely in which the WD were between NW and N that were caused by the high RH accompanied by the rainfall events, and the distributions of NCs were uniform when the wind was blowing in the other directions." So this high NC is the artificial aerosols created by porous surface of the ship and not natural aerosols? Because rainfall is supposed to cause wet removal or deposition of aerosols and thus decreased concentrations

26. Table 2: when you write accumulation mode, please mention the size, I don't think there's any use of comparing with n10-400 because your measurement starts from 500nm.

27. Line 274:284: include this is methods instead of results

28. Figure 7b shows same shape of size distribution, ofcourse, the concentrations are different. But doesn't same shape of distribution say same sources for the two regions? If one were affected by continental sources, wouldn't the distributions be slightly different?

29. Line 298: Can you elaborate this more? ". The marine aerosols decreased slowly with the increasing particle diameters below 1.114 μm due to the transport effect." You didn't discuss the 'transport effect' before, so its difficult for the reader to associate this.

30. In figure 7c, where is the correlation (r=-0.87) shown between Acc NC and distance? All is see is boxplot and distance points in red. Where's the correlation? The distance on 5/31 was higher than 5/27 but still number concentrations were higher on 5/31. How do you explain this?

31. Your accumulation mode starts from 500nm, how can you show that transport brings in particles which are atleast 500nm for contributing to increase in NC os accumulation mode? Can you cite someone who has shown the size of transport particles in this range? You also mentioned higher wind speed in offshore compared to pelagic areas. How can you say the higher NC at offshore was due to transport and not high WS?

32. In Figure 8, please add the arrows showing direction of air

33. Line 324: again words like significant doesn't make much sense. Talk in numbers. Out of all components dust shows the maximum change between pelagic and offshore, the difference in the rest of the components do not look 'siginificant'

34. Rephrase line 367: Fig 9a showed a clear diurnal variation emerged

35. During the daytime, WS increases and SST and difference in temperature between surface and 2m also decreases; so shouldn't the daytime NC be increasing because all the factors you mentioned for the transition period aligns for daytime, yet there's no clear trend in daytime. Why is that?

36. Line 433, add references of studies that showed entrained air decreased with increasing SST. Entrained where? sea surface or boundary layer? clarify

37. Instead of holding wither WS or SST constant for SST-T2m correlations, I suggest you perform multi linear regression or lasso regression or any suitable regression analysis to study the impact of these factors on NC variability. When you hold WS const for Fig 13, SST is still varying and when you hold SST const for Fig 14, WS is still varying.

38. Does the size distribution remain the same with increasing SST? What would you comment on the increase of diameter with warming SST as observed by Saliba (2019)?

---

## Referee Comment (RC3)

Review of "Diurnal variability and controlling mechanisms of marine aerosol distributions over the South China Sea: Insights from shipborne observations"
Qiao et al.

This study reports on aerosol properties measured using a TSI aerodynamic particle sizer (APS) and their relationship to environmental parameters measured on a 1-month cruise in the South China Sea and from reanalysis data. The paper aims to characterize the differences in aerosol number concentrations and size distributions between coastal and open-ocean regions and the implications transport has on the variability of modal concentrations. They argue that areas more proximate to the coast have higher aerosol number concentrations. While offshore, wind speed, SST, and the difference between SST and 2-m temperature (SST-T2m) influence aerosol concentration positively (wind speed through mechanical generation) and negatively (SST and SST-T2m through changes in buoyancy and bubble viscosity).

Many of the claims and arguments presented in this work are unsupported or they have been given with vague descriptions and unclear relationships to previous literature. Because this is an interesting dataset in an under-sampled region and some interesting results are obscured within this study, I feel that instead of a full rejection, this paper should undergo substantial major revision before being considered appropriate to be considered for publication. Below I have provided detailed list of major concerns and many technical comments, however many editorial issues persist throughout. I recommend the authors employ these corrections to improve this paper.

Main Comments
- The writing in many places is very hard to read due to grammatical errors and the use of frequent platitudes making the paper very difficult to follow. In some places I noted where these were and made suggestions, but the issues were far too numerous to point out each one. Many details are missing, and the discussion is vague without specific directed attention to the very detailed figures or putting the results in context with the broader literature context or studies within this region. The authors should make a concerted effort to carefully re-read the paper to ensure its clarity.

- In many cases, there is an incorrect or superfluous use of the article "the".
  - Examples:
    - Line 70: "[the] aerosol generation and transport…"
    - Line 71: "…found that [the] marine aerosol…"

- There was also the use of "meanwhile" and "on the other hand" when the authors were trying to further describe findings or procedures. In many, if not all, of the cases that either of these phrases were used they were unnecessary and confound clarity. I recommend the authors remove these phrases.

- The authors should use words like "the "correlations can explain", rather than declarative statements on correlations and other relationship because many of

their claims are based on mostly visual comparisons and not substantive correlations or quantitative analyses.

- It is the opinion of this reviewer that the word "production" or "emission" be used in place of "generation" when describing the marine sources of aerosol. In the way "generation" is used in this study, the authors are describing wind-generated aerosol production/emission, a more canonical use of the word. Either choice of terminology should be used consistently throughout the text.

- Citations in the text and references list should be checked for correct formatting. In many cases the authors have only listed the first author of papers rather than conforming to the proper citation style of this journal. This should be corrected.

- Abstract
  - The text in the graphical abstract is very difficult to read because the resolution is very poor.

- Introduction
  - The introduction lacks a clear narrative of the research problem and does not provide proper context that motivates the research questions. The current state of the introduction makes it hard for the reader to follow the motivation of this work. The authors should make sure to highlight what relevant prior has been that either supports or motivates why this work has been done. There is very little attention given to introducing the region, the region's sources of aerosol, and how this work will effectively fill in the gaps based on substantive research questions.

- Some comments on the methods of this work
  - Were the aerosols dried before being sampled? Are you sampling rain drops thus leading to some of the effects presented? Prior work has often shown (e.g. (Petters et al., 2006; Zheng et al., 2018)) precipitation acts as a large sink for accumulation and large accumulation-mode aerosol. I'm not sure I follow or buy the conclusions about wave droplet induced increases in accumulation-mode particles presented in this work based on the available observations.

  - Do the authors use aerosol composition from MERRA-2 in a similar size range of the APS? Many of the MERRA-2 aerosol species will have some non-negligible contributions from particles <0.5 μm that cannot be explained with changes in the APS alone.

  - Data screening: "Nucleation events" (Line 185 and in other passages) in the marine environment from new particle formation or otherwise are observable typically at sub-100nm sizes (in the nucleation, Aitken-mode range) and stop growing well below 500 nm. These are not measurable with an APS. How are the authors able to justify that such events can be

adequately observed with their measurement limitations? I am not confident that a number concentration criteria for screening ship pollution or continental influence would be meaningful from an APS alone due to it measuring mostly coarse particles. Plots and discussion of such a justification should be provided so that the reader is confident that there is fidelity in such a screening. Further, new particle formation is not discussed as a potential effect or limitation on the findings of this work and this should be included.

Technical/Editorial Comments
- Abstract, Line 17: I believe "increase" would be a more appropriate word here than "elevation"; "…a 120% [increase] in offshore aerosol number concentrations…"

- Abstract, Lines 26-27: This sentence is written vaguely and should be improved for clarification. Are the authors saying that the results of this work provide evidence to support differences in the spatial variability of marine aerosol in the SCS, and further, these results can help improve understanding of production and transport? The authors should rework this sentence to ensure these points are coming across clearly.

- Line 46: I would revise the latter part of this sentence as, "…and climate change, [there has been an increasing research focus on marine aerosols over the last] forty years."

- Line 47: "NC" is not defined before its first use here.

- Lines 47-49: The summary of findings in the Hoppel (1979, 1985) studies presented here are insufficient and vague. What do the authors mean by "associated with changes in meteorological parameters and oceanic air mass"? What meteorological parameters?

- Lines 49-57: This passage is not written clearly. The authors are attempting to provide a survey of mass concentration differences between different marine regions, but they are listed in a disjointed manner that makes it difficult for the reader to understand. I would recommend that this passage be shortened and combined in a way that illustrates the differences in reported aerosol mass in the different regions. Please also be cognizant of consistent unit usage, e.g. use ug m-3 as it is the most common of your reported masses.

- Lines 57: Revise "For the China waters, …" to "In marine regions off the coast of China,…"

- Lines 60-63: The authors should clarify what is meant by "discrepancies in the marine aerosol concentrations and size distributions." Do the authors mean "differences"? The ending of this sentence also seems like a bit of a tangent and

is not explained, ("especially from 10degN-20degN…" Why is this included and what is its relevance to the rest of the passage?

- Lines 63-65: I think a more substantive take away from this paragraph is that shipboard measurements provide better spatial (and temporal) context for aerosol measurements in diverse marine regions such as off the coast of China and they can further help improve characterizations by being updated. Can the authors please revise the sentence they have written in these lines to better convey that?

- Line 66: The sentence is written in a way that conveys a finding. If so, the authors need to provide support. I believe the authors are actually saying, "Aerosol generation and transport [can lead] to differences in marine aerosol concentrations and size distributions." Is that correct?

- Line 67: Clarify "aerosol components."

- Line 68: Are "weather events" synoptic weather patterns, mesoscale weather events, storms? Please clarify.

- Line 71: The authors have written "et al." after "relative humidity (RH)." Do they mean "etc."? "et al." or "etc." is not appropriate here. If there are additional pertinent meteorological parameters to include, please list them.

- Lines 71-75: The description of Tang RH effects on marine aerosol and the overall implication from the studies in the following sentence are not connected through the same logic. The authors are correct to note that RH changes can affect aerosol size through deliquescence and efflorescence, however, this is not related to the "wet deposition and dispersion" mentioned later. The RH effects on size come into play when the aerosol are being sampled, so if mentioning this point is to say that the previous work was inconsistent with their sampling procedures (drying, heating) and that makes comparison and aerosol characterization difficult, then that is what should be discussed here. Because the authors are instead discussing effects on generation and transport evidence of that effect should be discussed and not the Tang result.

- The reference for Tang et al. (1997) is not properly cited in the main text or reference list. In the reference list, the authors have only provided the name of the first author.

- Lines 75-78: The authors should specify what the relationships are between aerosol generation and wind speed from the studies cited. Is it the total aerosol concentration, are they size dependent?

- Line 79: "[They] explained that the SST …" Please clarify what or who is meant by "They."

- Lines 79-81: This is an inadequate summary of SST effects on aerosol properties. Please be specific based on the studies the authors have cited.

- The authors have made no mention of aerosol generation in the free troposphere (e.g. from new particle formation). This can have important implications for aerosol properties measured in the marine boundary layer and cannot be ignored in the discussion of important generation and transport drivers. Further, do they think, based on previous studies in this region, that new particle formation has any effect on the observed relationships?

- Line 66-88: Although I have provided rather detailed guidance on this entire paragraph, I believe the authors should completely rework this section. Very vague statements and disjointed thoughts persist throughout. The authors cite lots of work but don't use any of these citations for clear contextual support.

- Line 89: Please revise, "…most [observational] data…"

- Line 89-96: I believe the authors should remove this passage up to the start of the motivation sentence, "To address these, …" Everything prior is not needed and is discussed in the previous paragraphs.

- Line 102: Remove the semicolon after "respectively." This should begin a new sentence.

- APS data: do the authors use all of the channels in 0.5-10 μm diameter range? I believe previous work has shown that the first channel in the APS has issues with counting efficiency and sizing accuracy. Can the authors please provide clarification on if this channel was used and justification for why it is appropriate to use here?

- Line 119: delete "future."

- Reanalysis data: For the atmospheric dynamic/thermodynamic properties (temperature, wind speed, etc…) was the ERA5 reanalysis, MERRA-2, or a combination of ERA5 and MERRA-2 used? The authors discuss ERA5 in the first part of Section 2.3.1., but then say, "Meanwhile, the MERRA-2 was…" Why were two different datasets used for these variables? Why not use only MERRA-2 or ERA5? What motivated using two different datasets? I understand using MERRA-2 for the aerosol mass concentrations. Additionally, the authors should justify why they believe the coarse resolution of these datasets compared to the in situ cruise data are representative of the conditions measured where the ship is. Also, please provide a citation for MERRA-2 and spell out its acronym on the first use.

- Line 203: The authors need to define the size ranges used for their quantification of accumulation and coarse modes.

- Lines 202-203: I don't believe that it is appropriate for the authors to define aerosol number concentration integrated from the APS as the "total marine aerosol" or even the "total aerosol." A large portion of marine aerosol number concentrations come from substantial sub-500 nm particle contribution. As such, it would be much better suited if the authors revise this terminology; e.g. "APS integrated NC", "summed NC".

- Table 2: define the size range for "accumulation mode." Clarify in the caption that these are shipboard measurements or please specify the observational platform.

- Lines 206-207: The sentence "Due to the constraints…" is not necessary and should be removed.

- Lines 207-209: Delete "data recorded and" in the sentence "The shipboard observation data…"

- Lines 211-214: I don't understand how the authors came to these conclusions or what evidence is being used to support these claims of new particle formation being the cause of differences in accumulation-mode number concentrations. As I mentioned in one of my main comments, new particle formation and growth occurs at much smaller sizes than what is measured by the APS. The authors also do not provide any literature support for how they can argue this claim based on their measurements. Additionally, there is no discussion prior to or proceeding this sentence about westerlies and what that would mean for aerosol NC changes.

- Lines 213-216: I don't understand much of the discussion here or how it relates to the citation from the Atlantic. Did the Atlantic study measure the same size range? How can this study be used for comparison without mentioning these specific differences?

- Line 248: replace "region" with "range" or "bin."

- Figure 6: were these plots created using a fixed wind direction, relative humidity, precipitation, or other controlling factor? If not, how can the authors argue, especially based on the apparent low correlation and large scatter of the data, that wind generation is the primary mechanism for driving variability of this mode?

- Figure 7a: What do the different colors of the boxplots represent? If they are offshore and pelagic, you should use the same color scheme throughout the whole figure.

- Figure 7a: are the differences in pelagic and offshore number concentrations statistically significant for each mode and their sum?

- Lines 295-296: How does the bimodality of the distributions reported here compare to previous literature? Given the counting uncertainty in the lower bin of the APS,

I'm not sure I believe true bimodality is being observed here, nor do I believe it will be comparable to prior reports of bimodal marine size distributions such as in Hoppel et al (1986).

- Line 297: Please clarify what is meant by the aerosols were "evenly distributed in the 0.835 to 1.981 μm particle size range." Later in the text this term "evenly distributed" is mentioned again. It should be replaced with something more specific.

- Line 298: Where do the authors describe a "transport effect" on the size distribution below 1.114 μm? Please clarify.

- Line 301-305: Is this discussion only about the accumulation mode, coarse mode, or sum? Please specify.

- Lines 303-305: Were only 2 data points used for the offshore correlation? In Figure 7c, there are only 2 dates and two bars pertaining to offshore. 2 data points are not sufficient for a correlation. Did the authors use all data points for those days or just the average in the bar charts? Please clarify (1) what was used for the correlation and (2) what data is being shown in figure 7c; the caption says "diurnal variations" which is very vague.

- Line 310: "meteorological element distributions" is a very confusion description. This should be revised to "meteorological parameters" as in the table header.

- Line 310: The authors say the meteorological parameters are "significantly different" between offshore and pelagic areas. Based on the means and standard deviations this does not appear to be the case as the absolute differences are within only a few percent between the areas. The authors should please explain this claim and provide statistical evidence to support it.

- Lines 312-313: "In addition to the WS influence, the frequency…" Where is this shown?

- Lines 314-325: This passage and its discussion of effects on the aerosol is exceptionally inadequate. Significant jumps to conclusions are made throughout. (1) are the aerosols emitted from Guangzhou and Hainan and the "islands and countries surrounding SCS" expected to be observable in the size range of measurement of the APS? What evidence is there to support this? (2) "…underwent atmospheric transport, transformation, and deposition processes…" is very vague and not an appropriate claim based on the available measurements and analysis of this study. Please provide specific description of processes that the authors think the aerosol experienced that can explain the differences.

- Line 325-327: The authors need to justify that the dust and sulfate aerosol are representative of continental aerosol sources by providing citable studies.

- Lines 327-330: Have DMS, OC, and SO2 been shown to be in high concentrations in pelagic regions of the SCS? The "degree of [...] marine biological activity" is alluded to in the following section (Lines 355), but nothing related to this is discussed and how it might explain the differences between pelagic and offshore regions.

- Line 357: "...the meteorological parameters had obvious day-night differences." The word "obvious" should be removed here and replaced with "it is expected that there are diurnal differences." The differences are not "obvious" because they have not yet been shown.

- Line 359-363: The threshold of 120 cm-3 is not comparable to the Saliba et al. (2019). In that study the condensation nuclei concentration (particles >10 nm) was used, while this study is using mostly large accumulation and coarse mode aerosol. Please clarify the discrepancy and justify this choice of threshold. Were other thresholds tested and what support is available to make this choice?

- Lines 364-367: Please specify the hours used for each time.

- Lines 367-383: I see no "clear diurnal variation" in Figure 9. Figure 9a shows a very minimal increase in mean accumulation mode aerosol and no change in coarse mode. The plots in 9c show basically similar medians with interquartile ranges that are nearly identical for each mode and their respective time periods. Have the authors tested if these differences are statistically significant? Again, differences in the mean concentrations here seem to vary by only 1-5%. What are the differences observed in the size distributions of Figure 9b? These are not discussed clearly in the text and as a I reader I see no real changes. These should be quantified as a change in peak diameter, width, number, etc.

- Lines 428: the values are "more negative" not "smaller than" -0.75.

- Section 3.3.2. First Paragraph (SST influence): the authors should comment on the fact that the correlation found here for aerosol concentrations and SST occur for a very small range of SST of about 1-2 deg C. This is likely much smaller than the field and laboratory studies used for comparison. Do the authors think this has any effect on the observed correlations/slopes and the claims the authors make about entrainment and density changes that influence aerosol number concentrations? Terms like "daughter bubbles" are not described and make this discussion confusing. Please clarify this discussion for readability.

- Lines 432-433: Please clarify what is meant by "the influences of the SST on the NCs might be different in different seas due to the different components of the seawater."

- Line 433: "according to the results of the previous studies" What studies?

- Section 3.3.2. Second Paragraph (SST-T2m): The authors spend quite a lot of time making declarative statements about what's influencing the SST-T2m relationship to the aerosol concentration based on previous work. For such declarative statements, similar analysis exercises need to be carried out. They declare that SST-T2m was the "major determinant of atmospheric stability" which led to the "upward transport" of marine aerosol in the boundary layer. Other such declarative statements are made further in the paragraph, but no such results are shown. If the authors don't mean to declare such factors definitively describe their observations, they should be careful to instead place their findings in context with prior work rather than discuss with certainty.

- Line 496: Did the authors use an anomaly for SST-T2m or is it the difference between SST and T2m. Please clarify.

- Line 501-502: The authors mention "rapid solar radiation shifts" that drive changes in the aerosol concentrations. What is meant by this? Do they mean just day night differences? Please clarify as this is not discussed prior.

References

HOPPEL, W., FRICK, G., and LARSON, R.: EFFECT OF NONPRECIPITATING CLOUDS ON THE AEROSOL SIZE DISTRIBUTION IN THE MARINE BOUNDARY-LAYER, Geophysical Research Letters, 13, 125-128, 10.1029/GL013i002p00125, 1986.

Petters, M., Snider, J., Stevens, B., Vali, G., Faloona, I., and Russell, L.: Accumulation mode aerosol, pockets of open cells, and particle nucleation in the remote subtropical Pacific marine boundary layer, Journal of Geophysical Research-Atmospheres, 111, 10.1029/2004JD005694, 2006.

Zheng, G., Wang, Y., Aiken, A., Gallo, F., Jensen, M., Kollias, P., Kuang, C., Luke, E., Springston, S., Uin, J., Wood, R., and Wang, J.: Marine boundary layer aerosol in the eastern North Atlantic: seasonal variations and key controlling processes, Atmospheric Chemistry and Physics, 18, 17615-17635, 10.5194/acp-18-17615-2018, 2018.

---

## Author Response (AR1)

**Summary**

We thank the three reviewers for the patient and meticulous review of the manuscript. Following those comments, the manuscript has been carefully revised. We have addressed each of the reviewers' comments. All the modifications and corrections are marked in red in the manuscript text.

Below are our point-to-point responses to the reviewers' comments and suggestions, with the reviewers' comments (RC) in black, our response in red, and *the revised manuscript content in italicized blue font*.

**Answer to Reviewer 1**

RC1.1: Graphical abstract: make the y-axis on the right same limits for the size distributions and whats the unit? Is the right y-axis same as left y-axis?

Response: We confirm that the right y-axis has the same unit as the left y-axis, both representing aerosol NC (cm⁻³). For the initial manuscript version, we make the y-axis on the right same limits for the size distributions. However, using these same limits failed to clearly demonstrate the characteristics of coarse mode particles; therefore, we adopt the present limits based on the editor's suggestion.

RC1.2: The second panel of graphical abstract: are those correlations for pelagic or offshore? Not clear from the figure

Response: These correlations are specific to the pelagic regions. We have explicitly added "pelagic region" in the second panel of graphical abstract to eliminate ambiguity.

RC1.3: Introduction: line 45: " Hoppel (1979, 1985) studied the aerosol NC and the particle size distribution on the east coast of the United States, and the significant changes in the particle size distribution can be associated with the changes in meteorological parameters and oceanic air mass." . When you say significant changes in PSD, my question is change from what? To what? Did he compare spatial distribution of aerosols? Or what you mean? Please clarify

Response: We appreciate the reviewer's request for clarification regarding Hoppel's work (Hoppel, 1979; 1985). In these studies, Hoppel observed increased aerosol NC and the number size distribution with rising wind speeds off the U.S. East Coast. The term "significant changes in particle size distribution (PSD)" specifically refers to shifts toward higher concentrations across the size spectrum under high-wind conditions, compared to low-wind periods.

Meanwhile, we consider that this citation is inconsistent with the theme of this section - "Researchers' reports on aerosol mass concentration and number concentration". Therefore, we have removed this sentence and will cite it in the next section that discusses the influence of meteorological factors on aerosols.

RC1.4: Line 50: "In the Arctic, Leck (1996) reported that the submicrometer aerosol (Dp ≤ 1000 nm) mass concentrations during the International Arctic Ocean Expedition (IAOE-91) cruise; for instance, the average mass concentration was 0.76 μg m-3 over the ocean.:" rewrite the sentence. "reported that" doesn't make sense if you don't follow it with result.

Response: We have revised this sentence. Meanwhile, as requested by Reviewer 3, we have streamlined and integrated these paragraphs. The final version is as follows:

*Early observations by Prospero (1979) across multiple marine areas showed notable variations in marine aerosol concentrations,,ranging from 3.34 to 8.71 μg m$^{-3}$. Subsequent reported measured data verify substantial regional marine aerosol concentration differences between different ocean areas. In polar regions, submicrometer aerosol (Dp ≤ 1000 nm) mass concentrations averaged 0.76 μg m$^{-3}$ in the Arctic (Leck & Persson, 1996) versus 3.15 μg m$^{-3}$ in the Antarctic (Savoie et al., 1993). In the Pacific Ocean, the PM$_{2.5}$ (Dp ≤ 2500 nm) concentration averaged 12.3 ± 9.1 μg m$^{-3}$ in the Western Pacific (Ma et al., 2022) versus 140 ± 48.1 μg m$^{-3}$ in the Bohai Sea (Han et al., 2019). In the Indian Ocean, Pant et al. (2009) observed that the average micrometer aerosols (500 nm ≤ Dp ≤ 10000 nm) mass concentrations were 8.89 μg m$^{-3}$.*

RC1.5: Rewrite this too for clarity: ". In terms of the Antarctic, Savoie (1993) reported the submicrometer aerosol (Dp ≤ 1000 nm) concentrations, and the mean concentrations were 3.15 μg m-3 at Marsh."

Response: Please see response to RC1.4.

RC1.6: All your references measure in ug/m3, and Sakerin (2015) 's value is in ng/m3. I suggest use same units so its easy for the reader to compare

Response: Please see response to RC1.4.

RC1.7: ". For the China waters, Kim (2009) found that the average submicrometer aerosol particle (10 nm ≤ Dp ≤ 300 nm) concentrations were 4335 ± 2736 cm-3 over the East China Sea and 5972 ± 2736 cm-3 over the Yellow Sea." This reference feels out of place specially when the previous and following references mention units in ug/m3.

Response: We appreciate the reviewer's remark on the unit inconsistency when citing Kim (2009). To enhance coherence and clarity, we have added a transition sentence and repositioned the original sentence to prevent potential confusion among readers.

*In the Indian Ocean, Pant et al. (2009) observed that the average micrometer aerosols (500 nm ≤ Dp ≤ 10000 nm) mass concentrations were 8.89 μg m$^{-3}$. In addition to aerosol mass concentrations, researchers have also observed aerosol number concentrations (NCs) differences. For instance, in marine regions off the coast of China, Kim et al. (2009) found that the average submicrometer aerosol particle (10 nm ≤ Dp ≤ 300 nm) concentrations were 4335 ± 2736 cm$^{-3}$ over the East China Sea and 5972 ± 2736 cm$^{-3}$ over the Yellow Sea.*

RC1.8: "All in all, there were some discrepancies in the marine aerosol concentrations and size distributions between the different ocean areas;" what discrepancy? That they aren't able to come to a common consensus for mass concentration? Or what you mean here?

Response: We sincerely appreciate this insightful comment. The term "discrepancies" specifically refers to discrepancies in aerosol concentrations (both number concentrations and mass concentrations) and size distributions (e.g., particle size distributions) across ocean areas. This is demonstrated by variations between two open oceans (the Arctic and the Southern Oceans), as well as variations between inland seas (e.g., the Yellow Sea) and continental marginal seas (e.g., the East China Sea). These differences are caused by the differences in aerosol production and transport.

These regional differences also highlight the urgent need for targeted studies in specific marine areas - particularly in data-sparse regions such as the South China Sea, which further underscores the value of our present study in this region.

RC1.9: "the major measurement data are relatively outdated and need to be updated." What you mean by outdated? Like previous measured values have changed or science behind them changed? What exactly and why outdated?

Response: We sincerely appreciate this valuable comment. The "outdated" does not refer to changes in the scientific principles behind aerosol measurements, but to the previous measured values.

First, over the past two decades, remarkable changes in aerosol concentrations and compositions have been observed worldwide (Fioletov et al., 2023; Gupta et al., 2022). Zhang et al. (2025) reported pronounced discrepancies in annual mean aerosol concentrations across regions such as Europe, North America, and Southeast Asia using AERONET station data. Critically, over the past three decades, Aerosol Optical Depth has increased substantially in Southeast Asia and Chinese coastal regions (Zhao et al., 2016). However, most of the publicly available in-situ measurement results of marine aerosols in the South China Sea can only be traced back to before 2020 (Kong et al., 2016; Su et al., 2022), and there is a lack of newly updated observational data about recent five years. This long time span makes the existing data unable to reflect the current status of aerosol properties in the region.

Second, rapid urbanization, industrialization, and population growth in Asia have significantly increased anthropogenic aerosol emissions in recent years; these aerosols are transported to the South China Sea, altering the region's aerosol concentration and composition. Global warming has caused variations in wind speed (Zheng et al., 2016) and sea surface temperature (Forestieri et al., 2018; Paulot et al., 2020): wind speed (a key driver of marine aerosol production) and sea surface temperature (which affects sea surface tension, sea surface density, and bubble breaking processes) both influence the marine aerosol generation efficiency and aerosol particle size. Additionally, climate change has led to geographical and annual variations in marine

phytoplankton and biological content (Asch et al., 2019), further causing differences in primary and secondary organic aerosols (Chevassus et al., 2025).

In summary, updating in-situ aerosol measurement data for the South China Sea is necessary to obtain accurate and timely insights into the region's aerosol properties.

Reference:

Asch, R. G., Stock, C. A., and Sarmiento, J. L.: Climate change impacts on mismatches between phytoplankton blooms and fish spawning phenology, Glob. Change Biol., 25, 2544–2559, https://doi.org/10.1111/gcb.14650, 2019.

Chevassus, E., Fossum, K. N., Ceburnis, D., Lei, L., Lin, C., Xu, W., O'Dowd, C., and Ovadnevaite, J.: Marine organic aerosol at Mace Head: effects from phytoplankton and source region variability, Atmos. Chem. Phys., 25, 4107–4129, https://doi.org/10.5194/acp-25-4107-2025, 2025.

Fioletov, V. E., McLinden, C. A., Griffin, D., Abboud, I., Krotkov, N., Leonard, P. J. T., Li, C., Joiner, J., Theys, N., and Carn, S.: Version 2 of the global catalogue of large anthropogenic and volcanic SO2 sources and emissions derived from satellite measurements, Earth Syst. Sci. Data, 15, 75-93, https://doi.org/10.5194/essd-15-75-2023, 2023.

Forestieri, S. D., Moore, K. A., Martinez Borrero, R., Wang, A., Stokes, M. D., and Cappa, C. D.: Temperature and Composition Dependence of Sea Spray Aerosol Production, Geophys. Res. Lett., 45, 7218–7225, https://doi.org/10.1029/2018gl078193, 2018.

Gupta, G., Venkat Ratnam, M., Madhavan, B., and Narayanamurthy, C.: Long-term trends in Aerosol Optical Depth obtained across the globe using multi-satellite measurements, Atmos. Environ., 273, 118953, https://doi.org/10.1016/j.atmosenv.2022.118953, 2022.

Kong, Y. W., Sheng, L. F., Liu, Q., and Li, X. Z.: Impact of marine atmospheric process on aerosol number size distribution in the South China Sea, (in Chinese), Environ. Sci., 37, 2443-2452, 10.13227/j.hjkx.2016.07.005, 2016.

Paulot, F., Paynter, D., Winton, M., Ginoux, P., Zhao, M., and Horowitz, L. W.: Revisiting the impact of sea salt on climate sensitivity, Geophys. Res. Lett., 47, e2019GL085601. https://doi.org/10.1029/2019gl085601, 2020.

Su, Y., Han, Y., Luo, H., Zhang, Y., Shao, S., and Xie, X.: Physical-optical properties of marine aerosols over the South China Sea: shipboard measurements and MERRA-2 reanalysis, Remote Sens., 14, 2453, https://doi.org/10.3390/rs14102453, 2022.

Zhang, Z., Li, J., Che, H., Dong, Y., Dubovik, O., Eck, T., Gupta, P., Holben, B., Kim, J., Lind, E., Saud, T., Tripathi, S. N., and Ying, T.: Long-term trends in aerosol properties derived from AERONET measurements, Atmos. Chem. Phys., 25, 4617–4637, https://doi.org/10.5194/acp-25-4617-2025, 2025.

Zhao, X., Heidinger, A. K., and Walther, A.: Climatology Analysis of Aerosol Effect on Marine Water Cloud from Long-Term Satellite Climate Data Records, Remote Sens., 8, 300, https://doi.org/10.3390/rs8040300, 2016.

Zheng, C. W., Pan, J., and Li, C. Y.: Global oceanic wind speed trends, Ocean Coast. Manag., 129, 15-24, https://doi.org/10.1016/j.ocecoaman.2016.05.001, 2016.

RC1.10: Line 77: are generation functions and source functions same? In that case use source function. May be cite Gong et al for source function involving wind speed?

Response: The generation functions are same with source functions. We have changed generation functions to source functions. We sincerely appreciate your recommendation of Gong's work. This paper has now cited it in the revised version.

RC1.11: Is SCS analogous to any other place on earth in terms of meteorology? I ask this because studying marine aerosols at a single location which isn't representative of similar areas would make little less sense but if it were to explain (or represent) marine aerosols for larger areas, it would make more sense. What I mean to ask is why is this location important to study?

Response: Thank you for raising this important point. The South China Sea is analogous to some continental marginal seas in terms of geography and meteorology, such as the Coral Sea, the Arabian Sea, and the Gulf of Mexico.

As a continental marginal sea, the South China Sea is located on the continental margin and separated from the open ocean by peninsulas, islands, or island arcs. The aerosol measurement results can clearly reflect the influence of continental air masses in the offshore regions. The South China Sea has large latitudinal and longitudinal range and area (01°12′N-23°24′N, 99°00′E-122°08′E, covering 3.5 million square kilometers). In the pelagic regions, the aerosol measurement results can clearly reflect the distribution characteristics of background aerosols.

Compared with the open ocean, continental marginal seas are affected by topography and land, resulting in clearer changes in wind speed, sea surface temperature, and air temperature difference. This makes it easier for us to identify the relationships between these meteorological factors and aerosol production.

In summary, we can analyze the influences of continental air masses on marine aerosols and the effects of meteorological factors on aerosol production. Most importantly, in-situ aerosol measurement data in the South China Sea region (especially between 10°N and 20°N) are very scarce, and the major measurement data are relatively outdated. Therefore, we have selected this region for our study.

RC1.12: Line 94. It is mentioned that diurnal variations are lacking. My question is why is it important? Add a hypothesis or reasoning why we want to learn about diurnal variations of marine aerosols and what impacts do diurnal variations have?

Response: Thank you for raising this important point. Aerosols are crucial for radiation transfer, cloud microphysical processes, and the climate system. If the diurnal variation of aerosols follows a certain pattern, it can more clearly and directly indicate that aerosol generation and transport are influenced by specific parameters. The discovery of the diurnal variation pattern opens many new questions for future research to elucidate the mechanism underlying this phenomenon and the direct impact of diel cycle of marine aerosol on the radiation balance. Furthermore, on a larger scale, it also involves links to cloud microphysical processes, which in turn

relate to energy fluxes and the climate. Ultimately, these insights contribute to the improvement of atmospheric models.

We have revised this sentence to highlight the importance of aerosol diurnal variation.

*The diurnal scale of marine aerosol variation can provide valuable information about their production and transport (Flores et al., 2021), and how these processes are influenced by meteorological parameters. Understanding the diurnal variation is also crucial for improving atmospheric models. Studies on the scale of diurnal variation in marine aerosol remain scarce, and there is an urgent need to clarify the specific connection between these diurnal variation and meteorological parameters to better understand aerosol production and transport.*

RC1.13: Line 119: ". Fig. 1 showed that the particle losses were small in the size range from 0.5 μm and 10 μm. Thereby, aerosol data within the size range of 0.5 to 10 μm were selected for future analysis in this study." You only measured this size range and calculated loss for this range, which is understandable. But saying that this range was selected for future analysis due to a small loss raises a question of whether you know the losses of other size ranges?

Response: We appreciate your constructive suggestions. We have redrawn the particle losses for the Model 3321 APS spectrometer across the full size range (0.5-20 μm) in Fig. 1. The figure revealed a dramatic increase in the particle loss at particle diameters exceeding 10 μm, accordingly we excluded data for particles >10 μm in subsequent analyses. Concurrently, we have revised this sentence to eliminate potential ambiguities.

*We used the Particle Loss Calculator (PLC) to calculate the particle losses for the Model 3321 APS spectrometer in this cruise (Fig. S1) (Von Der Weiden et al., 2009). Fig. S1 revealed a dramatic increase in aerosol particle loss at particle diameters exceeding 10 μm. Meanwhile, the accuracy of the aerosol data for particle diameters between 0.5 μm and 10 μm, as measured by the Model 3321 APS spectrometer, had been fully validated in previous studies (Pagels et al., 2005; Peters et al., 2003; Peters, 2006). Thereby, aerosol data within the size range of 0.5 to 10 μm were selected for analysis in this study.*

[Figure]

*Fig. S1 The calculated particle losses for the Model 3321 APS spectrometer in this cruise.*

RC1.14: I don't think equation 1 is readable. Its all symbols and I think format needs to be changed
Response: Thanks very much for your suggestion. We have revised this equation.

RC1.15: Line 204; avoid using words like drastically. If you use, give a quantitative measure of it'
Response: Thanks very much for your suggestion. We have deleted "drastically".

RC1.16: Line 203: During the shipboard observation period, the average total marine aerosol NC was $54.01 \pm 35.37$ cm-3, the NC of aerosol accumulation mode was $52.35 \pm 34.96$ cm-3, and the NC of aerosol coarse mode was $1.66 \pm 0.83$ cm-3.; do the mean values include/exclude measurements during data events?
Response: The mean values excluded measurements during data events.

RC1.17: Line 205: unnecessary use of 'meanwhile'
Response: Thanks very much for your suggestion. We have deleted "meanwhile".

RC1.18: Rephrase this sentence for clarity: Line 207 ". The shipboard observation data recorded and showed the overall average values and standard deviations of marine aerosol NCs under different temporal and geographical conditions, which were used to compare with the marine aerosol NCs observed."
Response: Thanks very much for your suggestion. We have revised this sentence.
*The shipboard observational data showed overall average values and standard deviations of maritime aerosol NCs under different temporal and geographical conditions. We used these data to compare with the marine aerosol NCs during this cruise period.*

RC1.19: Line 212: "This suggested that the NC of aerosol accumulation mode in the East China Sea might affected by the higher frequency of the new marine aerosol particle" can you cite some studies? Do newly formed particle have high growth rate that they can impact accumulation mode aerosols so much? Cite some studies which have shown the growth of small aerosols in the mentioned region. And studies which show npf is sparse in SCS.
Response: Thank you for your constructive comments. As Reviewer 3 pointed out, the size range (nanometer scale) where new particle formation and growth processes occur is far smaller than the measurement range of the APS (0.5–30 μm). Therefore, the claim that new particle formation causes differences in accumulation mode (0.5–2 μm) NC is unreliable and cannot be supported by the measured data. Current studies focus on new particles growing and forming new cloud condensation nuclei (CCN) that are typically 50 to 100 nm across. No study clearly demonstrates that new particles exhibit a sufficiently high growth rate to exert a significant influence on

accumulation mode aerosols. Nor is there any study clearly indicating that new particle formation is sparse in the central region of the SCS. Therefore, we have removed the claim that "new particle formation is the cause of differences in accumulation-mode number concentrations".

Instead, we have provided the reason for the difference in aerosol NCs between the East China Sea and the South China Sea: Aerosol emissions from the Yangtze River Delta region are higher than those from the Pearl River Delta region (Li et al., 2017). Due to the influence of aerosol transport, a greater amount of continental and anthropogenic aerosols from the Yangtze River Delta are delivered to the East China Sea compared to the amount transported from the Pearl River Delta to the South China Sea.

*Aerosol emissions from the Yangtze River Delta region are higher than those from the Pearl River Delta region (Li et al., 2017). Due to the influence of aerosol transport, a greater amount of continental and anthropogenic aerosols from the Yangtze River Delta were delivered to the East China Sea compared to the amount transported from the Pearl River Delta to the South China Sea.*

RC1.20: Line 214: please rephrase this for clarity "Meanwhile, the total marine aerosol NC observed in this study contained the aerosol coarse mode (2 μm ≤ Dp ≤ 10 μm) and the part of aerosol accumulation mode (500 nm ≤ Dp ≤ 2000 nm), and the NC was slightly lower than the marine aerosol NC in the Atlantic by Flores et al. (2020)". Why sudden comparison with Atlantic? Are SCS and Atlantic supposed to have similar concentrations? Or why just compare with Atlantic?

Response: We have revised this sentence.

*The SCS is one of the marginal seas of the Western Pacific. The summed NC observed in this study (54 cm$^{-3}$) was slightly lower than NC in the Western Pacific (83 cm$^{-3}$) by Flores et al. (2020).*

We have reselected the measurement data from the Flores et al. (2020) experiment, which were collected in the Western Pacific. The South China Sea is one of the marginal seas of the Western Pacific; meanwhile, both our experiment and the Western Pacific experiment in Flores were conducted in spring and located in the tropical zone and westerlies. Therefore, we initially hypothesize that the aerosol number concentrations from the two measurements should be similar.

RC1.21: What seasons did Cai et al. (2020) and Kong et al. (2016) carry their study in? easier to mention, it leaves the reader well informed. Did you compare the rain events between their study and your? is it possible they had less rain which kept the concentrations higher?

Response: We appreciate the reviewer for this useful suggestion. We have added the specific months and seasons of all cruises in Table 2. Specifically, Kong et al. (2016) conducted their observations in Autumn (2012.09 - 2012.10), while Cai et al. (2020) carried out their study in Summer (2018.08). This allows readers to gain a more comprehensive understanding of the research background.

**Table 2**

*Summary of the available study results on the shipboard observation of    marine aerosol NC (cm$^{-3}$)*

| Region | Time | Season | Latitude | Longitude | Parameter | Value | Parameter | Value | Reference |
|---|---|---|---|---|---|---|---|---|---|
| South China Sea | 2023.05 - 2023.06 | Spring | 21°N - 8°N | 115°E - 110°E | Accumulation mode ($n_{500-2000}$) | $52.4 \pm 35.0$ | $n_{500-10000}$ | $54.0 \pm 35.3$ | This Study |
| South China Sea | 2018.08 | Summer | 23°N - 19°N | 118°E - 108°E | $n_{400-32000}$ | 61 | | | Cai et al., 2020 |
| South China Sea | 2012.09 - 2012.10 | Autumn | 21°N - 20°N | 118°E - 113°E | $n_{120-10000}$ | 175 | | | Kong et al., 2016 |
| South China Sea | 2005.05 | Spring | 20°N - 18°N | 118°E - 113°E | Accumulation mode ($n_{50-2000}$) | $50.3 \pm 19.5$ | | | Lin et al., 2007 |
| East China Sea | 2005.05 | Spring | 30°N - 26°N | 122°E - 117°E | Accumulation mode ($n_{50-2000}$) | $109.2 \pm 51.8$ | | | Lin et al., 2007 |
| East China Sea | 2017.04 - 2017.05 | Winter | 28°N - 20°N | 130°E - 120°E | $n_{250-2500}$ | $57.4 \pm 40.9$ | $n_{2500-10000}$ | $57.5 \pm 41.3$ | Ma et al., 2022 |
| Western Pacific | 2017.04 - 2017.05 | Spring | 20°N - 0°N | 180°E - 130°E | $n_{100-19800}$ | $83 \pm 30$ | | | Flores et al., 2020 |

*Note.* In the column of the "Parameter", "n" indicated the NC and the subscripts indicated the particle size (nm); in the column of the "Latitude", "N" represented north latitude. The results of this study and these references were the overall average aerosol NCs.

We compared the rainfall events in their studies with those in ours. Kong et al. (2016) reported a conclusion consistent with ours: the marine aerosol NCs increased significantly during rainfall periods. Cai et al. (2020) observed a decrease in marine aerosol NCs during rainfall; the rainfall events were triggered by Tropical Storm Bebinca. During this rainfall periods, the air mass shifted from continental polluted air masses to marine clean air masses; moreover, the typhoon resulted in the removal of air pollutant in Huizhou and in Hong Kong, ultimately led to lower aerosol NCs in their study.

Regarding whether "they have less rain kept the concentrations higher", we cannot directly confirm this conclusion. The key reason is that the average aerosol NCs in our study were specifically calculated by excluding all rainfall periods. However, neither Cai et al. (2020) nor Kong et al. (2016) explicitly stated in their manuscripts whether their reported average aerosol NCs excluded rainfall events or included the entire observation period.

RC1.22: Line 218: "Although the differences in the observation seasons, the study region, and the particle size might influenced the average NC observations, it can still show that the marine aerosol was significantly affected by the continental transport

and the anthropogenic activity in the offshore areas according to the latitude and longitude." Why did you mention differences in study region? The line before this says Cai and Kong did their study in the same region as yours. Next line says differences in study region. Its confusing.

Response: We sincerely appreciate your detailed question. We apologize for the confusing expressions "differences in the study region". Inconsistent expressions between the manuscript can indeed easily cause misunderstanding and confusion for readers. To resolve this confusion, we have deleted the confusing expressions "differences in the study region".

We initially intended to use "the study region" to refer to different marine sub-regions of the South China Sea. For instance, the studies of Cai et al. (23°N - 19°N) and Kong et al. (21°N - 20°N) were conducted in the northern South China Sea, while our study (21°N - 8°N) almost covers the entire South China Sea (including its northern, central, and southern parts). Rather than suggesting that Cai, Kong, and we measured marine aerosols in different ocean areas. Different marine sub-regions of the South China Sea can also be expressed using distinct latitude and longitude ranges.

*Regarding the marine aerosol NCs in the same ocean area, the observations of Cai et al. (2020) and Kong et al. (2016) were significantly higher than the observations in this study. Although the differences in the observation seasons and the particle size might influenced the average NC observations, it can still show that the marine aerosol was significantly affected by the continental transport and the anthropogenic activity in the offshore regions according to the latitude and longitude.*

RC1.23: Line 222: "However, some studies found that the aerosols might be generated on the porous surface when impinged by liquid droplets" porous surface of what? I know the next lines make it clear but it would be better if you introduce the concept when you first mention it.

Response: Thanks very much for your suggestion. We have add the explanation.

*However, intense precipitation events can paradoxically elevate aerosol Ns. Some studies indicate that impaction of liquid droplets on porous surfaces (e.g., the ocean and ship surfaces) may generate aerosol particles (Bird et al., 2010; Joung & Buie, 2015; Zhou et al., 2020).*

RC1.24: Line 229: you mention only one size 4068nm. It's hard to say if ambient aerosols are just of one particular size. Please give a range.

Response: Thanks for the reviewer's comment. The specific range has been added as suggested.

*3786 to 4371 nm*

RC1.25: Line 243: ". High NCs ($\geq$ 150 cm-3) were observed almost entirely in which the WD were between NW and N that were caused by the high RH accompanied by the rainfall events, and the distributions of NCs were uniform when the wind was blowing in the other directions." So this high NC is the artificial aerosols created by porous surface of the ship and not natural aerosols? Because rainfall is supposed to

cause wet removal or deposition of aerosols and thus decreased concentrations

Response: These high concentrations of aerosol particles include both the artificial aerosols created by droplets impinging on the porous surface of ships and the natural aerosols created by droplets striking the ocean surface.

While wet deposition is indeed the dominant aerosol removal mechanism under most precipitation scenarios, some observations (Bird et al., 2010; Joung & Buie, 2014; Zhou et al., 2020) reveal a different regime where heavy rainfall episodes temporarily enhance aerosol concentrations through secondary production mechanisms. This phenomenon is specifically attributable to the fact that the droplets can release aerosols when they influence porous surfaces (e.g., the ocean surface and ship superstructure), and these aerosols can deliver elements of the porous media to the environment. Hence, after accounting for the observation environment and rainfall intensity, it is evident that short-duration heavy rainfall resulted in numerous raindrops impacting the ocean and ship surfaces, generating aerosol particles. Subsequently, the monitoring instrument captured some of these aerosol particles, ultimately contributing to the increased aerosol particle concentration, which can also be observed in Fig. 3 (the blue-shaded region). Consolidated evidence from prior research and current findings suggests that short-duration heavy rainfall may lead to a transient increase in accumulation mode particles.

We do not deny the mechanism by which rainfall can cause wet removal or deposition of aerosols. What is mentioned in this paper may merely be a special phenomenon, and there is no conflict between the two.

Reference:

Bird, J. C., de Ruiter, R., Courbin, L., and Stone, H. A.: Daughter bubble cascades produced by folding of ruptured thin films, Nature, 465,759-762, https://doi.org/10.1038/nature09069, 2010.

Joung, Y., Buie, C.: Aerosol generation by raindrop impact on soil, Nat. Commun., 6, 6083, https://doi.org/10.1038/ncomms7083, 2015.

Zhou, K., Wang, S., Lu, X., Chen, H., Wang, L., Chen, J., Yang, X., Wang, X.: Production flux and chemical characteristics of spray aerosol generated from raindrop impact on seawater and soil, J. Geophys. Res.-Atmos., 125, e2019JD032052, https://doi.org/ 10.1029/2019JD032052, 2020.

[Figure]

Fig. 3 The time series of the shipboard observations in the SCS from 21 May to 3 June 2023. The blue-shaded regions represented periods affected by rain events. (a) Trend of the aerosol size distributions. (b) Trends of NCs of the two aerosol particle modes (black solid line represented the NC of the coarse mode, and red solid line

represented the NC of the accumulation mode). (c) Trend of the WD. (d) Trends of the TOBS (dark orange solid line), T2m (light orange solid line), and SST (blue solid line). (e) Trends in the RH (gray solid line), the VIS (red solid line), and the rainfall intensity (dark blue solid line).

RC1.26: Table 2: when you write accumulation mode, please mention the size, I don't think there's any use of comparing with n10-400 because your measurement starts from 500nm.

Response: We appreciate your suggestion. All marine aerosol NC measurements now specify the particle size ranges covered, and data for the $n_{10-400}$ category have been removed as suggested.

RC1.27: Line 274:284: include this is methods instead of results

Response: We have relocated the offshore distance calculations to the Section 2.3.3.

*2.3.3 Distances from the coast*

*The ArcGIS path distance method was used to calculate distances from the coast. In equidistant projection, ship positions were used as input data, and coastline position data were used as reference lines for distance analyses. Considering the actual surface distance as well as horizontal and vertical factors, the shortest distance from the ship to the coastline can be calculated.*

The justification for selecting 50 km as the critical threshold for dividing the offshore and pelagic regions in the SCS falls within the scope of the Discussion section; so we have retained this part.

*This marine scientific research campaign started southward from the harbor of Zhanjiang (21°16'21.12" N, 110°23'45.17" E) on 21 May and reached up to the southernmost (8°5' N) point of this cruise on 3 June. In different latitudes of the SCS, there were vastly different marine aerosol distribution characteristics, meteorological parameters, and marine aerosol transport sources. Therefore, we assessed features of marine aerosol distribution at various distances from coast. We conducted real-time analysis of the 72-hour backward trajectories of air masses at the ship's location (Fig. 6a, b). The backward trajectory analysis indicated that the air masses had last passed over continental areas on 22 May 2023, 11:00 local time (LT), at a point 50 km from the coast (the red solid lines in Fig. 6b). Consequently, for all sampling locations within this 50 km boundary, the air masses had directly passed over mainland areas. This meant they carried continental and anthropogenic aerosols that ultimately influenced the aerosol distributions (Braun et al., 2020; Wu & Boor, 2021). For regions more than 50 km from the coast, the backward trajectory results consistently showed that the air masses did not pass over any mainland areas before reaching the sampling site (the blue solid lines in Fig. 6a). The prevailing wind direction was primarily from the southwest (Fig. 3c) in these regions, so aerosols could not be directly transported from the continent to the ship's location. Additionally, continental and anthropogenic aerosols, which were emitted from islands and countries surrounding the SCS, lost their original characteristics through the long-duration (over 72 hours) transport. These aerosols underwent atmospheric long-range*

*transport, dry deposition, wet deposition, and aging processes. Such processes led to the removal of continental aerosols or their gradual dilution and mixing with natural aerosols (Hodshire et al., 2019;Ohata et al., 2016; Xu et al., 2021). Over time, the continental and anthropogenic aerosols transformed or integrated into the background aerosols. Hence, 50 km from the coast was taken as the boundary distance to distinguish offshore and pelagic regions in this study.*

RC1.28: Figure 7b shows same shape of size distribution, of course, the concentrations are different. But doesn't same shape of distribution say same sources for the two regions? If one were affected by continental sources, wouldn't the distributions be slightly different?

Response: We sincerely appreciate the reviewer for the insightful suggestion. The same distribution shape does not indicate that the two regions have the same aerosol sources.

We plotted the number size distributions at the two time points with the largest difference in the distance from the coast (New Fig. 1). When the distance from the coast was 5 km, the measured aerosols were significantly affected by continental sources. When the distance from the coast was 500 km, the measured aerosols were barely influenced by continental sources. Notably, despite the distinct differences in continental source influence between these two regions, the aerosols still exhibited the same number size distribution shape, with only a more obvious difference in concentrations (higher in the offshore regions and lower in the pelagic regions). This directly demonstrates that similar distribution shapes do not imply the same aerosol sources.

Meanwhile, our finding is consistent with previous studies. Specifically, Ma et al. (2022) and Kong et al. (2016), who also conducted observations in the SCS, showed that the influence of continental sources mainly alters the concentration of marine aerosols (by adding anthropogenic particles) but does not lead to significant differences in aerosol distribution shape (within the range of 0.5–10 μm).

[Figure]

New Fig. 1 The NCs of average size distributions for marine aerosols of 0.5 to 10 μm diameters in different distances.

Reference:

Kong, Y. W., Sheng, L. F., Liu, Q., and Li, X. Z.: Impact of marine atmospheric process on aerosol number size distribution in the South China Sea, (in Chinese), Environ. Sci., 37, 2443-2452, 10.13227/j.hjkx.2016.07.005, 2016.

Ma, X., Jing, Z., Chang, P., Liu, X., Montuoro, R., Small, R. J., Bryan, F. O., Greatbatch, R. J., Brandt, P., Wu, D., Lin, X., and Wu, L.: Western boundary currents regulated by interaction between ocean eddies and the atmosphere, Nature, 535, 533-537, https://doi.org/10.1038/nature18640, 2016.

RC1.29: Line 298: Can you elaborate this more? ". The marine aerosols decreased slowly with the increasing particle diameters below 1.114 μm due to the transport effect." You didn't discuss the 'transport effect' before, so its difficult for the reader to associate this.

Response: Thanks very much for your suggestion. We have changed "transport effect" to "the influence of aerosol transport", and have also added discussion on "the influence of aerosol transport".

*Due to the influence of aerosol transport, the continental air masses carried continental and anthropogenic aerosols, which ultimately affected aerosol distributions in the 0.5-5.0 μm particle size range. The number size distributions in the offshore regions were obviously higher than in the pelagic regions in the 0.5-5.0*

*μm particle size range. The findings were consistent with the previous studies (Braun et al., 2020; Lorenzo et al., 2023).*

RC1.30: In figure 7c, where is the correlation (r=-0.87) shown between Acc NC and distance? All is see is boxplot and distance points in red. Where's the correlation? The distance on 5/31 was higher than 5/27 but still number concentrations were higher on 5/31. How do you explain this?

Response: Thank you for your constructive comments.

As mentioned in the manuscript, the NC of the accumulation mode (indicated by the height of the blue box) showed a decreasing trend with the increase distance from the coast (marked by red dots), which reflected a negative correlation between the NC and the distance. The correlation coefficient was obtained through our calculations. Therefore, in the revised manuscript, we clearly state that the corresponding correlation coefficient between the NC of the accumulation mode and the distance from the coast are derived from calculations, rather than directly observed from the graph.

*It was obvious from Fig. 7c that the NC of the accumulation mode showed a decreasing trend with the increasing distance from the coast, and the correlation coefficient between the NC of the accumulation mode and the distance from the coast was calculated as R = -0.87.*

On May 31 and May 27, the research vessel was located in the pelagic ocean regions. In the pelagic ocean region, the correlation coefficient between the NC of the accumulation mode and the distance from the coast was very low (R = -0.28), indicating that aerosol NCs were less affected by the distance in this regions. Compared with May 27 (5 m s$^{-1}$, 2.9 °C), the higher wind speed (10 m s$^{-1}$) and lower sea-air temperature difference (0.9 °C) on May 31 contributed to the higher aerosol NCs.

RC1.31: Your accumulation mode starts from 500nm, how can you show that transport brings in particles which are atleast 500nm for contributing to increase in NC of accumulation mode? Can you cite someone who has shown the size of transport particles in this range? You also mentioned higher wind speed in offshore compared to pelagic areas. How can you say the higher NC at offshore was due to transport and not high WS?

Response: We have added relevant research literature. Braun et al. (2020) and Lorenzo et al. (2023) demonstrated that in the South China Sea, continental air masses (carrying continental and anthropogenic aerosols) significantly influences aerosol concentration distribution within the 0.5–5 μm (i.e., 500 nm–5 μm) size range. Consistent with these findings, our results also show that continental air masses affected the aerosol distributions in the same 0.5–5 μm range, leading to notably higher number size distributions in offshore regions than in pelagic regions.

*Due to the influence of aerosol transport, the continental air masses carried continental and anthropogenic aerosols, which ultimately affected aerosol distributions in the 0.5-5.0 μm particle size range. The number size distributions in*

*the offshore regions were obviously higher than in the pelagic regions in the 0.5-5.0 μm particle size range. The findings were consistent with the previous studies (Braun et al., 2020; Lorenzo et al., 2023).*

Reference:

Braun, R. A., Aghdam, M. A., Bañaga, P. A., Betito, G., Cambaliza, M. O., Cruz, M. T., Lorenzo, G. R., MacDonald, A. B., Simpas, J. B., Stahl, C., and Sorooshian, A.: Long-range aerosol transport and impacts on size-resolved aerosol composition in Metro Manila, Philippines, Atmos. Chem. Phys., 20, 2387–2405, https://doi.org/10.5194/acp-20-2387-2020, 2020.

Lorenzo, G. R., Arellano, A. F., Cambaliza, M. O., Castro, C., Cruz, M. T., Di Girolamo, L., Gacal, G. F., Hilario, M. R. A., Lagrosas, N., Ong, H. J., Simpas, J. B., Uy, S. N., and Sorooshian, A.: An emerging aerosol climatology via remote sensing over Metro Manila, the Philippines, Atmos. Chem. Phys., 23, 10579–10608, https://doi.org/10.5194/acp-23-10579-2023, 2023.

The wind speed in the offshore regions (10.7 m s⁻¹) registered a 0.2-fold increase compared to that in the pelagic regions (8.6 m s⁻¹); such a small difference in wind speed (a 0.2-fold increase) could not account for the large difference in NCs (a 1.2-fold increase). Meanwhile, as shown in New Fig. 1, the wind speed was 6.0 m s⁻¹ at an offshore distance of 5 km and 6.1 m s⁻¹ at an offshore distance of 5 km. Although the difference in wind speed was extremely small, there were significant differences in aerosol NCs and size distribution. This further indicates that the difference in aerosol NCs between offshore and pelagic regions is more significantly affected by continental transport.

[Figure]

New Fig. 1 The NCs of average size distributions for marine aerosols of 0.5 to 10 μm diameters in different distances.

RC1.32: In Figure 8, please add the arrows showing direction of air
Response: We have added the arrows in Fig. 6.

[Figure]

*Fig. 6 (a) The 72-h backward trajectory air mass source traces in the offshore (red solid lines) and pelagic (blue solid lines) regions. (b) Detailed map of the backward trajectory air mass source traces passing through the mainland areas (© Google Earth). The white arrows represented the direction of air mass transport.*

RC1.33: Line 324: again words like significant doesn't make much sense. Talk in numbers. Out of all components dust shows the maximum change between pelagic and offshore, the difference in the rest of the components do not look 'siginificant'
Response: Thank you for your helpful comments. We have removed words such as "significant" and instead used specific numerical values for illustration.

*Fig. 7d indicated a difference in the distribution of marine aerosol components between offshore and pelagic regions. In the offshore regions, the proportions of dust (DUST$_{10}$; Dp ≤ 10 μm) and sulfate aerosols (SO$_4^{2-}$) were 5.04 % and 1.36 %, which were higher than those in the pelagic regions (1.45 % and 0.97 %, respectively). The higher concentrations of dust and sulfate aerosols further indicate that continental aerosols have influenced the aerosol components in the offshore regions. Meanwhile, in the pelagic regions, the proportions of dimethyl sulfide (DMS), organic carbon (OC), and sulfur dioxide (SO$_2$) were 0.15 %, 1.2 %, and 1.7 %. These proportions were higher than those in the offshore regions (0.1 %, 0.84 %, and 0.56 %, respectively).*

RC1.34: Rephrase line 367: Fig 9a showed a clear diurnal variation emerged
Response: Thanks very much for your suggestion. We have revised this sentence.
*Fig. 8a showed a clear diurnal variation. For the accumulation mode, the variations became readily visible and followed a definite pattern:*

RC1.35: During the daytime, WS increases and SST and difference in temperature between surface and 2m also decreases; so shouldn't the daytime NC be increasing because all the factors you mentioned for the transition period aligns for daytime, yet there's no clear trend in daytime. Why is that?
Response: The aerosol NC, WS, SST, and SST-$T_{2m}$ at the end of the daytime were 62.1 cm$^{-3}$, 10.0 m s$^{-1}$, 29.8 °C, and 0.8 °C, respectively. In contrast, the values of these variables at the start of the daytime were 47.8 cm$^{-3}$, 8.5 m s$^{-1}$, 29.9 °C, and 1.2 °C, respectively. As noted by the reviewer, the WS increased during the daytime, while both SST and SST-$T_{2m}$ decreased. Notably, the NC exhibited a distinct increase (from 47.8 cm$^{-3}$ to 62.1 cm$^{-3}$, a 0.3-fold increase). This increasing trend in aerosol number concentration during the daytime is also observable in New Fig. 2. Furthermore, as shown in Fig. S2 between 14:00 and 18:00, the increase in WS and the decreases in both SST and SST-$T_{2m}$ contributed to the rise in aerosol NCs.
The difference in aerosol NC (14.3 cm$^{-3}$) may be difficult to perceive due to the relatively large vertical axis range, which likely led to the reviewer's misunderstanding. However, we chose this vertical axis range primarily to prevent overlapping of the multiple lines (for WS, SST, SST-$T_{2m}$, and NC), thereby making it easier for readers to observe the variation of each parameter.

[Figure]

New Fig. 2 The variations of total NCs, SST, SST-$T_{2m}$, and WS in daytime.

RC1.36: Line 433, add references of studies that showed entrained air decreased with increasing SST. Entrained where? sea surface or boundary layer? clarify

Response: We have revised this sentence.

*In the pelagic region of the SCS, consolidated evidence from prior research and current findings suggests that elevated SST likely suppress near-surface air entrainment volumes, consequently decreasing the plunging jets.*

RC1.37: Instead of holding wither WS or SST constant for SST-T2m correlations, I suggest you perform multi linear regression or lasso regression or any suitable regression analysis to study the impact of these factors on NC variability. When you hold WS const for Fig 13, SST is still varying and when you hold SST const for Fig 14, WS is still varying.

Response: Thanks very much for the valuable and insightful comment. We fully acknowledge the limitation pointed out by the reviewer: when we held either WS or SST constant in Figures 13 and 14, the other variable (SST or WS) was still varying. This single-factor control approach cannot fully eliminate the influence of the other variable (SST or WS) on NC. The varying SST or WS may potentially obscure the independent relationship between SST-$T_{2m}$ and NC.

However, we would like to emphasize that:

The correlation coefficient between SST-$T_{2m}$ and NC reaches as high as R = -0.9 with a very significant p-value (p < 0.001) (Fig. 12), and the negative trend is visually striking even with the scattering caused by WS/SST fluctuations. This suggests that despite the variability of WS and SST, the dominant negative relationship between SST-$T_{2m}$ and NC is still robustly captured.

The current figures (Fig. 13 and 14) are designed to intuitively illustrate the qualitative trend between SST-$T_{2m}$ and NC under WS and SST intervals conditions. We believe that the current figures can already clearly present the preliminary connection between SST-T2$_m$ and aerosol NC.

We completely agree that multi-factor regression (e.g., multiple linear regression or lasso regression) is essential for quantitatively disentangling the individual and combined effects of WS, SST, and SST-$T_{2m}$ on NC variability. Therefore, in the subsequent parameterization modeling, we will prioritize the adoption of these methods to derive a more precise and reliable formula describing the relationship between SST-$T_{2m}$ and NC, while accounting for the interference of WS and SST fluctuations. As mentioned in our manuscript, we plan to incorporate SST-T2m into the source function in subsequent studies to improve the accuracy of existing source functions.

[Figure]

Fig. 12 The relationship between the NC of all aerosol particle modes and WS (a), SST (b), and SST-$T_{2m}$ (c). The error bars represented the standard deviations. The R represented the Pearson correlation coefficients, and the p values were performed to test whether the correlations were significant.

RC1.38: Does the size distribution remain the same with increasing SST? What would you comment on the increase of diameter with warming SST as observed by Saliba (2019)?

Response: Thanks very much for your valuable comments. New Fig. 3 shows the number size distributions of marine aerosols at different sea surface temperatures (SST). Qualitatively, the size distribution almost maintains the same shape with warming SST. Quantitatively, the geometric standard deviation (GSD) shows no significant variation. The mode diameter ($d_m$) slightly increases as the sea surface temperature rises.

We calculated $d_m$ and GSD of marine aerosol number size distribution under different sea surface temperatures (New Table 1). The $d_m$ and GSD were obtained by fitting the 0.5 to 5.0 μm range of the measured size distribution with a single lognormal mode. The selected size range and calculation method are consistent with those used by Saliba et al. (2019). Importantly, in the South China Sea, we also observed the phenomenon that the diameter increases as the sea surface temperature rises (from 0.564 μm at SST = 28 °C to 0.582 μm at SST = 30 °C), which is consistent with the findings of Saliba et al. (2019).

This consistency verifies the broad applicability of the SST-dm relationship across different ocean areas, particularly in the tropical marginal sea (South China Sea) that was less studied previously. Moreover, their research conclusions have great significance for understanding marine aerosol production and its radiative impacts in the marine boundary layer. They provide constructive insights for our future research. Specifically, we will optimize the expression of the SST-dm relationship for the South China Sea to better predict marine aerosol concentrations. Ultimately, this regionally optimized expression will be incorporated into the parameterization of future climate models, aiming to improve the prediction accuracy in the western Pacific marginal seas.

[Figure]

New Fig. 3 The NCs of average size distributions for marine aerosols of 0.5 to 10 μm diameters in different sea surface temperatures.

**New Table 1** Summary of mode diameter of the marine aerosol number size distribution ($d_m$) and geometric standard deviation (GSD) within different sea surface temperatures.

| SST (℃) | $d_m$ (μm) | GSD |
|---|---|---|
| 28 ℃ | 0.564 | 1.137 |
| 29 ℃ | 0.570 | 1.145 |
| 30 ℃ | 0.582 | 1.266 |

Reference:
Saliba, G., Chen, C.-L., Lewis, S., Russell, L. M., Rivellini, L.-H., Lee, A. K. Y., Quinn, P. K., Bates, T. S., Haëntjens, N., Boss, E. S., Karp-Boss, L., Baetge, N., Carlson, C. A., and Behrenfeld, M. J.: Factors driving the seasonal and hourly variability of sea-spray aerosol number in the North Atlantic, P. Natl. Acad. Sci. USA., 116, 20309-20314, https://doi.org/10.1073/pnas.1907574116, 2019.

**Answer to Reviewer 2**

Summary
RC2.1: This manuscript present APS measurements conducted in the South China Sea. The authors use APS measurement and apportion them to "accumulation" and "coarse" mode particles. The authors conclude that there exist significant differences in aerosol number concentrations and size distributions of the aerosol in offshore regions and pelagic regions. Furthermore, the authors find an inverse relationship between sea-air temperature differences and aerosol number concentrations. Overall, the authors need to address the below points before this manuscript can be published.

Response: We sincerely thank the referee for the thoughtful and constructive comments on our paper. These comments made our study much more targeted, structured, and understandable. In response to your suggestions, we have incorporated additional details in the manuscript and rewritten the text following the major and general comments below.

**Major comments:**

RC2.2: I have two major concerns for this study. First is the use of a single instrument (APS) to quantify the marine size distribution over the South China Sea. The APS measures aerosol concentrations in the size range of 0.5 - 20 μm with high uncertainty in the first channel which aggregates all aerosol with aerodynamic diameter <=0.5 μm. I am not convinced that the authors can achieve a defensible case for separating accumulation mode and coarse particle mode using the APS as the only measuring instrument.

Response: We appreciate the reviewer's insightful and valuable comments regarding the use of the APS as the sole instrument for quantifying aerosol size distributions and separating particle modes. These points are critical for validating the reliability of our aerosol data, and we have addressed them through supplementary experiments and detailed analyses.

As noted in the TSI official documentation, the APS (Aerodynamic Particle Sizer) is designed to measure aerosol NCs in the size range of 0.5-20 μm. Specifically, the first channel of the APS used in this study is defined for particles with an aerodynamic

diameter ≤0.523 µm, which has raised questions about whether it aggregates all aerosol particles smaller than 0.5 µm (the instrument's lower limit). To resolve this ambiguity, we conducted a 15-day field inter-comparison experiment with multiple aerosol instruments to validate the accuracy of the APS, particularly its first channel.

To verify the APS measurements, we performed a field inter-comparison experiment from 2 October to 17 October 2025 at a decommissioned wharf in Zhuhai, Guangdong Province, China (22°12′ N, 113°37′ E). This site is remote from industrial emissions and roads, with an unobstructed 180° view of the northern SCS, to ensure marine aerosol measurements are achievable. Three instruments were deployed side-by-side in an open environment with 10-cm long tubes. These tubes were fixed to the railing at 30° to the horizontal and faced the sea surface (to simulate the previous observation scenario and minimize terrestrial interference). The three instruments comprised: a Model 3321 APS (measuring 0.5-20 µm) spectrometer (TSI Incorporated, USA), a Portable Optical Particle Spectrometer (POPS) (measuring 0.115-3.37 µm; Handix Scientific, USA), and a Model 11-D Portable Aerosol Spectrometer (measuring 0.25-30 µm; GRIMM, Germany). The aerosol data resolution was set to 10 min in this inter-comparison experiment. Key results included:

1. Consistency in number size distributions: The aerosol number size distributions measured by all three instruments showed high similarity (Fig. S2a) and low discrepancy, confirming general agreement in capturing particle size trends.

2. Concentration comparison for sub-2 µm particles: Since direct channel-to-channel matching was not feasible due to the differing size bins, we compared total concentrations within overlapping ranges relevant to our study's accumulation mode: 0.5–1.981 µm for APS, 0.475-1.99 µm for POPS, and 0.488-2.14 µm for GRIIM 11-D. All three instruments exhibited consistent diurnal trends (Fig. S2b). Strong correlations between APS and the other instruments (R = 0.92 vs. GRIIM 11-D; R = 0.94 vs. POPS) further validate the APS's accuracy (Figs. S3a, S3b).

[Figure]

Fig. S2 (a) The NCs of average size distributions for different aerosol measurement instruments (black solid line represented the APS data, orange solid line represented the 11-D data, and red solid line represented the POPS data). (b) Trends of NCs for different aerosol measurement instruments.

[Figure]

[Figure]

Fig. S3 The scatter plots of (a) NCs of 11-D data and APS data, (b) NCs of POPS data and APS data.

3. Implications for the first APS channel: During this observation, the average NC for sub-2 μm particles was 66.5 cm$^{-3}$ for APS, 75.36 cm$^{-3}$ for GRIIM 11-D, and 98.01 cm$^{-3}$ for POPS. If the first channel of the APS were able to aggregate all aerosol particles smaller than 0.5 μm, the average sub-2-μm aerosol NC measured by the APS would be significantly higher than those measured by the POPS and 11-D. However, through calculation and comparison, we found that the average NC of the APS was instead the lowest, with differences of 8.86 cm$^{-3}$ and 31.51 cm$^{-3}$ compared to those of the 11-D and POPS, respectively. These obvious differences are caused by the additional aerosol particles that the 11-D and POPS instruments can measure compared to the APS. This phenomenon indicates that the size bins of APS are smaller than those of the 11-D and POPS. Since the 11-D starts measuring from 0.488 μm, the first channel of the APS (≤0.523 μm) is not significantly affected by particles smaller than 0.488 μm. The first channel range is likely 0.5–0.523 μm as mentioned in the TSI official documentation.

We have added details of this field inter-comparison experiment to the Methods section and included supplementary figures (Figs. S2, S3) to substantiate these claims. We believe these results demonstrate the APS's suitability and reliability for our analysis of marine aerosol size distributions in the SCS.

The field inter-comparison experiment confirms that APS reliably captures particles in these ranges, and the clear bimodal distribution observed (peaks at 0.542 μm and 1.981 μm) further supports the validity of this mode separation.

In the Method section, we have added the explanations regarding the selection of accumulation mode and coarse mode. In this study, the accumulation mode ranges from 0.5 to 1.981 μm, and the coarse mode ranges from 1.981 to 10.37 μm.

RC2.3: My second concern is how the authors define pelagic regions as being 50 km or more from the coast. Even at moderate wind speeds of 6 m/s, it would only take about two and a half hours to transport the aerosol from continents to the ship's location. This period is significantly shorter than the time it would take to remove

continental aerosol through wet or dry deposition. Even at 300 km from the shore, it would take about 14 hours for the air originating over the continent to reach the ship's location. As a result, I don't find the author's claim that the aerosols sampled 50 km away from the coast are free from continental influence to be defensible.

Response: Thank you very much for your valuable comment on the definition of pelagic regions. In fact, our definition of the pelagic region ($\geqslant$ 50 km from the coast) is not solely based on distance; it is comprehensively determined by combining the 72-h backward trajectory air mass source traces with the prevailing meteorological conditions in the South China Sea during summer. Specifically:

1. During the entire cruise, we conducted real-time analysis of the 72-hour backward trajectories of air masses at the ship's location (Fig. 6a, b). The backward trajectory results indicated that the last instance (at 11:00 on May 22) when the air masses passed over continental areas occurred at a location 50 km from the coast. Consequently, for all sampling locations within this 50 km boundary, the air masses had directly passed over continental areas - meaning they carried continental and anthropogenic aerosols, which ultimately exerted an influence on aerosol distributions in the offshore regions.

2. For the regions where the ship was located at $\geqslant$ 50 km from the coast (defined as pelagic regions in the original text), the backward trajectory results consistently showed that the air masses did not pass through any continental areas before reaching the sampling site (Fig. 6a). On the one hand, the prevailing wind direction was primarily from the southwest (Fig. 3c) in these regions, so aerosols could not be directly transported from the continent to the ship's location. On the other hand, the continental and anthropogenic aerosols, which were emitted from islands and countries surrounding the SCS, lost their original characteristics through the long-duration (over 72 hours) transport. These aerosols underwent atmospheric transport, deposition, and aging processes. Such processes resulted in gradual dilution and mixing with natural aerosols. Over time, the continental and anthropogenic aerosols transformed or integrated into the background aerosols, losing their original characteristics.

Overall, we consider it reasonable that the aerosols sampled 50 km away from the coast are not affected by continental aerosols during the whole cruise period.

[Figure]

Fig. 6 (a) The 72-h backward trajectory air mass source traces in the offshore (red solid lines) and pelagic (blue solid lines) regions. (b) Detailed map of the backward trajectory air mass source traces passing through the mainland areas (© Google Earth).

In the revised manuscript, we have supplemented the following content to better defend this definition:

We have explicitly emphasized that the "50 km" is not an arbitrary distance threshold but a critical boundary derived from the combination of trajectory analysis and meteorological conditions—its core significance lies in marking the "boundary where air masses are completely free from continental trajectory influence" rather than a simple "distance threshold for excluding continental aerosol transport".

We have moved the distributions of the backward trajectory air mass to this section. The detailed data of air mass backward trajectories at typical pelagic sampling sites (≥ 50 km offshore) clearly show the oceanic origin of the air masses and their lack of contact with continental areas.

*Considering the actual surface distance and horizontal and vertical factors, the shortest distance can be calculated from the ship to the coastline positions. Meanwhile, we conducted real-time analysis of the 72-hour backward trajectories of air masses at the ship's location (Fig. 6a, b). The backward trajectory analysis indicated that the last time the air masses passed over continental areas was at 11:00 on May 22, at a point 50 km from the coast (the red solid lines in Fig. 6b). Consequently, for all sampling locations within this 50 km boundary, the air masses had directly passed over mainland areas. This meant they carried continental and anthropogenic aerosols that ultimately influenced the aerosol distributions. For regions more than 50 km from the coast, the backward trajectory results consistently showed that the air masses did not pass over any mainland areas before reaching the sampling site (the blue solid lines in Fig. 6a). On the one hand, the prevailing wind direction was primarily from the southwest (Fig. 3c) in these regions, so aerosols*

*could not be directly transported from the continent to the ship's location. On the other hand, the continental and anthropogenic aerosols, which were emitted from islands and countries surrounding the SCS, lost their original characteristics through the long-duration (over 72 hours) transport. These aerosols underwent atmospheric long-range transport, dry deposition, wet deposition, and aging processes. Such processes led to the removal of continental aerosols or their gradual dilution and mixing with natural aerosols (Hodshire et al., 2019;Ohata et al., 2016; Xu et al., 2021). Over time, the continental and anthropogenic aerosols transformed or integrated into the background aerosols. Hence, distance from the coast of 50 km was taken as the boundary distance to distinguish offshore and pelagic regions in this study.*

We believe that with the above supplementary analysis and explanation, our definition of the pelagic region ($\geq$ 50 km from the coast) and the claim that the aerosols sampled in the pelagic regions are not affected by continental aerosols have become more rigorous and defensible.

**Specific comments:**

RC2.4: Line 47: define NC
Response: The definition of NC has been added as suggested.
*number concentration (NC)*

RC2.5: line 70: define SST
Response: The definition of NC has been added as suggested.
*sea surface temperature (SST)*

RC2.6: line 79: "surface tension" and not "tension"
Response: We have changed "tension" to "surface tension".

RC2.7: Line 90: not sure what is meant by : "the subsequent updates simultaneously were lacking." reword
Response: Thanks for the reviewer's comment. To clarify, our intended meaning was that "the marine aerosol NCs and meteorological observation data in the SCS are relatively outdated; subsequent updates also remain limited".
As suggested by Reviewer 3, this part has been deleted.

RC2.8: Line 120: "were selected for future analysis". I think the work "future" is misleading here. I suggest removing
Response: Thanks very much for your suggestion. We have deleted "future" (L120)

RC2.9: Figure 1 can be moved to the SI
Response: We appreciate the reviewer for this useful suggestion. We have moved Fig. 1 to the SI.

RC2.10: Line 185: "The new aerosol generation events often accompanied the increased nucleation events." This sentence needs rewording

Response: We have revised this sentence and added more descriptions.

*The previous studies confirmed that new aerosol generation events were frequently initiated by episodes of intensified nucleation. This elevated nucleation activity, characterized by rapid cluster formation, provided the initial seed population essential for atmospheric particle production. Subsequent growth via condensation of vapours ultimately caused an increase in NCs.*

RC2.11: Line 202: "Fig. 4 a-b presented the trends of the aerosol size distribution and the comparison of the accumulation and coarse mode particle NCs" The authors do not describe how the two aerosol modes were separated? This should be explicitly mentioned in the method section. Was there a threshold diameter above which the authors consider particles to be coarse? (based on the next paragraph it seems that the threshold is 2 μm which seems arbitrary). Also accumulation mode particles are generally thought to include 0.1 - 0.5 μm particles which are unfortunately not measured by the APS. Please refer to my general comment above.

Response: In the Method section, we have added the explanations regarding the selection of accumulation mode and coarse mode. In this study, the accumulation mode ranges from 0.5 to 1.981 μm, and the coarse mode ranges from 1.981 to 10.37 μm.

We acknowledge that the traditional accumulation mode (0.1–2 μm) includes particles smaller than 0.5 μm, which are not measured by the APS. However, based on the current aerosol data, we have found that some of our research results can complement those of previous studies. For instance, Atwood et al. (2017) observed that in the SCS, continental transport causes variations in the aerosol size distribution shape within the range of 40–500 nm. Our study, in turn, just supplements the impact of continental transport on the aerosol size distribution in the SCS for particles larger than 500 nm.

Our research also demonstrates certain value: it fills gaps in shipboard aerosol measurement results in the SCS and identifies the influence of sea-air temperature difference on aerosol NC, which is crucial for the future improvement of marine aerosol source functions.

RC2.12: Line 204: "We found that the marine aerosol NC changed drastically with the temporal differences during the shipboard observation period." by how much? This is not described in the text

Response: Thanks for the reviewer's comment. We have rewritten this sentence (L204).

*During the shipboard observation period, the average summed (0.5 μm ≤ Dp ≤ 10.37 μm) NC was 54.01 cm$^{-3}$, NC of aerosol accumulation mode was 52.35 cm$^{-3}$, and NC of aerosol coarse mode was 1.66 cm$^{-3}$. For these three aerosol modes, the NCs varied from 18.46 to 89.38 cm$^{-3}$, 17.39 to 87.31 cm$^{-3}$, and 0.83 to 2.49 cm$^{-3}$, respectively, during the shipboard observation period, exhibiting substantial temporal fluctuations.*

RC2.13: Line 222: "However, some studies found that the aerosols might be generated on the porous surface when impinged by liquid droplets (Bird et al., 2010; Joung & Buie, 2015; Zhou et al., 2020)." I fail to understand the meaning of this sentence and how it logically connects with the previous one

Response: We have revised this sentence and added more descriptions.

*Wet deposition through scavenging by rainfall process is a critical sink for aerosols (Atlas & Giam, 1998; Radke et al., 1980). However, intense precipitation events can paradoxically elevate aerosol Ns. Some studies indicate that impaction of liquid droplets on porous surfaces may generate aerosol particles (Bird et al., 2010; Joung & Buie, 2015; Zhou et al., 2020). By accounting for observation environment and rainfall intensity, short-duration heavy rainfall resulted in numerous raindrops impacting the ocean and ship surfaces, generating aerosol particles. Subsequently, the monitoring instrument captured some of these aerosol particles, ultimately contributing to the increased aerosol particle concentration observed in Fig. 4 (the blue-shaded regions).*

RC2.14: Lin 232: "The correlation coefficient between the two aerosol particle modes was R = 0.71." The fact that the authors observed a good correlation between the "accumulation" and "coarse" particle modes is further evidence that the two sources are dependent and likely sea-spray. However, from observation, we know that sulfates and organics also contribute significantly to the accumulation mode particle size distribution and number (e.g., Saliba et al. 2020 https://doi.org/10.1029/2020JD033145) . As a result, if the entire accumulation mode was included, we should not expect such high correlations between the two.

Response: Thank you for raising this important point. We acknowledge that sulfate and organic aerosols typically contribute significantly to the accumulation mode particle size distribution and number (e.g., Saliba et al. 2020). However, our analysis demonstrates that these components contributed negligibly ($\leq$ 3% combined) to aerosol composition during the study period, while sea spray aerosols constituted > 90% (Fig. 6d). Crucially, the Model 3321 APS spectrometer used in this cruise measures part accumulation-mode particles within the particle diameters (0.5 - 1.981 μm). Thus, the high correlation (R = 0.71) between the "part accumulation" and "coarse" particle mode particles is reliable.

We would like to thank you again for your constructive comments. Meanwhile, Saliba's study can also strongly support the discussion in this paper regarding the differences in aerosol composition between offshore and pelagic regions. Therefore, we have cited this study in the revised manuscript to enhance the scientific rigor and reliability of the manuscript.

RC2.15: Line 238: "The observed air temperature was in agreement with the reanalyzed air temperature from Merra-2,..." Provide R2 of relation

Response: We have added the correlation coefficient between the observed air temperature and the reanalyzed air temperature from Merra-2 (L240).

*The observed air temperature was consistent with MERRA-2 reanalyzed air temperature (R = 0.719), whereas the average $T_{2m}$ and SST over the whole observation period were closer to 29.0 ℃ and 29.7 ℃.*

RC2.16: Line 240. I assume that VIS refers to visibility. How was this calculated?

Response: We thank the reviewer for seeking clarification. "VIS" does indeed refer to meteorological visibility, which was directly measured (not calculated) using a Belfort Model 6400 visibility sensor deployed onboard (L133 and 138). The Belfort Model 6400 is a forward-scatter visibility sensor. Its transmitter emits a scattered light measurement system uses an infrared LED (wavelength: 880 nm, power: 330 mW) to project a controlled beam into an atmospheric sample volume. Particulate matter and aerosols within this volume induce forward scattering of the incident IR light. The receiver quantifies the intensity of forward-scattered IR radiation. This scattering signal is then used to calculate the atmospheric extinction coefficient. VIS is derived algorithmically from the extinction coefficient using the established relationship:

VIS = 3 / Extinction Coefficient

RC2.17: Line 248: "For example, the NCs changed little in the region of 0-6 m s-1 WS because the WS was low for activation of the spume droplets and the marine aerosol generations." This is misleading, you can still generate sea-spray particles at low wind speeds. Spume represents one generation mechanism of sea-spray, usually at high wind speed. Bubble bursting process can occur at much lower speeds

Response: We sincerely thank the reviewer for this critical clarification. The text has been revised to:

*For example, marine aerosol NCs generated by the bubble bursting process at low WS showed little variation, and the low WS was insufficient to activate spume droplet production. Consequently, no significant variation in NCs were observed in the 0-6 m $s^{-1}$ WS range.*

RC2.18: Figure 4: the time axis for (a) is not aligned with the rest of the figures (b - e). Align all subplots to make it easier to interpret the figure.

Response: All subplots in Figure 4 (a-e) are now aligned on the time axis to enable clearer interpretation.

RC2.19: Figure 6: Again, the fact that both "accumulation" and "coarse" particle modes exhibited similar dependence with wind speed suggests to me that you are only capturing a part of the accumulation mode particles that are dominantly sea-spray particles. Other accumulation mode particles, mainly those formed through secondary processes, are not included in your definition of accumulation mode.

Response: Thank you for raising this important point. We think the accumulation mode measurement data include particles formed through secondary processes.

Firstly, secondary marine aerosols mainly include: secondary inorganic marine aerosols dominated by sulfate ($SO_4^{2-}$) (i.e. the sulphate not associated with primary sea spray) (Charlson et al. 1987; O'Dowd et al. 1997; Shaw 1983), and secondary

organic marine aerosols dominated by methane sulphonic acid (an oxidation product of the dimethyl sulfide (DMS)) (Berresheim et al. 2002; Facchini et al. 2008). As shown in Figure 7d, these aerosols contribute minimally and therefore do not significantly affect the relationship between accumulation mode particles and wind speed.

[Figure]

Fig. 7 Classification of the shipboard observation path in the SCS: (a) Accumulation and coarse mode particle sizes graded NCs in the offshore and pelagic regions. For the box plots, the boxes represented the 25th to 75th percentile value, the black whisker represented the maximum and minimum range, the black triangle represented the 1.5 inter-quartile range, the black diamond marker represented the mean value, and the black horizontal line represented the median value. (b) The NCs of average size distributions (the solid lines and circles) and standard deviations (the shaded areas) for marine aerosols in the offshore and pelagic regions. (c) The daily average variations of the proportions and the NCs of two aerosol particle modes were shown with the distances from coast. (d) The distributions of marine aerosol components in the offshore and pelagic regions. The pie charts showed the average aerosol composition based on the mass concentrations from the Merra-2 aerosol dataset during the whole cruise period.

Additionally, the box plots in Figure 5 of the manuscript show that the dependence of accumulation and coarse mode particles on wind speed is not similar. The coarse mode particles exhibit a clear positive linear relationship with wind speed, whereas the accumulation mode shows no such relationship. Meanwhile, as noted by Reviewer 3, a certain degree of dispersion persists in the scatter plot of NCs versus WS, even after applying constraints on specific meteorological parameters (e.g., relative

humidity and rainfall). These differing dependencies and the scatter in the data indicate that, besides sea spray particles, our observations also captured aerosols formed by secondary processes.

[Figure]

Fig. 5 The scatter plots of (a) NCs of the aerosol accumulation mode and WS, (b) NCs of the aerosol coarse mode and WS. The observational data were binned to the WS intervals equal to 3 m s⁻¹; the boxes represented the 25th to 75th percentile value, the black whisker represented the 1.5 inter-quartile range, the black diamond marker represented the mean value, and the black horizontal line represented the median value in the box plots.

[Figure]

Thus, we think that the accumulation mode particles in our definition include those formed through secondary processes.


RC2.20: Line 292: "with little differences in the offshore and pelagic regions, where the NCs of the coarse mode in the offshore areas were 2.68 cm$^{-3}$ and 1.57 cm$^{-3}$ in the pelagic areas" Looking at Fig 7, it seems that the differences (although small) might be statistically significant. Why is that?

Response: Thank you very much for your valuable comment. We apologize for the misunderstanding caused by the phrase "with little differences" and agree that the differences are statistically significant (although small). We have rewritten this part and clarified that these differences are statistically significant. Below, we will explain why this difference is statistically significant:

The difference between the offshore and pelagic regions is 1.11 cm$^{-3}$, which may seem "small" in absolute numerical terms, but it is critical to compare it to the standard error (the denominator of the t-value):

$$\text{Standard error (SE)} = \sqrt{\frac{S^2_{\text{offshore}}}{N_{\text{offshore}}} + \frac{S^2_{\text{pelagic}}}{N_{\text{pelagic}}}} \approx 0.03085$$

where $S_{\text{offshore}}$ is the standard deviation of NC in the offshore regions, $N_{\text{offshore}}$ is the number of samples in the offshore regions, $S_{\text{pelagic}}$ is the standard deviation of NC in the pelagic regions, $N_{\text{pelagic}}$ is the number of samples in the pelagic regions.

The difference (1.11) is approximately 36 times larger than the SE (0.03085). This large ratio directly leads to a high t-value (t ≈ 36), which far exceeds the critical threshold for $p < 0.05$ ( $|t| > 1.96$). In addition, the large sample size effectively suppresses random fluctuations in the shipboard observation data, strengthening the statistical significance of the mean difference. Therefore, the differences are statistically significant ($p < 0.001$).

*The average NC of the accumulation aerosol mode in the offshore regions was 2.2 times higher than that in the pelagic regions, and the differences were statistically significant ($p < 0.001$). Similarly, the comparison of the coarse mode NC revealed the same result as that for the accumulation mode. The coarse mode NC in the offshore regions (2.68 cm$^{-3}$) was also significantly higher than in the pelagic regions (1.57 cm$^{-3}$), with a statistically significant difference ($p < 0.001$).*

RC2.21: Figure 7: Why are coarse mode particle concentrations higher in the off-shore region compared to the pelagic region? Are the differences statistically significant?

Response: Compared with pelagic regions, the higher WS (Table 3) in offshore regions leads to a higher production of marine aerosols, and these regions are also more strongly influenced by the transport of continental air masses. The combined effects of this increased aerosol production and transport ultimately result in a higher coarse mode particle concentration. The differences are statistically significant ($p < 0.001$).

RC2.22: Line 300: "However, in the 5.0-10 μm particle size range, the number size distributions in the offshore areas were in excellent agreement with those in the pelagic areas." However, at concentrations of ~1 cm-3 the uncertainty of the instrument can be as large which would make any comparison meaningless. I suggest the author add a discussion of measurement uncertainty.

Response: Thank you for your constructive comment. We fully agree that the measurement uncertainty of the instrument under specific aerosol number concentration conditions may lead to a significant increase in errors, which could ultimately render the comparison results meaningless. In response to this critical consideration, we have supplemented a dedicated discussion on measurement uncertainty in the revised manuscript.

*However, in the 5.0-10 μm particle size range, the number size distributions from offshore and pelagic areas exhibited close agreement, demonstrating consistent correlation patterns that remained robust against instrumentation limitations. This comparability persists despite APS measurements in this range having inherent uncertainties reaching up to 130% (Pfeifer et al., 2016), primarily due to inefficient particle detection at concentrations approaching 1 cm-3. Throughout the cruise, continuous 5-second sampling yielded 64,180 valid samples, which through statistical averaging reduced measurement uncertainty to 0.5%. At this negligible level, the distribution characteristics and cross-regional correlations are considered reliably preserved..*

RC2.23: Line 309: "After the ship entered the pelagic area, the influence of air mass transport almost disappeared" I don't think this statement is true. At 50 km away from the shore and under a moderate wind speed of 6 m/s, it takes about 2.5hr for air originating over the continents to reach the ship's location which is significantly shorter than the time it takes to remove anthropogenic aerosol by wet and dry deposition. This sentence should be removed.

Response: Thank you for your suggestion. We concur that the claim "the influence of air mass transport has almost disappeared" is inaccurate and conveys a misleading implication. As clarified in our response to RC2.3, the backward trajectory results consistently showed that the air masses did not pass through any continental areas before reaching the sampling site in the pelagic regions. Thus, the phrase has been

revised to "continental air mass transport" to precisely reflect the findings and context of the study.

*After the ship entered the pelagic area, the influence of continental air mass transport almost disappeared.*

RC2.24: Line 323: "Therefore, in the pelagic environments, the marine aerosol was not significantly affected by aerosol transport and anthropogenic activity" Same comment as above. The authors do not have detailed in-situ chemical speciation measurements to back these claims

Response: This sentence has been removed as suggested.

RC2.25: Line 409: remove the word "obvious" as some correlations are not that obvious (see next comment)

Response: The "obvious" has been removed as suggested.

RC2.26: Line 415: "Therefore, under the WS increased accompanied by synergistic influences of the gas-to-particle conversion and sea surface wind physical friction, the NCs increased in the pelagic region." I don't understand this sentence. Suggest the authors re-write it

Response: We have revised this sentence.

*Therefore, increased WS both intensified bubble rupture by enhancing sea surface friction and promoted air-sea gas transfer (Jaeglé et al., 2011; Mårtensson et al., 2003). This increased activity elevated production of marine aerosols and natural marine precursors, ultimately raising the NCs in the pelagic region.*

RC2.27: Figure 11-15, I suggest the authors make the legend fonts bigger. These are hard to see. Also, a lot of information on these figures is redundant. I suggest the authors pick the most relevant figure and keep it in the main manuscript and move the remaining to the SI. This will make the manuscript more readable.

Response: Thanks for the constructive comment. The relevant figures have been amended following your recommendation.

**Answer to Reviewer 3**

Summary

RC3.1: This study reports on aerosol properties measured using a TSI aerodynamic particle sizer (APS) and their relationship to environmental parameters measured on a 1- month cruise in the South China Sea and from reanalysis data. The paper aims to characterize the differences in aerosol number concentrations and size distributions between coastal and open-ocean regions and the implications transport has on the variability of modal concentrations. They argue that areas more proximate to the coast have higher aerosol number concentrations. While offshore, wind speed, SST, and the

difference between SST and 2-m temperature (SST-T2m) influence aerosol concentration positively (wind speed through mechanical generation) and negatively (SST and SST-T2m through changes in buoyancy and bubble viscosity).

Many of the claims and arguments presented in this work are unsupported or they have been given with vague descriptions and unclear relationships to previous literature. Because this is an interesting dataset in an under-sampled region and some interesting results are obscured within this study, I feel that instead of a full rejection, this paper should undergo substantial major revision before being considered appropriate to be considered for publication. Below I have provided detailed list of major concerns and many technical comments, however many editorial issues persist throughout. I recommend the authors employ these corrections to improve this paper.

Response: We appreciate the reviewer for the thorough reading and thoughtful comments and suggestions. These comments and suggestions made our study more targeted, structured, and understandable, which greatly improve the quality of the manuscript.

**Major comments:**

RC3.2: The writing in many places is very hard to read due to grammatical errors and the use of frequent platitudes making the paper very difficult to follow. In some places I noted where these were and made suggestions, but the issues were far too numerous to point out each one. Many details are missing, and the discussion is vague without specific directed attention to the very detailed figures or putting the results in context with the broader literature context or studies within this region. The authors should make a concerted effort to carefully re-read the paper to ensure its clarity.

Response: We sincerely appreciate the reviewer's insightful comments on the readability, detail completeness, and discussion depth of our manuscript. These comments are crucial for enhancing the overall quality and academic rigor of our work. We have carefully addressed each concern through systematic revisions, and the specific measures are outlined as follows:

To improve readability, we have conducted a full-text review to correct grammatical issues (e.g., inconsistent tenses, improper use of articles, and disjointed sentence structures) and remove platitudes.

To address the "missing details" concern, we have added specific information and discussion to enhance the persuasiveness of our study.

To resolve the "vague discussion" issue, we have expanded the discussion section by integrating our findings with existing studies and regional characteristics of the SCS.

These revisions have significantly improved the clarity, completeness, and academic depth of our manuscript.

RC3.3: In many cases, there is an incorrect or superfluous use of the article "the" .

Response: Thank you for your careful review and valuable comment regarding the use of the article "the". We fully agree with your observation and have conducted a

systematic check and revision of the entire manuscript to address incorrect or superfluous use of "the". We believe these adjustments have significantly improved the grammatical accuracy and readability of the manuscript.

RC3.4: There was also the use of "meanwhile" and "on the other hand" when the authors were trying to further describe findings or procedures. In many, if not all, of the cases that either of these phrases were used they were unnecessary and confound clarity. I recommend the authors remove these phrases.

Response: Thanks very much for the comment on the clarity of logical connectors. We fully agree that the unnecessary use of transitional phrases such as "meanwhile" and "on the other hand" may disrupt the readability of the text and obscure the core logical flow of findings or procedures.

Following your suggestion, we have carefully reviewed the entire manuscript to identify all instances where "meanwhile" and "on the other hand" were used. For each case, we evaluated whether the phrase was essential for conveying logical relationships. Where the phrase did not contribute to clarifying the connection between sentences or paragraphs, we have removed it. We appreciate your attention to this detail, as it has helped improve the overall readability and rigor of the manuscript.

RC3.5: The authors should use words like "the "correlations can explain", rather than declarative statements on correlations and other relationship because many of their claims are based on mostly visual comparisons and not substantive correlations or quantitative analyses.

Response: Thank you sincerely for pointing out this important issue regarding the language used to describe correlations or quantitative analyses in our manuscript. Your comment has helped us recognize that our previous phrasing was overly definitive. In the original manuscript, a large number of discussions were based on mostly visual comparisons, without effectively leveraging the substantive correlations or quantitative analyses of the observational data.

In response to your comment, we have thoroughly revised the relevant sections in the manuscript. Specifically, we have supplemented substantive correlation and quantitative analyses (including calculations of correlation coefficients or significance levels, and data comparison), and have based our key discussions on these results. Correspondingly, we have revised all declarative statements about correlations and quantitative analyses to adopt more cautious phrasing such as "the correlations can explain", "suggests a potential relationship", or "differences can explain".

Furthermore, we have taken great care to contextualize our findings within existing studies. Rather than presenting these findings as definitive conclusions, we have framed our observations as extensions or complements to prior studies, highlighting consistencies and discrepancies while acknowledging potential limitations. This approach, we believe, better reflects the incremental nature of scientific inquiry and ensures that our claims are appropriately contextualized.

RC3.6: It is the opinion of this reviewer that the word "production" or "emission" be

used in place of "generation" when describing the marine sources of aerosol. In the way "generation" is used in this study, the authors are describing wind-generated aerosol production/emission, a more canonical use of the word. Either choice of terminology should be used consistently throughout the text.

Response: Thanks for this constructive suggestion, which helps standardize the terminology in our manuscript and align it with canonical academic expressions in the marine aerosol field.

We fully agree with your comment that "production" or "emission" is more appropriate than "generation" when describing the marine sources of aerosol. In the revised manuscript, we have uniformly replaced all instances of "generation" (when referring to marine aerosol sources) with "production" to ensure terminological consistency throughout the text.

RC3.7: Citations in the text and references list should be checked for correct formatting. In many cases the authors have only listed the first author of papers rather than conforming to the proper citation style of this journal. This should be corrected.

Response: Thank you for pointing out this formatting inconsistency, which is crucial for ensuring the manuscript adheres to the journal's academic standards. We sincerely apologize for the oversight in the initial submission where some in-text citations only listed the first author, failing to conform to the proper citation style of this journal.

To address this, we have thoroughly corrected all non-compliant citation content and conducted a meticulous check of the entire manuscript. After these revisions, we have verified that all citations in the text and references list now fully comply with the proper citation style of this journal. We appreciate your attention to this detail, which has helped enhance the professionalism and standardization of the manuscript.

RC3.8: Abstract: The text in the graphical abstract is very difficult to read because the resolution is very poor.

Response: Thanks very much for the valuable comment. We apologize for the inconvenience caused by the poor readability of the graphical abstract in the initial submission. To address this, we have improved the resolution of the graphical abstract in the revised manuscript and have also uploaded it separately as a high-resolution standalone file in the additional materials, ensuring that the graphical abstract is clear and legible.

We appreciate your careful review and hope that the revised graphical abstract meets the required standards.

RC3.9: Introduction: The introduction lacks a clear narrative of the research problem and does not provide proper context that motivates the research questions. The current state of the introduction makes it hard for the reader to follow the motivation of this work. The authors should make sure to highlight what relevant prior has been that either supports or motivates why this work has been done. There is very little attention given to introducing the region, the region's sources of aerosol, and how this work will effectively fill in the gaps based on substantive research questions.

Response: Thank you for this constructive comment, which has guided us to significantly improve the clarity and contextual depth of the Introduction. We fully agree that the original Introduction lacked a clear narrative of the research problem, sufficient background to motivate the study, and detailed context for the research region. These issues have now been addressed through a comprehensive rewrite of this chapter.

The specific details of the revisions can be found in the author's tracked changes.

RC3.10: Were the aerosols dried before being sampled? Are you sampling rain drops thus leading to some of the effects presented? Prior work has often shown (e.g. (Petters et al., 2006; Zheng et al., 2018)) precipitation acts as a large sink for accumulation and large accumulation-mode aerosol. I'm not sure I follow or buy the conclusions about wave droplet induced increases in accumulation-mode particles presented in this work based on the available observations.

Response: Thank you for the critical comment. All aerosol were passed through a drying system before being sampled. In addition, the sampling inlet was equipped with a rain shield to prevent raindrops or large hydrometeors from being collected. The increased aerosol NC during rainfall periods was not caused by the sampling of raindrops.

We fully acknowledge prior studies (Petters et al., 2006; Zheng et al., 2018) that highlight precipitation as a major sink for accumulation mode particles. However, our cruise observational data still showed a significant increase in aerosol NC during rainfall periods after rain-shielding measures. We ruled out ship pollution by analyzing wind direction data, and thus we preliminarily speculate that rainfall caused the significant increase in aerosol NC. Notably, this phenomenon aligns with findings from previous studies (Bird et al., 2010; Zhou et al., 2020), which reported transient increases in aerosol NCs during rainfall periods. Joung & Buie (2014) indicate that the droplets can release aerosols when they influence porous surfaces, and these aerosols can deliver elements of the porous media to the environment. Hence, after accounting for the observation environment and rainfall intensity (short-duration heavy rainfall), we deduce that short-duration heavy rainfall resulted in numerous raindrops impacting the ocean and ship surfaces, generating aerosol particles. Subsequently, APS captured some of these aerosol particles, ultimately contributing to the increased aerosol particle concentration, which can also be observed in Fig. 3 (the blue-shaded region). Consolidated evidence from prior research and current findings suggests that short-duration heavy rainfall may lead to a transient increase in accumulation mode particles.

We do not deny the mechanism by which rainfall can cause wet removal or deposition of aerosols. What is mentioned in this paper may merely be a special phenomenon, and there is no conflict between the two.


[Figure]

Fig. 3 The time series of the shipboard observations in the SCS from 21 May to 3 June 2023. The blue-shaded regions represented periods affected by rain events. (a) Trend of the aerosol size distributions. (b) Trends of NCs of the two aerosol particle modes (black solid line represented the NC of the coarse mode, and red solid line represented the NC of the accumulation mode). (c) Trend of the WD. (d) Trends of the TOBS (dark orange solid line), T2m (light orange solid line), and SST (blue solid line). (e) Trends in the RH (gray solid line), the VIS (red solid line), and the rainfall intensity (dark blue solid line).

RC3.11: Do the authors use aerosol composition from MERRA-2 in a similar size range of the APS? Many of the MERRA-2 aerosol species will have some non-negligible contributions from particles <0.5 μm that cannot be explained with changes in the APS alone.

Response: Thanks very much for the insightful comment. We confirm that we used aerosol composition data from MERRA-2 within a size range comparable to the APS measurements, and we have added this clarification in the revised manuscript.

*sea salt (SEAS$_{10}$; Dp $\leqslant$ 10 μm), dust (DUST$_{10}$; Dp $\leqslant$ 10 μm)*

Many of the MERRA-2 aerosol species have some non-negligible contributions from particles <0.5 μm, and these contributions were not overlooked in our discussion of aerosol composition differences. While such contributions cannot be directly explained by changes in APS data (which focuses on $\geqslant$ 0.5 μm particles), we have provided preliminary inferences based on coastal vs. pelagic regional differences. For example, dimethyl sulfide (DMS) concentrations are higher in pelagic regions, likely due to greater phytoplankton biomass there, which emits more DMS. Increased DMS, can further lead to higher $SO_2$ through atmospheric chemical processes (e.g., oxidation reactions).

Additionally, it is important to clarify that MERRA-2 data was used as auxiliary data in this study, primarily to illustrate how continental transport drives differences in aerosol distributions between offshore and pelagic regions. These differences are ultimately partially reflected in aerosol size distributions (0.5 μm $\leqslant$ Dp $\leqslant$ 10 μm) and NCs (as measured by the APS).

RC3.12: Data screening: "Nucleation events" (Line 185 and in other passages) in the marine environment from new particle formation or otherwise are observable typically at sub-100nm sizes (in the nucleation, Aitken-mode range) and stop growing well below 500 nm. These are not measurable with an APS. How are the authors able to justify that such events can be adequately observed with their measurement limitations? I am not confident that a number concentration criteria for screening ship pollution or continental influence would be meaningful from an APS alone due to it measuring mostly coarse particles. Plots and discussion of such a justification should be provided so that the reader is confident that there is fidelity in such a screening. Further, new particle formation is not discussed as a potential effect or limitation on the findings of this work and this should be included.

Response: Thank you for the critical and valuable comment, which has helped us

correct misleading descriptions and improve the scientific rigor of our manuscript. We apologize for the confusion caused by our inaccurate references to "nucleation events". As you noted, nucleation events and new particle formation in marine environments typically occur in the sub-100 nm size (in the nucleation, Aitken-mode range) and stop growing well below 500 nm. Since the APS measurement is limited to the 500-30000 nm range, it cannot detect particles in these smaller size ranges. Additionally, no studies have explicitly demonstrated that newly formed particles grow rapidly enough to affect marine aerosols in the >500 nm range measurable by the APS. Thus, our earlier implication that nucleation events could influence the observational data (e.g. NCs) was unsupported and incorrect.

To address this, we have removed all references to nucleation events and new particle formation in the revised manuscript. The data screening described in this section is solely intended to further exclude potential ship emissions, complementing the wind direction-based screening:

Under the stable meteorological conditions, aerosol NCs should remain relatively constant (Hoppel, 1979, 1985; Russell et al., 1996). When continental transport present, the continental transport would cause sustained high NCs over several hours (Saha et al., 2022; Wang et al., 2020). Therefore, the sharp increasing NCs data (one order of magnitude higher than the average at that time) without corresponding meteorological parameter changes or evidence of continental transport are might attributed to local ship emissions (either from our ship or nearby ships) and thus excluded. The further data screening based on the unreasonable NCs may further exclude the influence of ship emissions.

We agree that discussing new particle formation and nucleation events are inappropriate here, as such processes lie outside the detection range of the APS and cannot be validated with our data. Including such discussions would introduce unsupported inferences, undermining the robustness of our findings.

We appreciate your constructive feedback again, which has helped clarify these methodological details and prevent reader confusion. The revisions strengthen the accuracy and reliability of our manuscript.


*ii) Further data screening with unreasonable NCs. Aerosol NCs remain relatively constant under stable meteorological conditions (Hoppel, 1979, 1985; Russell et al., 1996). In the presence of continental transport, sustained high NCs would persist for several hours (Saha et al., 2022; Wang et al., 2020). Therefore, we excluded the sharp decrease and increase in NCs data in the short term without changes in meteorological parameters and influences of continental transport, and all excluded data had NCs that were one order of magnitude higher or lower than the average NC at that time. So as to further screen out the possible influences produced by the ship emissions.*

**Technical/Editorial Comments:**

RC3.13: Abstract, Line 17: I believe "increase" would be a more appropriate word here than "elevation"; " …a 120% [increase] in offshore aerosol number concentrations …"
Response: Thanks very much for your suggestion. We have changed "elevation" to "increase".

RC3.14: Abstract, Lines 26-27: This sentence is written vaguely and should be improved for clarification. Are the authors saying that the results of this work provide evidence to support differences in the spatial variability of marine aerosol in the SCS, and further, these results can help improve understanding of production and transport? The authors should rework this sentence to ensure these points are coming across clearly.
Response: Thank you for your comments. We have written this sentence to enhance clarity.
The results of this study provide clear evidence for the impact of continental transport on the spatial variations of aerosols in the South China Sea (SCS). Meanwhile, the newly added SST-$T_{2m}$-NC relationship further improves the understanding of marine aerosol production. In summary, our results can improve understanding of marine aerosol transport and production.
The final revised abstract is as follows:
*Marine aerosols critically influence Earth's radiation budget and climate dynamics through their spatial distributions and components due to their production and transport processes. However, in-situ observational datasets remain limited, particularly in the South China Sea (SCS). Based on our comprehensive shipborne measurements, this study presents a quantitative analysis of marine aerosol distributions and compositional variations between offshore and pelagic regions over*

*the SCS. Our data demonstrate a 120% increase in offshore aerosol number concentrations (NCs, Dp < 10.37 μm) relative to pelagic baselines, featuring 120% higher accumulation-mode particles (Dp ≤ 1.981 μm) and 70% higher coarse-mode particles (1.981 μm < Dp < 10.37 μm), quantitatively confirming continental transport affects spatial distribution of marine aerosols. In contrast, in the pelagic regions, marine aerosols are virtually unaffected by continental source and distinctly represent characteristics of the local production. Meteorological analyses identified wind speed (WS) and sea surface temperature (SST) as primary regulators of NC. However, observed NC variations at fixed WS and SST values suggest additional controlling factors. We demonstrate that sea-air temperature differentials (SST-$T_{2m}$) exhibit a stronger correlation (r = -0.82, p < 0.01) with NC than the other meteorological parameters, where increased SST-$T_{2m}$ led to decreased marine aerosol production. This temperature gradient effect drives pronounced diurnal NC variations, with maximum differences of 35% observed between daytime, nighttime, and transition periods. These findings provide concrete evidence for the spatial and diurnal variability in marine aerosol distributions over the SCS, thereby further improving understanding of marine aerosol transport and production.*

RC3.15: Line 46: I would revise the latter part of this sentence as, " …and climate change, [there has been an increasing research focus on marine aerosols over the last] forty years."

Response: Thanks very much for your suggestion. We have revised this sentence.

*Due to their non-negligible influence on both radiation budget and climate change, there has been an increasing research focus on marine aerosols over the last forty years.*

RC3.16: Line 47: "NC" is not defined before its first use here.

Response: We have provided the definition for "NC" when it first appears in the manuscript.

*In addition to aerosol mass concentrations, researchers have also analyzed and reported on aerosol number concentrations (NCs);*

RC3.17: Lines 47-49: The summary of findings in the Hoppel (1979, 1985) studies presented here are insufficient and vague. What do the authors mean by "associated with changes in meteorological parameters and oceanic air mass"? What meteorological parameters?

Response: Please see response to RC1.3.

We appreciate the reviewer's request for clarification regarding Hoppel's work (Hoppel, 1979; 1985). In these studies, Hoppel observed increased aerosol NC and the number size distribution with rising wind speeds off the U.S. East Coast. The term "significant changes in particle size distribution (PSD)" specifically refers to shifts toward higher concentrations across the size spectrum under high-wind conditions, compared to low-wind periods.

Meanwhile, we consider that this citation is inconsistent with the theme of this section - "Researchers' reports on aerosol mass concentration and number concentration". Therefore, we have removed this sentence and will cite it in the next section that discusses the influence of meteorological factors on aerosols.

RC3.18: Lines 49-57: This passage is not written clearly. The authors are attempting to provide a survey of mass concentration differences between different marine regions, but they are listed in a disjointed manner that makes it difficult for the reader to understand. I would recommend that this passage be shortened and combined in a way that illustrates the differences in reported aerosol mass in the different regions. Please also be cognizant of consistent unit usage, e.g. use ug m-3 as it is the most common of your reported masses.

Response: Thank you very much for your valuable comment on improving the clarity, structure, and unit consistency of the manuscript. We highly appreciate your suggestion to streamline the content and explicitly illustrate regional differences in aerosol mass concentrations, and we have revised the manuscript accordingly.

*Early observations by Prospero (1979) across multiple marine areas showed notable variations in marine aerosol concentrations,,ranging from 3.34 to 8.71 μg m⁻³. Subsequent reported measured data verify substantial regional marine aerosol concentration differences between different ocean areas. In polar regions, submicrometer aerosol (Dp ≤ 1000 nm) mass concentrations averaged 0.76 μg m⁻³ in the Arctic (Leck & Persson, 1996) versus 3.15 μg m⁻³ in the Antarctic (Savoie et al., 1993). In the Pacific Ocean, the PM₂.₅ (Dp ≤ 2500 nm) concentration averaged 12.3 ± 9.1 μg m⁻³ in the Western Pacific (Ma et al., 2022) versus 140 ± 48.1 μg m⁻³ in the Bohai Sea (Han et al., 2019). In the Indian Ocean, Pant et al. (2009) observed that the average micrometer aerosols (500 nm ≤ Dp ≤ 10000 nm) mass concentrations were 8.89 μg m⁻³.*

RC3.19: Lines 57: Revise "For the China waters, …" to "In marine regions off the coast of China, …"
Response: We have revised this sentence in accordance with your comment.

RC3.20: Lines 60-63: The authors should clarify what is meant by "discrepancies in the marine aerosol concentrations and size distributions." Do the authors mean "differences"? The ending of this sentence also seems like a bit of a tangent and is not explained, ("especially from 10degN-20degN …" Why is this included and what is its relevance to the rest of the passage?
Response: We sincerely appreciate this insightful comment. The "discrepancies" mentioned here refer to "differences".
To avoid misunderstandings among readers, we have removed "size distributions" and revised the expression "discrepancies in marine aerosol concentrations and size distributions" to "differences in marine aerosol mass concentrations and NCs".
The note on latitude at the end (, especially from 10° N-20° N) was indeed irrelevant to the main topic, which might have caused confusion for readers. Therefore, we have

also deleted this sentence.

The final version is as follows:

*In summary, there are differences in marine aerosol mass concentrations and NCs between the different ocean areas; however, few studies exist on marine aerosol concentrations in the SCS (Kong et al., 2016; Su et al., 2022).*

RC3.21: Lines 63-65: I think a more substantive take away from this paragraph is that shipboard measurements provide better spatial (and temporal) context for aerosol measurements in diverse marine regions such as off the coast of China and they can further help improve characterizations by being updated. Can the authors please revise the sentence they have written in these Lines to better convey that?

Response: Thank you for this constructive suggestion. We agree that emphasizing the value of shipboard measurements in enhancing aerosol characterizations is a more substantive takeaway. As recommended, we have revised the final sentences to highlight that shipboard observations can provide better spatial and temporal context (particularly in diverse marine regions like off the coast of China) and that expanding and updating such measurements would help improve aerosol characterizations, especially for the SCS where current data are sparse and outdated. This revision can provide a more substantive insight.

*Given that shipboard measurements can provide better spatial and temporal context for marine aerosol measurements across diverse ocean areas such as offshore China, expanding and updating such shipboard observations have the potential to improve the characterization of marine aerosol in these regions.*

RC3.22: Line 66: The sentence is written in a way that conveys a finding. If so, the authors need to provide support. I believe the authors are actually saying, "Aerosol generation and transport [can lead] to differences in marine aerosol concentrations and size distributions." Is that correct?

Response: We sincerely appreciate this valuable comment. This is indeed the exact meaning we intended to convey. We have revised this sentence accordingly.

*Aerosol production and transport can lead to the differences in marine aerosol concentration and size distribution.*

RC3.23: Line 67: Clarify "aerosol components."

Response: We have clarified "aerosol components" in the revised manuscript.

*marine aerosol components (e.g. sea salt, dust, sulfate, organic carbon)*

RC3.24: Line 68: Are "weather events" synoptic weather patterns, mesoscale weather events, storms? Please clarify.

Response: We acknowledge the reviewer's comment. In the revised manuscript, we have changed "weather events" to "mesoscale weather events" and provided a clarification.

*mesoscale weather events (e.g. thunderstorm, sea breeze, typhoon)*

RC3.25: Line 71: The authors have written "et al." after "relative humidity (RH)." Do they mean "etc."? "et al." or "etc." is not appropriate here. If there are additional pertinent meteorological parameters to include, please list them.

Response: Thanks very much for the helpful comments. The use of "etc." or "et al." here is indeed inappropriate, so we have removed "et al."

*Furthermore, some key meteorological parameters of the air-sea interface could affect aerosol production and transport, such as wind direction (WD) and speed (WS), relative humidity (RH), and sea surface temperature (SST) (Carslaw et al., 2010; Hoppel, 1979, 1985).*

RC3.26: Lines 71-75: The description of Tang RH effects on marine aerosol and the overall implication from the studies in the following sentence are not connected through the same logic. The authors are correct to note that RH changes can affect aerosol size through deliquescence and efflorescence, however, this is not related to the "wet deposition and dispersion" mentioned later. The RH effects on size come into play when the aerosol are being sampled, so if mentioning this point is to say that the previous work was inconsistent with their sampling procedures (drying, heating) and that makes comparison and aerosol characterization difficult, then that is what should be discussed here. Because the authors are instead discussing effects on generation and transport evidence of that effect should be discussed and not the Tang result.

Response: We acknowledge the reviewer's constructive comment. To address the logical disconnect noted, we have removed the reference to Tang et al., (1997), as their findings on RH affecting aerosol size through deliquescence and efflorescence are not relevant to the discussion of RH impacts on aerosol generation and transport. Meanwhile, the original descriptions on the impact of RH on aerosols was not sufficiently focused on its effects on aerosol production and transport, and the citations were not entirely pertinent. We have revised the paragraph accordingly. The revised text now specifically emphasizes RH influences on particle dry deposition rates and secondary aerosol formation, with supporting references provided. The dry deposition rates are important to the aerosol transport, as higher dry deposition rates reduce the residence time of aerosols in the atmosphere and shortens their transport distance therein. Additionally, the formation of secondary aerosols (e.g., nitrate and sulfate) directly affects aerosol production. We believe the new descriptions on the role of RH in aerosol production and transport is now appropriate and convincing.

*Some studies revealed that rising RH increases particle dry deposition rates (Arimoto & Duce, 1986; Lo et al., 1999), which are important to aerosol transport, as higher dry deposition rates reduce the residence time of aerosols in the atmosphere and shorten their transport distance therein. Ding et al. (2021) found that elevated RH enhances secondary aerosol (e.g. nitrate and sulfate) formation, which directly affects aerosol production. Therefore, RH also affects aerosol transport and production.*

RC3.27: The reference for Tang et al. (1997) is not properly cited in the main text or reference list. In the reference list, the authors have only provided the name of the first author.

Response: We sincerely apologize for this mistake in our manuscript. To address this issue, we have carefully corrected this error in both the main text and the reference list in the revised version.

RC3.28: Lines 75-78: The authors should specify what the relationships are between aerosol generation and wind speed from the studies cited. Is it the total aerosol concentration, are they size dependent?

Response: Thanks very much for the comments. These cited studies derived the source functions by analyzing the relationship between aerosol particle size distribution and WS, and finally obtained aerosol concentration through simulation based on this source function. We have rewritten this paragraph.

*Some subsequent studies attempted to link NCs to observed WS (Andreas, 1998, 2010; Gong, 2003; Ovadnevaite et al., 2014; Smith et al., 1993; Yang et al., 2019). These studies derived source functions based on the relationship between aerosol particle size distribution and WS, thereby enabling the simulation of number size distribution and total aerosol NCs.*

RC3.29: Line 79: "[They] explained that the SST …" Please clarify what or who is meant by "They."

Response: We sincerely appreciate your detailed question. Our previous summary of SST effect on marine aerosol was insufficient and superficial; therefore, we have rewritten these sentences. In the revised manuscript, we provide an adequate summary of SST effect on marine aerosols and their production.

*In addition, SST dramatically influences the production of marine aerosols by affecting bubble bursting time and jet drop production efficiency (Zábori et al., 2012a). Jaeglé et al. (2011) and Mårtensson et al. (2003) further revealed that warmer SST might reduce seawater density and surface tension, ultimately leading to higher marine aerosol production. The reduced surface tension increases wave breaking efficiency, entraining more air into seawater to form bubbles. In addition, the reduced seawater density leads to more bubbles rising back to the sea surface. As these bubbles reach the surface and burst, they subsequently form marine aerosols.*

RC3.30: Lines 79-81: This is an inadequate summary of SST effects on aerosol properties. Please be specific based on the studies the authors have cited.

Response: Please see response to RC3.28.

RC3.31: The authors have made no mention of aerosol generation in the free troposphere (e.g. from new particle formation). This can have important implications for aerosol properties measured in the marine boundary layer and cannot be ignored in the discussion of important generation and transport drivers. Further, do they think, based on previous studies in this region, that new particle formation has any effect on the observed relationships?

Response: Thank you for highlighting this important point. We acknowledge that new particle formation (NPF) in the marine boundary layer is important for aerosol

production and transport. However, as you pointed out, the size range (nanometer scale) where new particle formation and growth processes occur is far smaller than the measurement range of the APS (0.5-30 μm). Meanwhile, no study clearly demonstrates that new particles exhibit a sufficiently high growth rate to exert a significant influence on accumulation mode aerosols. Thus, NPF processes do not directly affect the aerosol NCs or size distributions reported here. It also does not influence the specific relationships we observed between NCs and meteorological factors (e.g., wind speed, SST-T2m) in the measured size range (0.5-10 μm) for this study.

In addition, due to the size range limitations of the APS measurement instrument, we were unable to capture new particle formation events. If we had specifically mentioned the impact of new particle formation on aerosol production and transport in the Introduction, it might have caused misunderstanding and confusion among readers.

RC3.32: Line 66-88: Although I have provided rather detailed guidance on this entire paragraph, I believe the authors should completely rework this section. Very vague statements and disjointed thoughts persist throughout. The authors cite lots of work but don't use any of these citations for clear contextual support.

Response: Thanks very much for your valuable guidance on this paragraph. We have fully reworked this paragraph to address the issues of vague statements, disjointed thoughts, and insufficient contextual support from citations:

We clarified all the ambiguous expressions such as "aerosol components" and "weather events". We also restructured the logical framework of this paragraph to form a coherent and progressive structure: "Aerosol production and transport can lead to differences in aerosols → influences of 'distance and meteorological events' → the specific impacts of aerosol transport → the specific impacts of aerosol generation (caused by meteorological factors such as WS, RH, and SST) → potential effects of SST-T2m" → research on diurnal variations (which may more clearly demonstrate the differences between aerosol transport and generation). In addition, we closely tied each citation to specific viewpoints to enhance contextual support.

[revised manuscript text omitted]

RC3.33: Line 89: Please revise, " …most [observational] data …"
Response: Thank you for the constructive suggestions. We have corrected "observation data" to "observational data" and systematically reviewed and revised the relevant expressions throughout the text.

RC3.34: Line 89-96: I believe the authors should remove this passage up to the start of the motivation sentence, "To address these, …" Everything prior is not needed and is discussed in the previous paragraphs.
Response: The previous content was indeed redundant and unnecessary, so we have removed these sentences in the revised manuscript.

RC3.35: Line 102: Remove the semicolon after "respectively." This should begin a new sentence.
Response: Thank you for the comments. We have removed the semicolon after "respectively".
*According to these analyses, the specific relationships between the different meteorological parameters and marine aerosols were examined respectively. Finally, the overall results of marine aerosol particle size distributions and NCs in the SCS and the possible influence factors were given.*

RC3.36: APS data: do the authors use all of the channels in 0.5-10 μm diameter range? I believe previous work has shown that the first channel in the APS has issues with counting efficiency and sizing accuracy. Can the authors please provide clarification on if this channel was used and justification for why it is appropriate to use here?
Response: Thanks for your question. In our study, we utilized all APS channels within the 0.5–10 μm diameter range to characterize aerosol size distributions. Regarding concerns about the first channel's counting efficiency and sizing accuracy, we have addressed this through a 15-day field inter-comparison experiment conducted recently, specifically designed to validate APS performance (especially for the first channel) against other aerosol instruments.
To verify the APS measurements, we performed a field inter-comparison experiment from 2 October to 17 October 2025 at a decommissioned wharf in Zhuhai, Guangdong Province, China (22°12′ N, 113°37′ E). This site is remote from industrial emissions and roads, with an unobstructed 180° view of the northern SCS, to ensure marine aerosol measurements are achievable. Three instruments were deployed side-by-side in an open environment with 10-cm long tubes. These tubes were fixed to the railing at 30° to the horizontal and faced the sea surface (to simulate the previous observation scenario and minimize terrestrial interference). The three instruments

comprised: a Model 3321 APS (measuring 0.5-20 µm) spectrometer (TSI Incorporated, USA), a Portable Optical Particle Spectrometer (POPS) (measuring 0.115-3.37 µm; Handix Scientific, USA), and a Model 11-D Portable Aerosol Spectrometer (measuring 0.25-30 µm; GRIMM, Germany). The aerosol data resolution was set to 10 min in this inter-comparison experiment. Key results included:
1. Consistency in number size distributions: The aerosol number size distributions measured by all three instruments showed high similarity (Fig. S2a) and low discrepancy, confirming general agreement in capturing particle size trends.
4. Concentration comparison for sub-2 µm particles: Since direct channel-to-channel matching was not feasible due to the differing size bins, we compared total concentrations within overlapping ranges relevant to our study's accumulation mode: 0.5–1.981 µm for APS, 0.475-1.99 µm for POPS, and 0.488-2.14 µm for GRIIM 11-D. All three instruments exhibited consistent diurnal trends (Fig. S2b). Strong correlations between APS and the other instruments (R = 0.92 vs. GRIIM 11-D; R = 0.94 vs. POPS) further validate the APS's accuracy (Figs. S3a, S3b).
Based on the accurate sub-2 µm aerosol concentration data and particle size distribution results, we can infer that the measurement results for the first particle channel of the APS (included within the sub-2 µm range) are accurate and reliable.

[Figure]

Fig. S2 (a) The NCs of average size distributions for different aerosol measurement instruments (black solid line represented the APS data, orange solid line represented the 11-D data, and red solid line represented the POPS data). (b) Trends of NCs for different aerosol measurement instruments.

[Figure]

[Figure]

Fig. S3 The scatter plots of (a) NCs of 11-D data and APS data, (b) NCs of POPS data and APS data.

RC3.37: Line 119: delete "future."
Response: We have deleted "future".

RC3.38: Reanalysis data: For the atmospheric dynamic/thermodynamic properties (temperature, wind speed, etc…) was the ERA5 reanalysis, MERRA-2, or a combination of ERA5 and MERRA-2 used? The authors discuss ERA5 in the first part of Section 2.3.1., but then say, "Meanwhile, the MERRA-2 was …" Why were two different datasets used for these variables? Why not use only MERRA-2 or ERA5? What motivated using two different datasets? I understand using MERRA- 2 for the aerosol mass concentrations. Additionally, the authors should justify why they believe the coarse resolution of these datasets compared to the in situ cruise data are representative of the conditions measured where the ship is. Also, please provide a citation for MERRA-2 and spell out its acronym on the first use.

Response: Thanks very much for the comment. Initially, we planned to use the MERRA-2 dataset for both meteorological and aerosol parameters. However, Li et al. (2025) and Luo et al. (2020) evaluated the reliability of MERRA-2 and ERA5 reanalysis data in marine regions using satellite and in-situ observations. These studies found that ERA5 and MERRA-2 datasets show comparable precision for SST and $T_{2m}$ (with MERRA-2 having a slightly smaller bias), but ERA5 datasets exhibited higher accuracy than MERRA-2 datasets for WS (with ERA5 having a smaller bias). The both datasets demonstrated strong correlation with in-situ observations (R > 0.9) for atmospheric dynamic and thermodynamic properties. Notably, unlike urban or terrestrial areas, atmospheric dynamic and thermodynamic properties in open marine areas do not undergo abrupt changes. Thus, despite their coarser resolution relative to in situ cruise data, these reanalysis datasets are still deemed representative of the conditions measured at the ship's location. To sum up, we ultimately selected ERA5 for dynamic properties (WS and WD) and MERRA-2 for thermodynamic properties (SST and $T_{2m}$) as the auxiliary data.

In the revised manuscript, we have spelled out MERRA-2 as "Modern-Era

Retrospective Analysis for Research and Applications, Version 2" on the first use, and added relevant citations for MERRA-2 (Gelaro et al., 2017; Randles et al., 2017).


RC3.39: Line 203: The authors need to define the size ranges used for their quantification of accumulation and coarse modes.

Response: We have defined the size range for our quantification of accumulation and coarse modes.

*the accumulation (0.5 μm ≤ Dp ≤ 1.981 μm) and coarse (1.981 μm ≤ Dp ≤ 10 μm) mode particle NCs.*

RC3.40: Lines 202-203: I don't believe that it is appropriate for the authors to define aerosol number concentration integrated from the APS as the "total marine aerosol" or even the "total aerosol." A large portion of marine aerosol number concentrations come from substantial sub-500 nm particle contribution. As such, it would be much better suited if the authors revise this terminology; e.g. "APS integrated NC", "summed NC" .

Response: Thanks for the reviewer's constructive comments. It is inappropriate to define the aerosol number concentration obtained by integrating the APS data as "total marine aerosols" or even "total aerosols". To avoid the misunderstanding, we have revised all such expressions to "summed NC".

RC3.41: Table 2: define the size range for "accumulation mode." Clarify in the

caption that these are shipboard measurements or please specify the observational platform.

Response: We have added the particle size range of the "accumulation mode" in Table 2 and clarified in the table title that these data were all obtained from shipboard observations.

**Table 2**

*Summary of the available study results on the shipboard observation of marine aerosol NC (cm$^{-3}$)*

| Region | Time | Season | Latitude | Longitude | Parameter | Value | Parameter | Value | Reference |
|---|---|---|---|---|---|---|---|---|---|
| South China Sea | 2023.05 - 2023.06 | Spring | 21°N - 8°N | 115°E - 110°E | Accumulation mode ($n_{500-2000}$) | $52.4 \pm 35.0$ | $n_{500-10000}$ | $54.0 \pm 35.3$ | This Study |
| South China Sea | 2018.08 | Summer | 23°N - 19°N | 118°E - 108°E | $n_{400-32000}$ | 61 | | | Cai et al., 2020 |
| South China Sea | 2012.09 - 2012.10 | Autumn | 21°N - 20°N | 118°E - 113°E | $n_{120-10000}$ | 175 | | | Kong et al., 2016 |
| South China Sea | 2005.05 | Spring | 20°N - 18°N | 118°E - 113°E | Accumulation mode ($n_{50-2000}$) | $50.3 \pm 19.5$ | | | Lin et al., 2007 |
| East China Sea | 2005.05 | Spring | 30°N - 26°N | 122°E - 117°E | Accumulation mode ($n_{50-2000}$) | $109.2 \pm 51.8$ | | | Lin et al., 2007 |
| East China Sea | 2017.04 - 2017.05 | Winter | 28°N - 20°N | 130°E - 120°E | $n_{250-2500}$ | $57.4 \pm 40.9$ | $n_{2500-10000}$ | $57.5 \pm 41.3$ | Ma et al., 2022 |
| Western Pacific | 2017.04 - 2017.05 | Spring | 20°N - 0°N | 180°E - 130°E | $n_{100-19800}$ | $83 \pm 30$ | | | Flores et al., 2020 |

*Note.* In the column of the "Parameter", "n" indicated the NC and the subscripts indicated the particle size (nm); in the column of the "Latitude", "N" represented north latitude. The results of this study and these references were the overall average aerosol NCs.

RC3.42: Lines 206-207: The sentence "Due to the constraints …" is not necessary and should be removed.

Response: Thanks very much for the useful comment. We have removed this sentence.

RC3.43: Lines 207-209: Delete "data recorded and" in the sentence "The shipboard observation data …"

Response: Thanks very much for the useful comment. We have deleted "data recorded and".

RC3.44: Lines 211-214: I don't understand how the authors came to these conclusions or what evidence is being used to support these claims of new particle

formation being the cause of differences in accumulation-mode number concentrations. As I mentioned in one of my main comments, new particle formation and growth occurs at much smaller sizes than what is measured by the APS. The authors also do not provide any literature support for how they can argue this claim based on their measurements. Additionally, there is no discussion prior to or proceeding this sentence about westerlies and what that would mean for aerosol NC changes.

Response: Thank you for your constructive comments. As Reviewer 3 pointed out, the size range (nanometer scale) where new particle formation and growth processes occur is far smaller than the measurement range of the APS (0.5–30 μm). Therefore, the claim that new particle formation causes differences in accumulation mode (0.5–1.981 μm) NC is unreliable and cannot be supported by the measured data. No study clearly demonstrates that new particles exhibit a sufficiently high growth rate to exert a significant influence on accumulation mode aerosols. Nor is there any study clearly indicating that new particle formation is sparse in the SCS. Therefore, we have removed the claim that "new particle formation is the cause of differences in accumulation-mode number concentrations".

Instead, we have provided the reason for the difference in aerosol NCs between the East China Sea and the South China Sea: Aerosol emissions from the Yangtze River Delta region are higher than those from the Pearl River Delta region (Li et al., 2017). Due to the influence of aerosol transport, a greater amount of continental and anthropogenic aerosols from the Yangtze River Delta are delivered to the East China Sea compared to the amount transported from the Pearl River Delta to the South China Sea.

To improve accuracy, we have replaced the potentially misleading term "westerlies" with "aerosol transport."

*Aerosol emissions from the Yangtze River Delta region are higher than those from the Pearl River Delta region (Li et al., 2017). Due to the influence of aerosol transport, a greater amount of continental and anthropogenic aerosols from the Yangtze River Delta were delivered to the East China Sea compared to the amount transported from the Pearl River Delta to the South China Sea.*

RC3.45: Lines 213-216: I don't understand much of the discussion here or how it relates to the citation from the Atlantic. Did the Atlantic study measure the same size range? How can this study be used for comparison without mentioning these specific differences?

Response: Thank you for this insightful comment. We have reselected the measurement data from the Flores et al. (2020) experiment, which were collected in the Western Pacific. The South China Sea is one of the marginal seas of the Western Pacific; meanwhile, both our experiment and the Western Pacific experiment in Flores were conducted in spring and located in the tropical zone and westerlies. Therefore, we initially hypothesize that the aerosol number concentrations from the two measurements should be similar. We have revised this sentence.

*The SCS is one of the marginal seas of the Western Pacific. The summed NC observed in this study (54 cm⁻³) was slightly lower than NC in the Western Pacific (83 cm⁻³) by Flores et al. (2020).*

RC3.46: Line 248: replace "region" with "range" or "bin."
Response: Thanks for the comment. We have changed "region" to "range".

RC3.47: Figure 6: were these plots created using a fixed wind direction, relative humidity, precipitation, or other controlling factor? If not, how can the authors argue, especially based on the apparent low correlation and large scatter of the data, that wind generation is the primary mechanism for driving variability of this mode?
Response: These figures were created using wind direction (exclude wind directions ranging from 225° to 315°) and precipitation (rainfall intensity = 0) as control factors. The box plots in Figure 6 indicate a positive correlation between aerosol NC and WS. As WS increases, both the median and the 25th–75th percentiles of aerosol NC (represented by the box) increase. This correlation can be explained by the fact that wind generation drives variability of this mode. The remaining scatter of the data might caused by some particles forming through secondary processes.

[Figure]

Fig. 5 The scatter plots of (a) NCs of the aerosol accumulation mode and WS, (b) NCs of the aerosol coarse mode and WS. The observational data were binned to the WS intervals equal to 3 m s⁻¹; the boxes represented the 25th to 75th percentile value, the black whisker represented the 1.5 inter-quartile range, the black diamond marker represented the mean value, and the black horizontal line represented the median value in the box plots.

[Figure]

RC3.48: Figure 7a: What do the different colors of the boxplots represent? If they are offshore and pelagic, you should use the same color scheme throughout the whole figure.

Response: Thank you for this thoughtful comment. The different colors of the boxes represent the offshore and pelagic regions. In response to this suggestion, we have modified Figure 7 to ensure a consistent color scheme throughout.

[Figure]

Fig. 7 Classification of the shipboard observation path in the SCS: (a) Accumulation and coarse mode particle sizes graded NCs in the offshore and pelagic regions. For the box plots, the boxes represented the 25th to 75th percentile value, the black whisker represented the maximum and minimum range, the black triangle represented the 1.5 inter-quartile range, the black diamond marker represented the mean value, and the black horizontal line represented the median value. (b) The NCs of average size distributions (the solid lines and circles) and standard deviations (the shaded

areas) for marine aerosols of 0.5 to 10 μm diameters in the offshore and pelagic regions. (c) The daily average variations of the proportions and the NCs of two aerosol particle modes were shown with the distances from coast. (d) The distributions of marine aerosol components in the offshore and pelagic regions. The pie charts showed the average aerosol composition based on the mass concentrations from the Merra-2 aerosol dataset during the whole cruise period.

RC3.49: Figure 7a: are the differences in pelagic and offshore number concentrations statistically significant for each mode and their sum?
Response: Thank you very much for your valuable comment. We have rewritten this part and clarified that these differences are statistically significant with p < 0.001. Below, we will explain why this differences are statistically significant:
For example, the NC difference for the accumulation mode between the offshore and pelagic regions is 57.92 cm$^{-3}$, then we compare it to the standard error (the denominator of the t-value):

$$\text{Standard error (SE)} = \sqrt{\frac{S^2_{\text{offshore}}}{N_{\text{offshore}}} + \frac{S^2_{\text{pelagic}}}{N_{\text{pelagic}}}} \approx 1.88$$

where $S_{\text{offshore}}$ is the standard deviation of NC in the offshore regions, $N_{\text{offshore}}$ is the number of samples in the offshore regions, $S_{\text{pelagic}}$ is the standard deviation of NC in the pelagic regions, $N_{\text{pelagic}}$ is the number of samples in the pelagic regions.
The difference (57.92) is approximately 31 times larger than the SE (1.88). This large ratio directly leads to a high t-value (t ≈ 31), which far exceeds the critical threshold for p < 0.001 ( | t | >3.29). For the coarse and summed modes, they also both have high t-values (t ≈ 36 and 31 respectively). In addition, the large sample size effectively suppresses random fluctuations in the shipboard observation data, strengthening the statistical significance of the mean difference. Therefore, the differences are statistically significant (p < 0.001).

RC3.50: Lines 295-296: How does the bimodality of the distributions reported here compare to previous literature? Given the counting uncertainty in the lower bin of the APS, I'm not sure I believe true bimodality is being observed here, nor do I believe it will be comparable to prior reports of bimodal marine size distributions such as in Hoppel et al (1986).
Response: The bimodality of the distributions reported in this study is consistent with previous studies (Andronache, 2003; Braun et al., 2020). The number size distributions in the oceans exhibited a bimodal distribution, and the peak values both occurred at approximately 0.5 and 1.981 μm. Furthermore, the observed bimodality is reliable, as the accuracy of the APS data has been confirmed through recent multi-instrument comparative validation. In the original manuscript, we did not compare our findings with the marine aerosol bimodal distribution reported in the study by Hoppel et al (1986).

RC3.51: Line 297: Please clarify what is meant by the aerosols were "evenly

distributed in the 0.835 to 1.981 μm particle size range." Later in the text this term "evenly distributed" is mentioned again. It should be replaced with something more specific.

Response: Thank you very much for the comment. What we intend to express is that the dN/dlogDp values (representing the number size distribution) are relatively stable with no variations in the 0.835 to 1.981 μm particle size range. This sentence has been rewritten to convey this meaning clearly.

*The number size distributions exhibited a relatively stable value in the 0.835 to 1.981 μm particle size range.*

RC3.52: Line 298: Where do the authors describe a "transport effect" on the size distribution below 1.114 μm? Please clarify.

Response: Thanks very much for your suggestion. We have changed "transport effect" to "the influence of aerosol transport", and have also added discussion on "the influence of aerosol transport".

*Due to the influence of aerosol transport, the continental air masses carried continental and anthropogenic aerosols, which ultimately affected aerosol distributions in the 0.5-5.0 μm particle size range. The number size distributions in the offshore regions were obviously higher than in the pelagic regions in the 0.5-5.0 μm particle size range. The findings were consistent with the previous studies (Braun et al., 2020; Lorenzo et al., 2023).*

RC3.53: Line 301-305: Is this discussion only about the accumulation mode, coarse mode, or sum? Please specify.

Response: We have rewritten this section. In the revised manuscript, we have expanded the discussion to include accumulation mode, coarse mode, and the total NCs.

*Fig. 7c revealed a decreasing trend in NCs with increasing distance from the coast, and the correlation coefficients between the daily average NCs of accumulation mode, coarse mode, and the total NCs and the distance from the coast were calculated as R = -0.87, -0.67, and -0.81, respectively. The correlation analysis, based on hourly average NCs of accumulation and coarse modes versus the distance from the coast, yielded R = -0.59 and -0.50 for offshore regions, and R = -0.28 and -0.33 for pelagic regions. The same was true for the total NC; the correlation coefficient between the hourly average total NC and the distance was -0.56 in the offshore regions and -0.29 in the pelagic regions.*

RC3.54: Lines 303-305: Were only 2 data points used for the offshore correlation? In Figure 7c, there are only 2 dates and two bars pertaining to offshore. 2 data points are not sufficient for a correlation. Did the authors use all data points for those days or just the average in the bar charts? Please clarify (1) what was used for the correlation and (2) what data is being shown in figure 7c; the caption says "diurnal variations" which is very vague.

Response: Thank you for your constructive comments. We have specified the

variables used to calculate the correlation coefficients in the revised manuscript. The overall correlation coefficients were calculated using the daily average aerosol NCs and the distances from coast.

*Fig. 7c revealed a decreasing trend in NCs with increasing distance from the coast, and the correlation coefficients between the daily average NCs of accumulation mode, coarse mode, and the total NCs and the distance from the coast were calculated as R = -0.87, -0.67, and -0.81, respectively.*

As you pointing out, the number of data points is insufficient for a correlation analysis when using daily averages in offshore and pelagic regions. Therefore, we used the hourly average aerosol NCs and distances to calculate the correlation coefficients for these regions.

*The correlation analysis, based on hourly average NCs of accumulation and coarse modes versus the distance from the coast, yielded R = -0.59 and -0.50 for offshore regions, and R = -0.28 and -0.33 for pelagic regions. The same was true for the total NC; the correlation coefficient between the hourly average total NC and the distance was -0.56 in the offshore regions and -0.29 in the pelagic regions.*

Additionally, we have revised the caption of Figure 7c, changing "diurnal variation" to "daily average variation" for accuracy.

RC3.55: Line 310: "meteorological element distributions" is a very confusion description. This should be revised to "meteorological parameters" as in the table header.

Response: Thank you very much for the comment. We have changed "meteorological element" to "meteorological parameters".

RC3.56: Line 310: The authors say the meteorological parameters are "significantly different" between offshore and pelagic areas. Based on the means and standard deviations this does not appear to be the case as the absolute differences are within only a few percent between the areas. The authors should please explain this claim and provide statistical evidence to support it.

Response: Thanks very much for your comments. We apologize for the misunderstanding caused by our incorrect use of the term "significant differences". This was a wording error. What we intended to express was the discrepancies in meteorological parameters between offshore and pelagic regions. Therefore, we have revised "significant differences" to "discrepancies".

We appreciate you pointing out this imprecision, which has helped us improve the rigor of our manuscript's language.

*Table 3 revealed discrepancies in meteorological parameters between offshore regions and pelagic regions.*

RC3.57: Lines 312-313: " In addition to the WS influence, the frequency …" Where is this shown?

Response: Thank you for the comments. Through calculations, we found that the frequency of westerlies and southwest winds is basically the same in offshore and

pelagic regions, both around 80%. Therefore, the previous discussion is incorrect. Fig. 6 revealed that the distance from the coast is the main factor affecting continental aerosol transport, so we have rewritten this sentence to reflect that the difference in aerosol transport is caused by the distance from the coast.

*In addition, the offshore regions were relatively close to the coastline (Fig. 6b). Compared to pelagic regions, southwest and west winds in offshore regions could directly transport continental and anthropogenic aerosols to the ship's location from Guangdong and Hainan, China. Therefore, aerosol transport was higher in offshore regions.*

RC3.58: Lines 314-325: This passage and its discussion of effects on the aerosol is exceptionally inadequate. Significant jumps to conclusions are made throughout. (1) are the aerosols emitted from Guangzhou and Hainan and the "islands and countries surrounding SCS" expected to be observable in the size range of measurement of the APS? What evidence is there to support this? (2) " … underwent atmospheric transport, transformation, and deposition processes …" is very vague and not an appropriate claim based on the available measurements and analysis of this study. Please provide specific description of processes that the authors think the aerosol experienced that can explain the differences.

Response: Thank you for your constructive comments. We have rewritten this section. We have also adjusted the position of this paragraph (now placed at the beginning of Section 3.2) based on Reviewer 2's comments.

In offshore regions, aerosols emitted from Guangzhou, Hainan, and some surrounding island countries can be observed within the particle size range measured by the APS. We have provided relevant literature (Braun et al., 2020; Wu & Boor, 2021) to confirm that the transport of continental and anthropogenic aerosols has influences on NCs and number size distribution of aerosols larger than 0.5 μm.

*We conducted real-time analysis of the 72-hour backward trajectories of air masses at the ship's location (Fig. 6a, b). The backward trajectory analysis indicated that the air masses had last passed over continental areas on 22 May 2023, 11:00 local time (LT), at a point 50 km from the coast (the red solid lines in Fig. 6b). Consequently, for all sampling locations within this 50 km boundary, the air masses had directly passed over mainland areas. This meant they carried continental and anthropogenic aerosols that ultimately influenced the aerosol distributions (Braun et al., 2020; Wu & Boor, 2021).*

We have clarified the processes that aerosol particles undergo, which ultimately prevent them from affecting the NC and number size distribution of aerosols in pelagic regions. In addition, the revised paragraph focuses primarily on explaining how to distinguish between offshore and pelagic regions, and no longer discusses the observed differences caused by these processes.

*For regions more than 50 km from the coast, the backward trajectory results consistently showed that the air masses did not pass over any mainland areas before reaching the sampling site (the blue solid lines in Fig. 6a). The prevailing wind direction was primarily from the southwest (Fig. 3c) in these regions, so aerosols*

*could not be directly transported from the continent to the ship's location. Additionally, continental and anthropogenic aerosols, which were emitted from islands and countries surrounding the SCS, lost their original characteristics through the long-duration (over 72 hours) transport. These aerosols underwent atmospheric long-range transport, dry deposition, wet deposition, and aging processes. Such processes led to the removal of continental aerosols or their gradual dilution and mixing with natural aerosols (Hodshire et al., 2019;Ohata et al., 2016; Xu et al., 2021). Over time, the continental and anthropogenic aerosols transformed or integrated into the background aerosols. Hence, 50 km from the coast was taken as the boundary distance to distinguish offshore and pelagic regions in this study.*

RC3.59: Line 325-327: The authors need to justify that the dust and sulfate aerosol are representative of continental aerosol sources by providing citable studies.

Response: Thank you for this insightful comment. We have supplemented the relevant references in the revised manuscript to explicitly justify the that dust and sulfate aerosols are representative of continental aerosol sources.

*The higher concentrations of dust and sulfate aerosols further indicate that continental aerosols have influenced the aerosol components in the offshore regions (Geng et al., 2023; VanCuren, 2003).*


RC3.60: Lines 327-330: Have DMS, OC, and SO2 been shown to be in high concentrations in pelagic regions of the SCS? The "degree of [ …] marine biological activity" is alluded to in the following section (Lines 355), but nothing related to this is discussed and how it might explain the differences between pelagic and offshore regions.

Response: We appreciate this comment. The mass concentrations of DMS, OC, and $SO_2$ were higher in pelagic regions (0.18, 1.44, and 2.1 $\mu g\ m^{-3}$, respectively) than in offshore regions (0.1, 0.91, and 0.61 $\mu g\ m^{-3}$, respectively) of the SCS.

In the revised manuscript, we have further elaborated on the mechanism: pelagic marine biological activity (e.g., phytoplankton metabolism) directly drives these concentration differences, with supporting literature.

*Meanwhile, in the pelagic regions, the proportions of dimethyl sulfide (DMS), organic carbon (OC), and sulfur dioxide ($SO_2$) were 0.15 %, 1.2 %, and 1.7 %. These proportions were higher than those in the offshore regions (0.1 %, 0.84 %, and*

*0.56 %, respectively) due to the more frequent marine biological activities (e.g. phytoplankton metabolism) in the pelagic environments. For instance, phytoplankton releases DMS through cellular metabolism and lysis; DMS then undergoes atmospheric oxidation to form $SO_2$ (Kettle & Andreae, 2000). Additionally, phytoplankton also produces OC (O'Dowd et al., 2004). These marine biological activities directly contribute to higher proportions of DMS, $SO_2$, and OC in pelagic regions.*

RC3.61: Line 357: " …the meteorological parameters had obvious day-night differences." The word "obvious" should be removed here and replaced with "it is expected that there are diurnal differences." The differences are not "obvious" because they have not yet been shown.

Response: Thank you very much for the comment. This sentence has been rewritten as suggested.

*Beyond that, many meteorological parameters, such as WS, T2m, SST, and SST-$T_{2m}$, might influence the concentration and distribution of marine aerosols. It is expected that there are diurnal differences.*

RC3.62: Line 359-363: The threshold of 120 cm-3 is not comparable to the Saliba et al. (2019). In that study the condensation nuclei concentration (particles >10 nm) was used, while this study is using mostly large accumulation and coarse mode aerosol. Please clarify the discrepancy and justify this choice of threshold. Were other thresholds tested and what support is available to make this choice?

Response: Thanks for the constructive comment, which has helped us clarify difference between our threshold selection and Saliba et al. (2019).

Saliba et al. (2019) did not explicitly justify their choice of total particle number concentrations threshold was 2000 cm$^{-3.}$ The threshold for NCs they established was not intended to exclude the influence of continental transport, but rather to eliminate the impact of ship's own stack emissions. We had conducted data screening in Section 2.4, where observational data during periods of ship pollution have been removed. In our original manuscript, the statement that "NC screening was used to exclude continental transport" was incorrect, and we apologize for this misrepresentation.

Regarding the original threshold of 120 cm$^{-3}$, it was initially chosen based on statistical considerations (approximately the 99th percentile of a normal distribution of NCs in pelagic regions). We also tested other thresholds using 2σ and 3σ criteria. These tests showed only slight differences in NC values but no impact on the overall conclusions by using the different thresholds. The threshold of 120 cm$^{-3}$ lacks a clear physical mechanism justification. Hence, we have removed the NC threshold of 120 cm$^{-3}$, and the study (Saliba et al., 2019) have been properly cited in the revised manuscript. The 12 data points previously excluded (accounting for 0.4% of total observational data) have been reinstated. Reanalysis with these data included confirms that our core conclusions remain unchanged.

This revision eliminates readers' confusion regarding the selection of the threshold and ensures that the core conclusions of this study remain valid.

[Figure]

Fig. 8 (a) Diurnal variations of the total mean values of the NCs in the different aerosol particle modes. The vertical bars showed the standard errors (the shadow areas represented the transition periods between daytime and nighttime). (b) The NCs of average size distributions for marine aerosols of 0.5 to 10 μm diameters in different time periods. (c) The NCs of the different aerosol particle modes in different time periods. For the box plots, the boxes represented the 25th to 75th percentile value, the black whisker represented the maximum and minimum range, the black triangle represented the 1.5 inter-quartile range, the black diamond marker represented the mean value, and the black horizontal line represented the median value.

*In pelagic regions, the sources of 72-h backward trajectory air masses were from the ocean, and observational data were processed to exclude the continental influence. These aerosol data conformed to clean marine periods, which were proposed by Saliba (2019) to extract relatively clean marine aerosol data*

RC3.63: Lines 364-367: Please specify the hours used for each time.

Response: Thank you very much for the comment. This sentence has been rewritten as suggested.

RC3.64: Lines 367-383: I see no "clear diurnal variation" in Figure 9. Figure 9a shows a very minimal increase in mean accumulation mode aerosol and no change in coarse mode. The plots in 9c show basically similar medians with interquartile ranges that are nearly identical for each mode and their respective time periods. Have the authors tested if these differences are statistically significant? Again, differences in the mean concentrations here seem to vary by only 1-5%. What are the differences observed in the size distributions of Figure 9b? These are not discussed clearly in the text and as a I reader I see no real changes. These should be quantified as a change in peak diameter, width, number, etc.

Response: Thank you for this insightful comment. We acknowledge that the absolute numerical changes in Fig. 9a may appear small, but there is a gap of over 30 $cm^{-3}$ between the maximum and minimum values (with an average of only 49.22 $cm^{-3}$ in the pelagic regions). Moreover, there is an obvious shift in the trend of aerosol NCs during the day-to-night transition periods. For example, accumulation mode aerosol decreases continuously in night but increases continuously in the night-to-day transition (NDT) period. In summary, we consider there to be a clear diurnal variation. As for the accumulation mode in Figure 9c, the curves appear to show basically similar median distributions, with the interquartile ranges that are also almost identical. This is because the NC of local marine aerosols in the 0.5-1.981 μm range is inherently low (with an average of only 49.22 $cm^{-3}$ in the pelagic region), and we use a logarithmic axis. This can make the changes seem insignificant, but the mean concentrations at the bottom of Fig. 9c clearly show an increasing trend.

For accumulation and total modes, we found that the differences in different periods are statistically significant ($p < 0.01$) by using the calculation method described in Response 3.49. The statements has been added in the revised manuscript.

*The differences were all statistically significant ($p < 0.01$).*

We have quantified the changes in peak diameter and peak value for each periods, and have revised the text to explicitly discuss these quantifications and emphasize the variations.

*Comparisons of size distributions (Fig. 8b) showed that number size distributions exhibit a relatively stable value in 0.835-1.981 μm particle size range, and subtle differences emerged in this particle range. Quantitatively, peak diameter varied slightly across periods: 0.571 μm in nighttime, 0.567 μm in the NDT period, 0.569 μm in daytime, and 0.570 μm in the DNT period. More notably, the peak value was 147.05 $cm^{-3}$ in nighttime, then rose to 155.87 $cm^{-3}$ in NDT period, further increased to 165.60 $cm^{-3}$ in daytime, and reached the highest value of 206.79 $cm^{-3}$ in DNT period, registering a 0.4-fold increase relative to the nighttime baseline. The peak value showed a clear and continuous increasing trend, which may reveal variations in aerosol production. In addition, all size distributions for marine aerosols had the same shape. The consistent shape can be explained by their common marine origin and production mechanisms.*

RC3.65: Lines 428: the values are "more negative" not "smaller than" -0.75.

Response: Thanks for the comment. We have changed "smaller than" to "more negative".

RC3.66: Section 3.3.2. First Paragraph (SST influence): the authors should comment on the fact that the correlation found here for aerosol concentrations and SST occur for a very small range of SST of about 1-2 deg C. This is likely much smaller than the field and laboratory studies used for comparison. Do the authors think this has any effect on the observed correlations/slopes and the claims the authors make about entrainment and density changes that influence aerosol number concentrations? Terms like "daughter bubbles" are not described and make this discussion confusing. Please clarify this discussion for readability.

Response: Thank you for your constructive comments on Section 3.3.2. We have carefully addressed your concerns about the SST range, and clarification of "daughter bubbles".

On the small SST range (28 °C ≤ SST ≤ 31 °C) and its influence on correlations/slopes:

We acknowledge that the SST range in our study (28 °C ≤ SST ≤ 31 °C) is narrower than that in most field and laboratory studies for comparison. However, this narrow range is a reflection of the actual observational conditions in pelagic regions of the SCS during this cruise, where SST exhibits low spatial variability in the study period. Despite the small SST range, the negative correlations between SST and NCs remain significant (all R < −0.75), indicating that even subtle SST fluctuations can drive detectable changes in aerosol number concentrations (NCs) in this region. For the regression slopes, the narrow SST range may lead to a slightly underestimated magnitude of influence of SST (compared to studies with broader ranges), but it does not alter the direction of the trend (negative dependence) or the core claim: that SST regulates NCs via modifying near-surface air entrainment and bubble-related processes. This is because the physical mechanisms are still applicable to small SST variations. Even a small increase (1-2 °C) can reduce the near-surface air entrainment volume, thereby decreasing plunging jets and subsequent bubble formation. Therefore, we believe that the narrow temperature range does not affect the research conclusions of this paper.

Clarification of "daughter bubbles":

To improve readability, we have added a definition of "daughter bubbles" in the revised manuscript. These refer to small secondary bubbles generated at the edges of larger "central bubbles" when the latter rupture at the sea surface—a key process in marine aerosol formation, as daughter bubbles contribute to the production of submicron aerosols (Miguet et al., 2021; Sellegri et al., 2023).

*Meanwhile, the processes of the bubble rupture changed; the larger central bubbles (the primary bubbles rising to the sea surface) ruptured at the sea surface, small daughter bubbles (secondary bubbles with smaller diameters, generated at the edges*

*of central bubbles) were produced. Theses daughter bubbles are critical for formation of submicron marine aerosols (Miguet et al., 2021; Sellegri et al., 2023).*

RC3.67: Lines 432-433: Please clarify what is meant by "the influences of the SST on the NCs might be different in different seas due to the different components of the seawater."

Response: Thank you for your comment. We apologize for the ambiguity in the original manuscript. In this paragraph, we use "the different components of the seawater" to denote marine phytoplankton species.

Different phytoplankton species (e.g., diatoms vs. coccolithophores) have vastly different capacities for producing aerosol, and SST exerts distinct effects on the metabolic activity of different phytoplankton species (Lu et al., 2025; Yu et al., 2025). Therefore, the impact of SST on NC of phytoplankton-generated aerosols is different due to the regional differences in phytoplankton species and relative abundance.

However, our study does not include discussion or data on phytoplankton. Consequently, we have removed this particular phrasing to ensure the manuscript remains tightly focused on analyzing the effects of SST on marine aerosol NCs.

RC3.69: Section 3.3.2. Second Paragraph (SST-T2m): The authors spend quite a lot

of time making declarative statements about what's influencing the SST-T2m relationship to the aerosol concentration based on previous work. For such declarative statements, similar analysis exercises need to be carried out. They declare that SST-T2m was the "major determinant of atmospheric stability" which led to the "upward transport" of marine aerosol in the boundary layer. Other such declarative statements are made further in the paragraph, but no such results are shown. If the authors don't mean to declare such factors definitively describe their observations, they should be careful to instead place their findings in context with prior work rather than discuss with certainty.

Response: Thanks for your constructive comments. We fully acknowledge two key issues raised: our previous statements about SST-$T_{2m}$ were overly declarative, and we insufficiently distinguished "visual/comparative correlations" from "substantive quantitative analyses", which aligns with earlier comment (RC3.5) on avoiding definitive claims without robust quantitative support.

To address this, we have revised the relevant content to place our findings in the context of prior work more cautiously, rather than stating the influences with excessive certainty, and replaced absolute declarative language with cautious phrasing to reflect that our inferences are based on visual and comparative correlations (not fully validated quantitative mechanisms).

*Compared to the WS and SST, SST-$T_{2m}$ can better reflect the variations of the NCs (R > 0.90, Fig. 12). Meanwhile, the correlations can explain that NCs had a significant negative correlation with the SST-$T_{2m}$ (-1 °C ≤ SST ≤ 4 °C). Figs. 13 and 14 illustrated the NC of all aerosol particle modes versus SST-$T_{2m}$ respectively for WS and SST intervals, and further presented this negative correlation under controlled WS and SST intervals. Prior studies (Lewis et al., 2004; Yuan et al., 2019) had suggested that SST-$T_{2m}$ may be related to atmospheric stability and play a role in air convection, mechanical mixing over the ocean, and plume rise processes. As proposed in Song et al. (2023), SST-$T_{2m}$ could influence marine aerosol production by affecting atmospheric stability and thus the interfacial and effective production fluxes of marine aerosols by affecting the sea state, sea wave, and the process of the whitecap formation. Combining these previous inferences with our observational negative correlation between SST-$T_{2m}$ and NCs, it was plausible that SST-$T_{2m}$ could influence marine aerosol transport (e.g. potential upward transport driven by plume rise) and production. For example, increased SST-$T_{2m}$ may intensify plume rise, leading to reduced NCs near the sea surface. Additionally, increased SST-$T_{2m}$ might indirectly decrease aerosol production by altering atmospheric stability.*

RC3.70: Line 496: Did the authors use an anomaly for SST-T2m or is it the difference between SST and T2m. Please clarify.

Response: Thanks very much for your meticulous comments. We apologize for the misunderstanding caused by our incorrect use of "anomaly". We use the difference between SST and $T_{2m}$; therefore, we have revised this sentence accordingly.

*Notably, the SST-$T_{2m}$ exhibited the strongest correlation with NCs.*

RC3.71: Line 501-502: The authors mention "rapid solar radiation shifts" that drive changes in the aerosol concentrations. What is meant by this? Do they mean just day night differences? Please clarify as this is not discussed prior.

Response: Thank you very much for your valuable comment.

As the core medium linking diurnal variations to meteorological variations, the rapid solar radiation shifts caused by the day night differences, and ultimately result in obvious changes in meteorological parameters (e.g., WS, SST, and SST-T$_{2m}$). Such meteorological parameters changes further affect aerosol transport and production, ultimately triggering NCs fluctuations.

However, since the changes in solar radiation and the impact of solar radiation on meteorological factors have not been discussed in the previous text, it is inappropriate for us to replace "diurnal variations" with "rapid changes in solar radiation". The sudden mention of "rapid changes in solar radiation" would cause confusion among readers. Therefore, based on the discussions in the manuscript, we have rewritten the conclusions in this section, and the previously inappropriate mention "solar radiation" has been deleted. In the revised manuscript, we clearly state that during the sunrise and sunset, the rapid changes in meteorological parameters directly lead to NC fluctuations.

*WS, SST, and SST-T$_{2m}$ displayed distinct diurnal cycles, which may drive a distinct diurnal variation of NCs. Compared with the daytime, the combination of lower WS and higher SST and SST-T$_{2m}$ caused lower NCs in the nighttime. During sunrise and sunset, rapid variations in meteorological parameters triggered NC fluctuations. In the NDT transition (the transition period$_1$), stable WS left SST and SST-T$_{2m}$ as dominant NC regulators. In the DNT transition (the transition period$_2$), all aforementioned three factors jointly influenced NCs.*

---

## Referee Report (RR1)

**Second Review of Qiao et al. (2025)**

I thank the authors for their thorough responses to my and the other two reviewers' comments and the major revisions made to the text. I believe the paper has been much improved based on these suggestions and the incorporated changes. However, I still have a number of largely editorial and some technical comments and suggestions that I feel will strengthen the paper and overall improve its clarity before it is suitable for publication.

**General Comments**

- There are several places where the authors use the word "discrepancy" when they mean "differences." "Discrepancy" suggests compatibility while "difference" is a distinction between things. Please make sure that the right word is being used. I believe most times "discrepancy" is used in this paper it should be replaced with "difference", but please verify.

- The authors generally took my previous suggestion to remove the article "the" when it was superfluous, but in some places it was incorrected deleted, e.g. Section 3.3, Line 496: "Among them, [the] degree of impact…" The article is necessary here. Please verify that these are revised properly.

- I, again, encourage the authors to make a concerted effort to carefully re-read the paper in its entirety with revisions to ensure its clarity. There are several passages with missing words and punctuation errors that make the text a bit difficult to follow in some place.

**Response to Author Responses**

- RC3.38: I thank the authors for clarifying in their response why two different reanalysis datasets were used, however I believe this should also be briefly detailed in the main text; ERA5 for dynamical properties (because of better in situ agreement), MERRA2 for thermodynamics and composition.

- RC3.45: Please specify the range of summed NC in the Flores 2020 study for inclusion in the main text.

- RC3.56: these are not discrepancies they are differences

**Editorial Suggests and Technical Comments on the Revised Manuscript**

- Introduction, Line 58: "…between different areas." Replace "areas" with "regions." The same should be done for line 74.

- Introduction, Line 62: "…observed that the average micrometer aerosols…" I believe the phrasing here should not be "micrometer aerosols." Do the authors mean, "accumulation- to coarse-mode aerosols"? <1µm aerosol are not "micrometer aerosols" nor is this standard terminology.

- Introduction, Line 75: "…most available marine aerosol data…" Replace "data" with "measurements."

- Section 2.2.1, Line 162: "the" (and "diameter") should be included in the sentence, "…which has 52 size channels in [the] 0.5 to 20 µm [diameter] range."

- Section 2.2.1, Line 170: Replace "Thereby" with "Therefore"

- Section 2.3.1, Line 220: "For atmospheric aerosol [component data]…" The phrase "component data" is vague. This should be replaced with "composition."

- Section 2.3.1, Line 226: Here, and wherever appropriate, "discrepancy" should be replaced with "differences".

- Section 2.4, Line 264: Can the authors be a bit more specific when they say, "sharp decrease and increase in NCs"? Is this a visual inspection of NC changes or was a statistical method applied to identify these rapid changes in NC?

- Section 3.1, Lines 291-306: It is important for the authors to acknowledge that a potential leading cause of the differences in aerosol NC reported in this study as opposed to those compared to in literature (Table 2) are the differences in size range measured. This is not mentioned anywhere in these passages.

- Section 3.1, Line 343: Please revise, "The previous study proposed …" to "Previous work has proposed…"

- Table 3: I still don't believe it is appropriate to use the word "total" when discussing accumulation + coarse-mode aerosol. I recommend that the authors use "sum" here and elsewhere when the two modes are summed.

- Section 3.2, Lines 384-385: "These aerosols underwent atmospheric …" The authors should state that the aerosol "likely underwent dry deposition, wet deposition, and aging processes" associated with the "long-range transport." No evidence has been shown that each of these processes definitively happened. This is the same for the subsequent two sentences as well, which are worded a bit definitively without evidence.

- Section 3.2, Lines 400-403: The authors need to specify what statistical test was performed to determine statistical significance for the differences compared in these lines (and elsewhere mentioned).

- Section 3.3, Line 495: Here and elsewhere, please use "composition" instead of "components."

- Section 3.3, Line 510 (and elsewhere in this section): The authors have not revised the word "total" to "sum" as discussed in their response. Please make this revision. Figures 7-11 should also not say "total."

- Section 3.3.1, Line 570: How do the authors know that the "marine aerosols had a relatively short lifetime"? Where was this discussed or shown?

- Section 3.3.1, Lines 570-577: The language in these passages are still more declarative than what can be supported by the measurements. An example of where this can be improved is: "Under the influence of sea surface wind, ocean wave fluctuations and sea surface friction [MAY] increase with intensified wind stress." Word like "can", "may", "might suggest" would improve these passages. Things like "bubble rupture" and "wave fluctuations" were not reported in this study and are speculative based on the assumed relationship to the correlations, therefore should not be discussed definitively.

- Section 3.3.2, Line 594: The "accumulation mode was likely more sensitive to SST". Is this sensitivity implied because of the larger slope? Please clarify and state in the main text what is meant.

- Section 3.3.2, Line 594-597: "This observed trend was inconsistent…" The authors state that their trend is inconsistent with prior laboratory studies but "consistent

with the previous studies." Were these "previous studies" ambient measurements or also laboratory studies but using a different experimental set up? Please clarify and state the difference/similarity in the main text. The same is necessary for the following sentence which states, "a recent study." This description is vague and should state whether laboratory or ambient measurements.

- Section 3.3.2, Lines 604-613: This passage is still written confusingly and too conclusively. First, I don't understand why "meanwhile" is used at the beginning of the passage. If in relation to the previous sentence/discussion, do the authors mean to use a phrase like "accordingly"? I interpret the structure of this passage as: Near-surface air entrainment volumes and plunging jets were changed (prior sentence). [Accordingly], the bubble rupture changed (following sentence). This should be followed (as a new sentence) with "smaller daughter bubbles" were LIKELY produced BECAUSE larger drops ruptured; then, these smaller drops CAN produce submicron aerosol.

- Section 3.3.2, Line 608-610: Please revise (in brackets): "The [generation] of daughter bubbles [decreases] with an increasing ratio of seawater density to viscosity and a decreasing ratio of seawater viscosity to surface tension. [Therefore], under increasing SST, the ratio…"

- Section 3.3.2, Line 614: Delete "meanwhile."

- Section 3.3.2, Line 635: replace "meanwhile, they might" with "they might also"

- Section 3.3.2, Line 636-638: The sentences in these lines can be combined for clarity, "The difference in the SST-T2m might … during the experiment and should be considered further in subsequent targeted research."

- Conclusions, Lines 660-661: The authors should include the size ranges for their defined accumulation and coarse modes here.

- Conclusions, Lines 670,672 (elsewhere in this section): Please use "aerosol composition" instead of "aerosol components."

- Conclusions, Line 674: "…diminishing continental aerosol [contributions]." Do the authors mean concentrations? Please clarify and revise.

---

## Author Response (AR2)

**Summary**

We appreciate the valuable reviews and constructive feedback provided by the reviewers. We agree with the reviewers' suggestions and carefully revise the manuscript. Below are our point-to-point responses to the reviewers' comments and suggestions, with the reviewers' comments (RC) in black, our responses in red, and the revised manuscript content in italicized blue font.

**Minor revision-egusphere-2025-1463-referee-report-1**

**Second Review of Qiao et al. (2025)**

I thank the authors for their thorough responses to my and the other two reviewers' comments and the major revisions made to the text. I believe the paper has been much improved based on these suggestions and the incorporated changes. However, I still have a number of largely editorial and some technical comments and suggestions that I feel will strengthen the paper and overall improve its clarity before it is suitable for publication.

**General Comments:**

RC1.1: There are several places where the authors use the word "discrepancy" when they mean "differences." "Discrepancy" suggests compatibility while "difference" is a distinction between things. Please make sure that the right word is being used. I believe most times "discrepancy" is used in this paper it should be replaced with "difference", but please verify.

Response: Thank you for your meticulous suggestion. Following your suggestion, we have revised the entire manuscript, replacing "discrepancy" with "difference" where appropriate.

A representative revision example is provided below:

Section 2.3.1, Line 181: "We used above aerosol component data to discuss differences in aerosol distribution over the SCS."

RC1.2: The authors generally took my previous suggestion to remove the article "the" when it was superfluous, but in some places it was incorrected deleted, e.g. Section 3.3, Line 394: "Among them, [the] degree of impact …" The article is necessary here. Please verify that these are revised properly.

Response: Thank you for your meticulous comment regarding the appropriate use of the definite article "the". We have systematically reviewed the entire manuscript to verify all instances where "the" was previously adjusted, including the specific case noted (Section 3.3, Line 496: "Among them, [the] degree of impact …"). The missing "the" has been restored as it is grammatically necessary to modify the specific noun

phrase "degree of impact". All other cases of incorrect deletion of "the" have been corrected to ensure grammatical accuracy and consistency throughout the text. We greatly appreciate your attention to this detail, which enhances the rigor of the manuscript.

A representative revision example is provided below:

*Among them, the degree of impact on marine transport and background aerosols caused by continental transport and marine production differs greatly due to differences in the degree of continental transport and marine biological activities at different distances from the coast.*

RC1.3: I, again, encourage the authors to make a concerted effort to carefully re-read the paper in its entirety with revisions to ensure its clarity. There are several passages with missing words and punctuation errors that make the text a bit difficult to follow in some place.

Response: We sincerely appreciate the reviewer's valuable suggestion. We have carefully re-read the entire revised manuscript in a concerted effort to enhance its clarity. Specifically, we have thoroughly checked and corrected all identified missing words, punctuation errors, and other grammatical inconsistencies that may have hindered readability. We trust that these revisions have made the manuscript more coherent and accessible, and we apologize for any inconvenience caused by the initial oversights.

**Response to Author Responses:**

RC1.4: I thank the authors for clarifying in their response why two different reanalysis datasets were used, however I believe this should also be briefly detailed in the main text; ERA5 for dynamical properties (because of better in situ agreement), MERRA2 for thermodynamics and composition.

Response: Thank you for your valuable comment. We agree with your comment and have detailed the rationale for using two different reanalysis datasets in the main text: ERA5 was selected for dynamical properties (10-m wind speed, direction, and friction velocity) due to better in situ agreement, while MERRA-2 was chosen for thermodynamic variables (2-m temperature, sea surface temperature) and related variables (SST-T2m) given its excellent consistency with observational data in the South China Sea.

*The ERA5 hourly dataset used in this study was provided by the European Centre for Medium-Range Weather Forecasts (ECMWF) (Hersbach et al., 2023); ERA5 was selected for these dynamical properties due to better in situ agreement (Li et al., 2025).*

*The Modern-Era Retrospective Analysis for Research and Applications, Version 2 (MERRA-2) provides reanalyzed SST and T2m data that show excellent agreement with observational data in the SCS (r > 0.9) (Jiang et al., 2021). We selected SST and T2m data from the MERRA-2 meteorological dataset in this context (Gelaro et al., 2017).*

RC1.5: Please specify the range of summed NC in the Flores 2020 study for inclusion in the main text.

Response: Thank you for your valuable comment. We have revised the main text to explicitly specify the range of summed NC in Flores et al. (2020). We appreciate your attention to this detail, which improves the rigor of the manuscript.

*The average summed NC observed in this study (54 cm$^{-3}$, 0.5 μm ≤ Dp ≤ 1.98 μm) was slightly lower than the NC reported for the Western Pacific (83 cm$^{-3}$, 0.1 μm ≤ Dp ≤ 1.98 μm) by Flores et al. (2020).*

RC1.6: These are not discrepancies they are differences.

Response: Thank you for your meticulous comment. As noted in our response to RC1.1, we have changed "discrepancies" to "differences".

**Editorial Suggests and Technical Comments on the Revised Manuscript:**

RC1.7: Introduction, Line 58: " …between different areas." Replace "areas" with "regions." The same should be done for line 74.

Response: We have replaced "areas" with "regions".

RC1.8: Introduction, Line 62: " …observed that the average micrometer aerosols …" I believe the phrasing here should not be "micrometer aerosols." Do the authors mean, "accumulation- to coarse-mode aerosols"? <1μm aerosol are not "micrometer aerosols" nor is this standard terminology.

Response: Thanks very much for your insightful suggestion. We confirm that we intended to refer to "accumulation- to coarse-mode aerosols" here. To enhance terminological precision and conciseness, we have revised "micrometer aerosols" to "accumulation- to coarse-mode aerosols" in the revised manuscript, with the measured particle size range (500 nm ⩽ Dp ⩽ 10000 nm).

*In the Indian Ocean, Pant et al. (2009) observed that the average accumulation- to coarse-mode aerosols (500 nm ≤ Dp ≤ 10000 nm) mass concentrations were 8.89 μg m$^{-3}$.*

RC1.9: Introduction, Line 75: " …most available marine aerosol data …" Replace "data" with "measurements."

Response: We have replaced "data" with "measurements".

RC1.10: Section 2.2.1, Line 162: "the" (and "diameter") should be included in the sentence, " …which has 52 size channels in [the] 0.5 to 20 μm [diameter] range."

Response: We have added "the" and "diameter" to this sentence as suggested.

RC1.11: Section 2.2.1, Line 170: Replace "Thereby" with "Therefore"

Response: We have replaced "Thereby" with "Therefore".

RC1.12: Section 2.3.1, Line 220: "For atmospheric aerosol [component data]…" The phrase "component data" is vague. This should be replaced with "composition."
Response: Thanks very much for the suggestion. We have changed "component data" to "composition".

RC1.13: Section 2.3.1, Line 226: Here, and wherever appropriate, "discrepancy" should be replaced with "differences".
Response: Thank you for your constructive comments. As noted in our response to RC1.1, we have changed "discrepancies" to "differences".

RC1.14: Section 2.4, Line 264: Can the authors be a bit more specific when they say, "sharp decrease and increase in NCs"? Is this a visual inspection of NC changes or was a statistical method applied to identify these rapid changes in NC?
Response: Thank you for your thoughtful comment. To clarify, the "sharp decrease and increase in NCs" refers to statistically defined rapid changes (not visual inspection). We have added specific descriptions in the revised manuscript.
*Therefore, to further screen out the possible influence of ship emissions, we excluded data points where NCs exhibited a sharp short-term fluctuation (i.e. one order of magnitude higher or lower than the average NCs at that time) in the absence of changes in meteorological parameters and influences of continental transport.*

RC1.15: Section 3.1, Lines 291-306: It is important for the authors to acknowledge that a potential leading cause of the differences in aerosol NC reported in this study as opposed to those compared to in literature (Table 2) are the differences in size range measured. This is not mentioned anywhere in these passages.
Response: Thank you for your insightful comment. We fully acknowledge that differences in the measured particle size ranges are a potential leading cause of the differences in aerosol NCs between our study and the literature studies compared in Table 2. As reflected in the revised manuscript, we have explicitly added this point with specific details: we clarified that our study focused on particles with a size range of 0.5-1.98 μm, while Flores et al. (2020) included smaller particles (≥0.1 μm), and explicitly linked this size range difference to the observed NC variations. This revision enhances the completeness and rigor of our NC comparison discussion.
*The average summed NC observed in this study (54 cm$^{-3}$, 0.5 μm ≤ $D_p$ ≤ 1.98 μm) was slightly lower than the NC reported for the Western Pacific (83 cm$^{-3}$, 0.1 μm ≤ $D_p$ ≤ 1.98 μm) by Flores et al. (2020). Notably, this study focused on particles from 0.5 to 1.98 μm, while Flores et al. (2020) included smaller particles ($D_p$ ≥ 0.1 μm). Differences in the measured particle size ranges are a potential leading cause of the differences in marine aerosol NCs.*
*Although the differences in observation seasons and particle size ranges might influence the average NC observations,*

RC1.16: Section 3.1, Line 343: Please revise, "The previous study proposed …" to "Previous work has proposed …"

Response: Thank you for your comment. We have revised this sentence as suggested.

RC1.17: Table 3: I still don't believe it is appropriate to use the word "total" when discussing accumulation + coarse-mode aerosol. I recommend that the authors use "sum" here and elsewhere when the two modes are summed.

Response: Thanks for the reviewer's constructive comments. It is inappropriate to define the aerosol number concentration obtained by integrating the APS data as "total marine aerosols" or even "total aerosols". To avoid misunderstanding, we have used "sum" here and elsewhere when the two modes are summed.

Table 1 (c.f. Table 3 in the manuscript)
*Distributions of NCs for different aerosol particle modes in different ocean regions. Mean and SD, respectively, represent the mean values and standard deviations of the related meteorological parameters.*

| Observation Area | South China Sea | | | |
|---|---|---|---|---|
| Route Location | Offshore Region | | Pelagic Region | |
| Marine Aerosol Parameters | Mean | SD | Mean | SD |
| Accumulation Mode (cm$^{-3}$) | 105.57 | 25.52 | 47.65 | 31.63 |
| Coarse Mode (cm$^{-3}$) | 2.68 | 0.38 | 1.57 | 0.80 |
| Sum (cm$^{-3}$) | 108.25 | 25.43 | 49.22 | 31.97 |
| Accumulation Mode / Sum (%) | 97.52 | - | 96.81 | - |
| Coarse Mode / Sum (%) | 2.48 | - | 3.19 | - |
| Meteorological Parameters | | | | |
| WS (m s$^{-1}$) | 10.74 | 1.95 | 8.64 | 3.70 |
| RH (%) | 91.20 | 1.72 | 82.41 | 3.40 |
| $T_{OBS}$ (°C) | 28.19 | 0.57 | 29.18 | 0.87 |
| SST (°C) | 27.71 | 0.37 | 29.78 | 0.33 |

RC1.18: Section 3.2, Lines 384-385: "These aerosols underwent atmospheric …" The authors should state that the aerosol "likely underwent dry deposition, wet deposition, and aging processes" associated with the "long-range transport." No evidence has been shown that each of these processes definitively happened. This is the same for the subsequent two sentences as well, which are worded a bit definitively without evidence.

Response: Thank you for your thoughtful suggestion. We agree with your suggestion and have revised the manuscript to adopt a more tentative tone. Specifically, we added "likely" to indicate the inferential nature of the processes (dry deposition, wet deposition, aging, and transformation) for which direct evidence was not provided, aligning with the scientific rigor of the discussion.

*These aerosols likely underwent dry deposition, wet deposition, and aging processes associated with long-range transport. Such processes could have led to the removal of*

*continental aerosols or their gradual dilution and mixing with natural aerosols (Hodshire et al., 2019; Ohata et al., 2016; Xu et al., 2021). Over time, the continental and anthropogenic aerosols may have transformed into or been integrated with the background aerosols.*

RC1.19: Section 3.3, Line 495: Here and elsewhere, please use "composition" instead of "components."

Response: Thanks very much for the suggestion. We have changed "components" to "composition".

RC1.20: Section 3.3, Line 510 (and elsewhere in this section): The authors have not revised the word "total" to "sum" as discussed in their response. Please make this revision. Figures 7-11 should also not say "total."

Response: Thank you for your constructive suggestion. To avoid misunderstanding, we have revised all such expressions to "summed NC".

RC1.21: Section 3.3.1, Line 570: How do the authors know that the "marine aerosols had a relatively short lifetime"? Where was this discussed or shown?

Response: Thank you for your insightful comment. To address your concern, we have supplemented relevant literature to provide evidence for this statement. The revised manuscript clarifies that the inference that marine aerosols have a relatively short lifetime is based on existing atmospheric science studies, enhancing the rigor of the discussion.

*In the pelagic region, the NCs were strongly influenced by the local production and marine aerosols had a relatively short lifetime compared with continental aerosols (Liu et al., 2005; Qureshi et al., 2009).*

RC1.22: Section 3.3.1, Lines 570-577: The language in these passages are still more declarative than what can be supported by the measurements. An example of where this can be improved is: "Under the influence of sea surface wind, ocean wave fluctuations and sea surface friction [MAY] increase with intensified wind stress." Word like "can", "may", "might suggest" would improve these passages. Things like "bubble rupture" and "wave fluctuations" were not reported in this study and are speculative based on the assumed relationship to the correlations, therefore should not be discussed definitively.

Response: Thank you for this constructive comment. Your comment has helped us recognize that our previous phrasing was overly definitive. In response, we have revised the relevant passages by using speculative terms (e.g., "may", "might", "could") as suggested. The revised text is provided below:

*Under the influence of sea surface wind, ocean wave fluctuations and sea surface friction may increase with intensified wind stresses. Air bubbles generated and present on the sea surface might rupture to form numerous water droplets, which could eventually produce primary marine aerosol after evaporation and crystallization processes (Blanchard et al., 1980; Saliba et al., 2019). Therefore,*

*increased WS may both intensify bubble rupture by enhancing sea surface friction and promote air-sea gas transfer (Jaeglé et al., 2011; Mårtensson et al., 2003). These processes might elevate the production of marine aerosols and natural marine precursors, ultimately raising the NCs in the pelagic region.*

RC1.23: Section 3.3.2, Line 594: The "accumulation mode was likely more sensitive to SST" . Is this sensitivity implied because of the larger slope? Please clarify and state in the main text what is meant.

Response: We thank the reviewer for this insightful comment.

Yes, the statement that the accumulation mode is "more sensitive" to SST is indeed based on the comparison of the regression slopes presented in Fig. S4. In this context, a steeper (more negative) slope for the accumulation mode indicates that its number concentration changes more drastically per unit change in SST compared to the other modes (e.g., the coarse mode). This greater rate of change might be interpreted as higher sensitivity. We have revised the text in Section 3.3.2 to explicitly clarify this point, as shown below.

*The steeper negative slope observed for the accumulation mode, compared to the coarse mode, pointed to a greater responsiveness of its NC to changes in SST, suggesting that accumulation mode particles might be more sensitive to these variations.*

RC1.24: Section 3.3.2, Line 594-597: "This observed trend was inconsistent…" The authors state that their trend is inconsistent with prior laboratory studies but "consistent with the previous studies." Were these "previous studies" ambient measurements or also laboratory studies but using a different experimental set up? Please clarify and state the difference/similarity in the main text. The same is necessary for the following sentence which states, "a recent study." This description is vague and should state whether laboratory or ambient measurements.

Response: Thank you very much for your valuable comment. We agree with the reviewer that this distinction is crucial for clarity. We have revised the passages to explicitly state the nature (laboratory or field measurements) of all cited studies. The "previous studies" (Salter et al., 2014; Zábori et al., 2012b) and the "recent study" (Christiansen et al., 2019) referenced in the original submission were indeed laboratory studies. To avoid ambiguity, we have now grouped them together as "other laboratory studies" and clarified that their inconsistency with some laboratory findings (Keene et al., 2017; Forestieri et al., 2018) may stem from differences in experimental setups (e.g. plunging jet, water jet, or diffuser systems) or the water types used (e.g. natural, artificial, or synthetic seawater). Furthermore, we have specified that the study by Lehahn et al. (2014) was a field study based on shipborne measurements. These modifications are reflected in the updated manuscript.

*This observed negative correlation between SST and NCs was inconsistent with some laboratory studies (Keene et al., 2017; Forestieri et al., 2018) but consistent with other laboratory studies (Christiansen et al., 2019; Salter et al., 2014; Zábori et al., 2012b). These laboratory studies have shown disparate results, which may stem from*

*differences in experimental setups (e.g. plunging jet, water jet, or diffuser systems) or the water types used (e.g. natural, artificial, or synthetic seawater). A recent field study also reported decreasing NCs with rising SST based on shipborne measurements in the North Atlantic (Lehahn et al., 2014).*

RC1.25: Section 3.3.2, Lines 604-613: This passage is still written confusingly and too conclusively. First, I don't understand why "meanwhile" is used at the beginning of the passage. If in relation to the previous sentence/discussion, do the authors mean to use a phrase like "accordingly"? I interpret the structure of this passage as: Near-surface air entrainment volumes and plunging jets were changed (prior sentence). [Accordingly], the bubble rupture changed (following sentence). This should be followed (as a new sentence) with "smaller daughter bubbles" were LIKELY produced BECAUSE larger drops ruptured; then, these smaller drops CAN produce submicron aerosol.

Response: Thank you for your constructive suggestion, which has helped improve the clarity and rigor of our discussion. We sincerely apologize for the confusing structure and inappropriate use of "meanwhile" in the original passage, as well as the overly conclusive tone. Following your insightful guidance:

1. We replaced "meanwhile" with "accordingly" to clarify the causal link between changed near-surface air entrainment/plunging jets and altered bubble rupture.

2. We revised the tone to be more tentative by adding "likely" (as you recommended) to reflect the inferential nature of the relationship.

3. We revised the entire paragraph following your suggestion to address all of your concerns.

The revised passage in the manuscript reads as follows:

*Near-surface air entrainment volumes and plunging jets were changed. Accordingly, the bubble rupture changed. Small daughter bubbles (secondary bubbles with smaller diameters, generated at the edges of central bubbles) were likely produced because larger central bubbles (the primary bubbles rising to the sea surface) ruptured; then, these smaller bubbles could produce submicron aerosol. These daughter bubbles are critical for the formation of submicron marine aerosols (Miguet et al., 2021; Sellegri et al., 2023).*

RC1.26: Section 3.3.2, Line 608-610: Please revise (in brackets): "The [generation] of daughter bubbles [decreases] with an increasing ratio of seawater density to viscosity and a decreasing ratio of seawater viscosity to surface tension. [Therefore], under increasing SST, the ratio …"

Response: Thank you for your constructive suggestion. The sentences have been carefully modified as suggested.

RC1.27: Section 3.3.2, Line 614: Delete "meanwhile."

Response: Thank you for your meticulous comment. We have deleted "meanwhile".

RC1.28: Section 3.3.2, Line 635: replace "meanwhile, they might" with "they might

also"

Response: Thank you for your suggestion. The sentence has been carefully modified as suggested.

RC1.29: Section 3.3.2, Line 636-638: The sentences in these lines can be combined for clarity, "The difference in the SST-T2m might … during the experiment and should be considered further in subsequent targeted research."

Response: We appreciate the reviewer's suggestion. The sentence has been carefully modified as suggested.

RC1.30: Conclusions, Lines 660-661: The authors should include the size ranges for their defined accumulation and coarse modes here.

Response: Thank you for your meticulous comment. We have added the size ranges for accumulation and coarse modes in the revised manuscript.

RC1.31: Conclusions, Lines 670, 672 (elsewhere in this section): Please use "aerosol composition" instead of "aerosol components."

Response: Thanks very much for the suggestion. We have changed "aerosol components" to "aerosol composition".

RC1.32: Conclusions, Line 674: " …diminishing continental aerosol [contributions]." Do the authors mean concentrations? Please clarify and revise.

Response: Thank you for your meticulous comment. Indeed, in the context of describing the decreasing trend of aerosol number concentrations (NCs) with distance from the coast, the term "concentrations" is more accurate than "contributions". We have changed "contributions" to "concentrations" as suggested.

*Furthermore, NCs exhibited a negative correlation with distance from the coast, and this trend was consistent with diminishing continental aerosol concentrations.*

**Minor revision-egusphere-2025-1463-referee-report-2**

RC2.1. Graphical abstract (a): The authors said the right y-axis shows NC (Same as left y-axis). Ok sounds good. But what about the size distributions shown? Do they have any axis representation? Any diameter? Or is it more graphical only?

Response: We thank the reviewer for raising this point. Indeed, the original box plot in the graphical abstract did not explicitly present the aerosol size distributions. Following the reviewer's suggestion, we have now clearly labeled the specific particle diameter ranges on the box plot. Additionally, in the revised graphical abstract, we have added the average aerosol number size distributions to better illustrate the differences between the offshore and pelagic regions. We believe these improvements make the graphical abstract more effective in conveying key findings of our study in a visually engaging manner.

**Graphical Abstract**

[Figure]

RC2.2. Can you do a scatter plot between SST-T2m and Nc? And show the correlations?

Response: We thank the reviewer for this suggestion. As requested, we have provided the scatter plot between SST-$T_{2m}$ and summed NCs (Fig. 1) and shown their correlations.

[Figure]

Fig. 1 The scatter plots of summed NCs and SST-$T_{2m}$.

RC2.3. Whats the uncertainty in measuring SST and T2m? is the difference greater than uncertainty? Try the same with SST-T10m may be?

Response: Thank you for the insightful comment. According to previous studies and product documentation (Bosilovich et al., 2015; Gelaro et al., 2017; Molod et al., 2015), the measurement uncertainty is approximately ±0.2 ℃ for SST and ±1.0 ℃ for T2m. The mean difference (1.4 ℃) in our analysis is greater than these uncertainties. For SST-T10m, we note that the uncertainty for T10m is not explicitly documented in relevant literature or dataset descriptions. Therefore, we are unable to conduct a reliable uncertainty assessment for T10m and SST-T10m.

Reference:

Bosilovich, M. G., Akella, S., Coy, L., Cullather, R., Draper, C., Gelaro, R., Kovach, R., Liu, Q., Molod, A., Norris, P., Wargan, K., Chao, W., Reichle, R., Takacs, L., Vikhliaev, Y., Bloom, S., Collow, A., Firth, S., Labow, G., Partyka, G., Pawson, S., Reale, O., Schubert, S. D., and Suarez, M.: MERRA-2: Initial Evaluation of the Climate, Technical Report Series on Global Modeling and Data Assimilation, 43, doi:NASA/TM–2015-104606/Vol. 43, 2015.

Gelaro, R., McCarty, W., Suarez, M. J., Todling, R., Molod, A., Takacs, L., Randles, C. A., Darmenov, A., Bosilovich, M. G., Reichle, R., Wargan, K., Coy, L., Cullather, R., Draper, C., Akella, S., Buchard, V., Conaty, A., da Silva, A. M., Gu, W., Kim, G. K., Koster, R., Lucchesi, R., Merkova, D., Nielsen, J. E., Partyka, G., Pawson, S., Putman, W., Rienecker, M., Schubert, S. D., Sienkiewicz, M., and Zhao, B.: The Modern-Era Retrospective Analysis for Research and Applications, Version 2 (MERRA-2), J. Climate, 30, 5419–5454, https://doi.org/10.1175/jcli-d-16-0758.1, 2017.

Molod, A., Takacs, L., Suarez, M., and Bacmeister, J.: Development of the GEOS-5 atmospheric general circulation model: evolution from MERRA to MERRA2, Geosci. Model Dev., 8, 1339–1356, https://doi.org/10.5194/gmd-8-1339-2015, 2015.

RC2.4. If the instrument was at 10m height, then I think in the back-trajectory model, the altitude of 50m should be replaced with 10m. This is because you are measuring PNSDs at 10m height, if you want to make comments about these aerosols, then BJTs has to come down to this altitude.

Response: Thank you for this important correction. We agree with the reviewer's point. The backward trajectories have been recalculated at an altitude of 10 meters above ground level to better represent the air masses at the instrument height. The manuscript text and the corresponding figure (Fig. 2) have been updated accordingly.

*The trajectories were calculated at an altitude of 10 m above ground level to match the instrument sampling height*

[Figure]

*Fig. 2 (c.f. Fig. 6 in the manuscript) (a) The 72-h backward trajectory air mass source traces in the offshore (red solid lines) and pelagic (blue solid lines) regions. The light purple solid lines represented the ship track. (b) Detailed map of the backward trajectory air mass source traces passing through the mainland areas (© Google Earth). The white arrows represented the direction of air mass transport.*

RC2.5. Section 3.1 line 229: "for these three aerosol modes" ! well total aerosol, accumulation, coarse aren't modes. I agree that accumulation and coarse are modes but total isn't a mode. So don't use the word modes here

Response: Thank you for your comment. We apologize for the inaccurate terminology. We have revised the sentence structure in Line 229 to avoid referring to the total aerosol as a "mode". The revised passage in the manuscript reads as follows:

*During the shipboard observation period, the summed (Dp < 10.37 μm) NC varied from 18.46 to 89.38 cm⁻³, NC of accumulation mode varied from 17.39 to 87.31 cm⁻³, and NC of coarse mode varied from 0.83 to 2.49 cm⁻³. The NCs exhibited substantial temporal fluctuations. The average values for summed NC, accumulation mode NC, and coarse mode NC were 54.01 cm⁻³, 52.35 cm⁻³, and 1.66 cm⁻³, respectively.*

RC2.6. Can you show the mode fittings of accumulation and coarse mode? The 1.981 is still not convincing to be classified as accumulation mode. You see peaks at both 0.542 and 1.981um. So these two can be Dg of the modes. IF 1.981 is Dg of the coarse mode then there has to be some range of dia below 1.981 which falls in the coarse mode as well. I would like to see the mode parameters.

Response: Thanks very much for the valuable and constructive comments. As requested, we have supplemented the fitting parameters (Table 2) and specific size distributions (Fig. 3) of the accumulation and coarse modes.

As you mentioned, the measured distribution exhibits distinct peaks around 0.542 μm and 1.981 μm.

However, Fig. 3 shows that the directly observed bimodal peaks are actually formed

by the superposition of three submodes. These three log-normal modes accurately reflect the marine aerosol particle size distribution in the SCS. The specific fitted peak diameters (Dg) and geometric standard deviations (GSD) are listed in Table 2:

Table 2
Summary of the fitting parameters for the accumulation and coarse modes.

| Mode | Dg (μm) | GSD |
|---|---|---|
| Accumulation Mode 1 | 0.567 | 1.076 |
| Accumulation Mode 2 | 0.440 | 1.873 |
| Coarse Mode | 2.309 | 1.843 |

The intersection point between the fitted accumulation mode 2 curve and coarse mode curve is located near 1.78 μm, indicating that the traditional 1 μm boundary is not applicable in our case. The fitted number concentrations provide further justification for our classification: at 1.981 μm, the concentrations for accumulation mode 2 and coarse mode are 3.708 $cm^{-3}$ and 11.22 $cm^{-3}$, respectively. The contribution of accumulation mode 2 remains non-negligible (> 25%). In contrast, at 2.129 μm, the concentrations of these two modes are 1.898 $cm^{-3}$ and 11.35 $cm^{-3}$, respectively. Here, the coarse mode is nearly an order of magnitude higher than the accumulation mode 2, indicating its overwhelming dominance.

The classification of particles at 1.981 μm is based on a practical assessment of modal contributions to avoid analytical bias. Although the coarse mode might influence the size distribution below 1.981 μm (Dp ≤ 1.981 μm; dn/dlgDp: 33.65 $cm^{-3}$), classifying particles from 1.78 to 1.981 μm as part of the accumulation mode (Dp ≤ 1.981 μm; dn/dlgDp: 840.38 $cm^{-3}$) has a negligible impact, because relative to the total NC of the accumulation mode aerosols with diameters ≤ 1.981 μm, the contribution from coarse mode particles within this size range is negligible. Conversely, classifying particles from 1.78 to 1.981 μm as part of the coarse mode (Dp ≥ 1.981 μm; dn/dlgDp: 53.40 $cm^{-3}$) could introduce a non-negligible impact, since the contribution from accumulation mode particles (Dp ≥ 1.981 μm; dn/dlgDp: 6.74 $cm^{-3}$) in this range would be misattributed to the coarse mode.

To better elucidate the distribution characteristics and controlling factors of aerosol number concentrations (NCs) over the South China Sea (SCS), particles with 0.5 μm ≤ Dp ≤ 1.98 μm are attributed to accumulation mode, and particles with Dp ≥ 2.129 μm (i.e. > 1.981 μm) as the coarse mode. It does not compromise the validity of our core conclusions (e.g., the negative correlation between aerosol NCs and the sea-air temperature difference).

[Figure]

Fig. 3 Number size distribution of marine aerosols with multi-mode log-normal fitting. The measured data are shown as gray circles. The orange, blue, and red curves denote the fitted log-normal distributions for Accumulation Mode 1, Accumulation Mode 2, and Coarse Mode, respectively. The light blue solid line indicates the overall fitted distribution (sum of all modes).

RC2.7. Line 240: "The total NC observed in this study (54 cm-3) was slightly lower than NC in the Western Pacific (83 cm-3) by Flores et al. (2020)" may be add average? Like average NC observed!
Response: Thanks very much for the suggestion. We have added "average" to this

sentence as suggested.

*The average summed NC observed in this study (54 cm⁻³, 0.5 μm ≤ Dp ≤ 1.98 μm) was slightly lower than the NC reported for the Western Pacific (83 cm⁻³, 0.1 μm ≤ Dp ≤ 1.98 μm) by Flores et al. (2020).*

RC2.8. What explains the decrease in dn/dlogdp (Figure 3a) between 5/26 and 5/31?

Response: We thank the reviewer for raising this point. The decrease in dn/dlogdp between 5/26 and 5/31 might be attributed to three concurrent factors. First, the apparent decrease is primarily attributable to the contrast between the exceptionally high values recorded during the heavy rainfall event on 5/26 and the subsequent lower baseline levels. As clearly shown in Figure 3a, the heavy precipitation on 5/26 led to a significant enhancement of dn/dlogdp. Second, ship-borne measurements revealed a substantial decline in aerosol NCs from May 27 to 29 (i.e. 27th: 50.52 cm⁻³, 28th: 23.04 cm⁻³, and 29th: 27.14 cm⁻³), which directly led to a general reduction in dn/dlogdp. Finally, the observed decrease in relative humidity (i.e. 27th: 84.4%, 28th: 79.0%, and 29th: 78.9%) likely weakened aerosol hygroscopic growth, further suppressing the dn/dlogdp values, particularly at larger sizes. The confluence of these factors might explain the decreasing dn/dlogdp values between 5/26 and 5/31.

[Figure]

Fig. 4 (c.f. Fig. 3 in the manuscript) The time series of the shipboard observations in the SCS from 21 May to 3 June 2023. The blue-shaded regions represented periods affected by rain events. (a) Trend of the aerosol size distributions. (b) Trends of NCs of the two aerosol particle modes (black solid line represented the NC of the coarse mode, and red solid line represented the NC of the accumulation mode). (c) Trend of the WD. (d) Trends of the TOBS (dark orange solid line), T2m (light orange solid line), and SST (blue solid line). (e) Trends in the RH (gray solid line), the VIS (red solid line), and the rainfall intensity (dark blue solid line).

RC2.9. Line 274: "For example, marine aerosol NCs generated by the bubble bursting process at low WS showed little variation, and the low WS was insufficient to activate spume droplet production." Are you citing literature? Because you didn't 'measure' bubble bursting to talk about it

Response: Thank you for your suggestion. The relevant citations (Pietsch et al., 2018; Russell et al., 2023) have been added to support the statement, as suggested. The manuscript has been revised accordingly.

*For example, marine aerosol NCs generated by the bubble bursting process at low WS showed little variation, and the low WS was insufficient to activate spume droplet production (Pietsch et al., 2018; Russell et al., 2023).*

*Reference:*

*Pietsch, R. B., Grothe, H., Hanlon, R., Powers, C. W., Jung, S., Ross, S. D., and Schmale Iii, D. G.: Wind-driven spume droplet production and the transport of Pseudomonas syringae from aquatic environments, PeerJ, 6, e5663, https://doi.org/10.7717/peerj.5663, 2018.*

*Russell, L. M., Moore, R. H., Burrows, S. M., and Quinn, P. K.: Ocean flux of salt, sulfate, and organic components to atmospheric aerosol, Earth-Sci. Rev., 239, 104364, https://doi.org/10.1016/J.EARSCIREV.2023.104364, 2023.*

RC2.10. Line 338: ", the number size distributions from offshore and pelagic areas exhibited close agreement, demonstrating consistent correlation patterns that remained robust against instrumentation limitations." Correlation patterns of what exactly?

Response: We thank the reviewer for raising this point regarding the lack of clarity. The "consistent correlation patterns" refer to those of the average number size distributions between offshore and pelagic regions. We agree that this phrase was ambiguous and could be misinterpreted.

Our intended meaning was simply that the average size distributions in offshore and pelagic regions showed a close agreement, and this agreement was robust despite instrumental uncertainties. To address this, we have revised the sentence by removing the ambiguous phrase, as follows:

*However, in the 5.0-10 μm particle size range, the number size distributions from offshore and pelagic regions were largely consistent, and this consistency remained robust against instrumentation limitations.*

RC2.11. Add your ship track in Figure 6a

Response: We appreciate your constructive suggestions. We have added the ship track in Fig. 6a.

[Figure]

*Fig. 5 (c.f. Fig. 6 in the manuscript) (a) The 72-h backward trajectory air mass source traces in the offshore (red solid lines) and pelagic (blue solid lines) regions. The light purple solid lines represented the ship track. (b) Detailed map of the backward trajectory air mass source traces passing through the mainland areas (© Google Earth). The white arrows represented the direction of air mass transport.*

RC2.12. Line 368: "cellular metabolism and lysis" whats lysis?

Response: Thank you for your comment. "Lysis" here refers to cellular lysis, i.e., the breakdown of phytoplankton cell structures to release intracellular contents. To improve clarity, we have revised the sentence to explicitly state that it means "cellular lysis".

*For instance, phytoplankton releases DMS through cellular metabolism and cellular lysis...*

RC2.13. Line 356: ".Higher WS can enhance marine aerosol production." You observed higher WS at offshore (~10m/s) and lower at pelagic (~8m/s). You correctly concluded there's more production of aerosols in offshore regions. But later you say there's more production in pelagic regions because of more sea salt! Contradicting. If you mean more aerosol production in offshore, which type of aerosols are getting produced offshore?

Response: Thanks very much for pointing out this critical contradiction, which has helped refine the precision of our scientific expression. We have revised the erroneous wording in the original manuscript as follows.

*The proportion of sea salt aerosol (SEAS$_{10}$; Dp ≤ 10 μm) in the pelagic regions (94.33 %) was higher than that in the offshore regions (91.9%), indicating a significant contribution of SEAS$_{10}$ to the total marine aerosols in the pelagic environments.*

Regarding the type of aerosols produced in offshore regions (driven by higher wind speed), SEAS$_{10}$ (Dp ≤ 10 μm) is still the primary wind-generated aerosol. Additionally,

DUST$_{10}$ (Dp $\leq$ 10 μm) in offshore regions may be generated by wind-driven resuspension of coastal sediments (e.g., sandy shorelines, intertidal deposits).

We apologize again for this misleading wording and have revised the text to clarify the distinction. The revised manuscript has resolved the contradiction in the conclusions, enhanced logical coherence, and improved the overall scientific quality of this study.

RC2.14. Can you make a supplemental figure showing backward trajectory at pelagic locations throughout the year? Just to understand if these regions never receive continental influence.

Response: We thank the reviewer for this suggestion. As requested, we have generated the backward trajectories for the full year of 2023 in the pelagic regions. Fig. 6a presents the full-year backward trajectories at a location about 51 km from the coast (pelagic region), and Fig. 6b shows the corresponding cluster analysis results, including the percentage contribution of each air mass pathway. Fig. 7a shows the full-year backward trajectories at a location about 400 km from the coast (pelagic region), and Fig. 7b shows the corresponding cluster analysis results, including the percentage contribution of each air mass pathway. The results indicate that these regions may still receive continental influence.

[Figure]

Fig. 6 (a) The 72-h backward trajectory air mass source traces (the white solid lines) for 2023, at a location about 51 km from the coast. (b) Detailed cluster analysis for backward trajectories of air masses (© Google Earth).

[Figure]

Fig. 7 (a) The 72-h backward trajectory air mass source traces (the white solid lines) for 2023, at a location about 400 km from the coast. (b) Detailed cluster analysis for backward trajectories of air masses (© Google Earth).